# Asynchronous Gradient Play in Zero-Sum Multi-agent Games

**Ruicheng Ao**
Peking University
archer_arc@pku.edu.cn

**Shicong Cen & Yuejie Chi**
Carnegie Mellon University
{shicongc,yuejiec}@andrew.cmu.edu

## Abstract

Finding equilibria via gradient play in competitive multi-agent games has been attracting a growing amount of attention in recent years, with emphasis on designing efficient strategies where the agents operate in a decentralized and symmetric manner with guaranteed convergence. While significant efforts have been made in understanding zero-sum two-player matrix games, the performance in zero-sum multi-agent games remains inadequately explored, especially in the presence of delayed feedbacks, leaving the scalability and resiliency of gradient play open to questions. In this paper, we make progress by studying *asynchronous gradient plays* in zero-sum polymatrix games under delayed feedbacks. We first establish that the last iterate of entropy-regularized optimistic multiplicative weight updates (OMWU) method converges linearly to the quantal response equilibrium (QRE), the solution concept under bounded rationality, in the absence of delays. While the linear convergence continues to hold even when the feedbacks are *randomly* delayed under mild statistical assumptions, it converges at a noticeably slower rate due to a smaller tolerable range of learning rates. Moving beyond, we demonstrate entropy-regularized OMWU—by adopting two-timescale learning rates in a delay-aware manner—enjoys faster last-iterate convergence under fixed delays, and continues to converge provably even when the delays are arbitrarily bounded in an average-iterate manner. Our methods also lead to finite-time guarantees to approximate the Nash equilibrium (NE) by moderating the amount of regularization. To the best of our knowledge, this work is the first that aims to understand asynchronous gradient play in zero-sum polymatrix games under a wide range of delay assumptions, highlighting the role of learning rates separation.

## 1 Introduction

Finding equilibria of multi-player games via gradient play lies at the heart of game theory, which permeates a remarkable breadth of modern applications, including but not limited to competitive reinforcement learning (RL) (Littman, 1994), generative adversarial networks (GANs) (Goodfellow et al., 2014) and adversarial training (Mertikopoulos et al., 2018). While conventional wisdom leans towards the paradigm of centralized learning (Bertsekas & Tsitsiklis, 1989), retrieving and sharing information across multiple agents raise questions in terms of both privacy and efficiency, leading to a significant amount of interest in designing decentralized learning algorithms that utilize only local payoff feedbacks, with the updates at different agents executed in a symmetric manner.

In reality, there is no shortage of scenarios where the feedback can be obtained only in a delayed manner (He et al., 2014), i.e., the agents only receive the payoff information sent from a previous round instead of the current round, due to communication slowdowns and congestions, for example. Substantial progress has been made towards reliable and efficient online learning with delayed feedbacks in various settings, e.g., stochastic multi-armed bandit (Pike-Burke et al., 2018; Vernade et al., 2017), adversarial multi-armed bandit (Cesa-Bianchi et al., 2016; Li et al., 2019), online convex optimization (Quanrud & Khashabi, 2015; McMahan & Streeter, 2014) and multi-player game (Meng et al., 2022; Héliou et al., 2020; Zhou et al., 2017). Typical approaches to combatting delays include subsampling the payoff history (Weinberger & Ordentlich, 2002; Joulani et al., 2013), or adopting

---

Author are sorted alphabetically.

| Learning rate | Type of delay | Iteration complexity | |
|---|---|---|---|
| | | $\epsilon$-QRE | $\epsilon$-NE |
| single-timescale | none | $\tau^{-1}d_{\mathsf{max}}\|A\|_\infty \log \epsilon^{-1}$ | $d_{\mathsf{max}}\|A\|_\infty \epsilon^{-1}$ |
| | statistical | $\tau^{-2}d_{\mathsf{max}}^2\|A\|_\infty^2 (\gamma+1)^2 \log \epsilon^{-1}$ | $d_{\mathsf{max}}^2\|A\|_\infty^2 (\gamma+1)^2\epsilon^{-2}$ |
| two-timescale | constant | $\tau^{-1}d_{\mathsf{max}}\|A\|_\infty (\gamma+1)^2 \log \epsilon^{-1}$ | $d_{\mathsf{max}}\|A\|_\infty (\gamma+1)^2\epsilon^{-1}$ |
| | bounded | $\tau^{-2}nd_{\mathsf{max}}^3\|A\|_\infty^3 (\gamma+1)^{5/2}\epsilon^{-1}$ | $nd_{\mathsf{max}}^3\|A\|_\infty^3 (\gamma+1)^{5/2}\epsilon^{-3}$ |

Table 1: Iteration complexities of the proposed OMWU method for finding $\epsilon$-QRE/NE of zero-sum polymatrix games, where logarithmic dependencies are omitted. Here, $\gamma$ denotes the maximal time delay when the delay is bounded, $n$ denotes the number of agents in the game, $d_{\mathsf{max}}$ is the maximal degree of the graph, and $\|A\|_\infty = \max_{i,j} \|A_{i,j}\|_\infty$ is the $\ell_\infty$ norm of the entire payoff matrix $A$ (over all games in the network). We only present the result under statistical delay when the delays are bounded for ease of comparison, while more general bounds are given in Section 3.2.

adaptive learning rates suggested by delay-aware analysis (Quanrud & Khashabi, 2015; McMahan & Streeter, 2014; Hsieh et al., 2020; Flaspohler et al., 2021). Most of these efforts, however, have been limited to either the asymptotic convergence to the equilibrium (Zhou et al., 2017; Héliou et al., 2020) or the study of *individual regret*, which characterizes the performance gap between an agent's learning trajectory and the best policy in hindsight. It remains highly inadequate when it comes to guaranteeing *finite-time convergence* to the equilibrium in a multi-player environment, especially in the presence of delayed feedbacks, thus leaving the scalability and resiliency of gradient play open to questions.

In this work, we initiate the study of asynchronous learning algorithms for an important class of games called zero-sum polymatrix games (also known as network matrix games (Bergman & Fokin, 1998)), which generalizes two-player zero-sum matrix games to the multiple-player setting and serves as an important stepping stone to more general multi-player general-sum games. Zero-sum polymatrix games are commonly used to describe situations in which agents' interactions are captured by an interaction graph and the entire system of games are closed so that the total payoffs keep invariant in the system. They find applications in an increasing number of important domains such as security games (Cai et al., 2016), graph transduction (Bernardi, 2021), and more.

In particular, we focus on *finite-time last-iterate* convergence to two prevalent solution concepts in game theory, namely Nash Equilibrium (NE) and Quantal Response Equilibrium (QRE) which considers bounded rationality (McKelvey & Palfrey, 1995). Despite the seemingly simple formulation, few existing works have achieved this goal even in the synchronous setting, i.e., with instantaneous feedback. Leonardos et al. (2021) studied a continuous-time learning dynamics that converges to the QRE at a linear rate. Anagnostides et al. (2022) demonstrated Optimistic Mirror Descent (OMD) (Rakhlin & Sridharan, 2013) enjoys finite-time last-iterate convergence to the NE, yet the analysis therein requires continuous gradient of the regularizer, which incurs computation overhead for solving a subproblem every iteration. In contrast, an appealing alternative is the entropy regularizer, which leads to closed-form multiplicative updates and is computationally more desirable, but remains poorly understood. In sum, designing efficient learning algorithms that provably converge to the game equilibria has been technically challenging, even in the synchronous setting.

## 1.1 OUR CONTRIBUTIONS

In this paper, we develop provably convergent algorithms—broadly dubbed as *asynchronous gradient play*—to find the QRE and NE of zero-sum polymatrix games in a decentralized and symmetric manner with delayed feedbacks. We propose an entropy-regularized Optimistic Multiplicative Weights Update (OMWU) method (Cen et al., 2021), where each player symmetrically updates their strategies without access to the payoff matrices and other players' strategies, and initiate a systematic investigation on the impacts of delays on its convergence under two schemes of learning rates schedule. Our main contributions are summarized as follows.

- *Finite-time last-iterate convergence of single-timescale OMWU.* We begin by showing that, in the synchronous setting, the single-timescale OMWU method—when the same learning rate is adopted for extrapolation and update—achieves last-iterate convergence to the QRE at a linear

rate, which is independent of the number of agents as well as the size of action spaces (up to logarithmic factors). In addition, this implies a last-iterate convergence to an $\epsilon$-approximate NE in $\widetilde{\mathcal{O}}(\epsilon^{-1})$ iterations by adjusting the regularization parameter, where $\widetilde{\mathcal{O}}(\cdot)$ hides logarithmic dependencies. While the last-iterate linear convergence to QRE continues to hold in the asynchronous setting, as long as the delay sequence follows certain mild statistical assumptions, it converges at a slower rate due to a smaller tolerable range of learning rates, with the iteration complexity to find an $\epsilon$-NE degenerating to $\widetilde{\mathcal{O}}(\epsilon^{-2})$.

- *Finite-time convergence of two-timescale OMWU.* To accelerate the convergence rate in the presence of delayed feedback, we propose a two-timescale OMWU method which separates the learning rates of extrapolation and update in a delay-aware manner for applications with constant and known delays (e.g. from timestamp information). The learning rate separation is critical in bypassing the convergence slowdown encountered in the single-timescale case, where we show that two-timescale OMWU achieves a *faster* last-iterate linear convergence to QRE in the presence of constant delays, with an improved $\widetilde{\mathcal{O}}(\epsilon^{-1})$ iteration complexity to $\epsilon$-NE that matches the rate without delay. We further tackle the more practical yet challenging setting where the feedback sequence is permutated by bounded delays—possibly in an adversarial manner—and demonstrate provable convergence to the equilibria in an average-iterate manner.

We summarize the iteration complexities of the proposed methods for finding $\epsilon$-approximate solutions of QRE and NE in Table 1. To the best of our knowledge, this work presents the first algorithm design and analysis that focus on equilibrium finding in a multi-player game with delayed feedbacks. In contrast, most of existing works concerning individual regret in the synchronous/asynchronous settings typically yield average-iterate convergence guarantees (see e.g., Bailey (2021); Meng et al. (2022)) and fall short of characterizing the actual learning trajectory to the equilibrium.

## 1.2 Notation and paper organization

Denote by $[n]$ the set $\{1, \cdots, n\}$ and by $\Delta(S)$ the probability simplex over the set $S$. Given two probability distributions $p, p' \in \Delta(S)$, the KL divergence from $p'$ to $p$ is defined by $\mathsf{KL}(p \, \| \, p') := \sum_{k \in S} p(k) \log \frac{p(k)}{p'(k)}$. For any vector $z = [z_i]_{1 \leq i \leq n} \in \mathbb{R}^n$, we use $\exp(z)$ to represent $[\exp(z_i)]_{1 \leq i \leq n}$. The rest of this paper is organized as follows. Section 2 provides the preliminary on zero-sum polymatrix games and solution concepts. Performance guarantees of single-timescale OMWU and two-timescale OMWU are presented in Section 3 and Section 4, respectively. Numerical experiments are provided in Section 5 to corroborate the theoretical findings, and finally, we conclude in Section 6. The proofs are deferred to the appendix.

## 2 Preliminaries

In this section, we introduce the formulation of zero-sum polymatrix games as well as the solution concept of NE and QRE. We start by defining the polymatrix game.

**Definition 1** (Polymatrix game). *Let $\mathcal{G} := \{(V, E), \{S_i\}_{i \in V}, \{A_{ij}\}_{(i,j) \in E}\}$ be an $n$-player polymatrix game, where each element in the tuple is defined as follows.*

- *An undirected graph $(V, E)$, with $V = [n]$ denoting the set of players and $E$ the set of edges;*

- *For each player $i \in V$, $S_i$ represents its action set, which is assumed to be finite;*

- *For each edge $(i,j) \in E$, $A_{ij} \in \mathbb{R}^{|S_i| \times |S_j|}$ and $A_{ji} \in \mathbb{R}^{|S_j| \times |S_i|}$ represent the payoff matrices associated with player $i$ and $j$, i.e., when player $i$ and player $j$ choose $s_i \in S_i$ and $s_j \in S_j$, the received payoffs are given by $A_{ij}(s_i, s_j)$, $A_{ji}(s_j, s_i)$, respectively.*

**Utility function.** Given the strategy profile $s = (s_1, \cdots, s_n) \in S = \prod_{i \in V} S_i$ taken by all players, the utility function $u_i : S \to \mathbb{R}$ of player $i$ is given by

$$u_i(s) = \sum_{j:(i,j) \in E} A_{ij}(s_i, s_j).$$

Suppose that player $i$ adopts a mixed/stochastic strategy or policy, $\pi_i \in \Delta(S_i)$, where the probability of selecting $s_i \in S_i$ is specified by $\pi_i(s_i)$. With slight abuse of notation, we denote the expected

utility of player $i$ with a mixed strategy profile $\pi = (\pi_1, \cdots, \pi_n) \in \Delta(S)$ as

$$u_i(\pi) = \mathbb{E}_{s_i \sim \pi_i, \forall i \in V} [u_i(s)] = \sum_{j:(i,j) \in E} \pi_i^\top A_{ij} \pi_j. \tag{1}$$

It turns out to be convenient to treat $\pi_i$ and $\pi$ as vectors in $\mathbb{R}^{|S_i|}$ and $\mathbb{R}^{\sum_{i \in V} |S_i|}$ without ambiguity, and concatenate all payoff matrices associated with player $i$ into

$$A_i = (A_{i1}, \cdots, A_{in}) \in \mathbb{R}^{|S_i| \times \sum_{j \in V} |S_j|}, \tag{2}$$

where $A_{ij}$ is set to 0 whenever $(i, j) \notin E$. In particular, it follows that $A_{ii} = 0$ for all $i \in V$. With these notation in place, we can rewrite the expected utility function (1) as

$$u_i(\pi) = \pi_i^\top A_i \pi, \tag{3}$$

where $A_i \pi \in \mathbb{R}^{|S_i|}$ can be interpreted as the expected utility of the actions in $S_i$ for player $i$. In addition, we denote the maximum entrywise absolute value of payoff by $\|A\|_\infty = \max_{i,j} \|A_{ij}\|_\infty = \max_i \|A_i\|_\infty$, and the maximum degree of the graph by $d_{\max} = \max_{i \in V} \deg_i$, where $\deg_i$ is the degree of player $i$. Moreover, we denote $S_{\max} = \max_i |S_i|$ as the maximum size of the action space over all players.

**Zero-sum polymatrix games.** The game $\mathcal{G}$ is a zero-sum polymatrix game if it holds that $\sum_{i \in V} u_i(s) = 0, \forall\ s \in S$. This immediately implies that for any strategy profile $\pi \in \Delta(S)$, it follows that $\sum_{i \in V} u_i(\pi) = 0$.

**Nash equilibrium (NE).** A mixed strategy profile $\pi^\star = (\pi_1^\star, \cdots, \pi_n^\star)$ is a Nash equilibrium (NE) when each player $i$ cannot further increase its own utility function $u_i$ by unilateral deviation, i.e., $u_i(\pi_i', \pi_{-i}^\star) \le u_i(\pi_i^\star, \pi_{-i}^\star)$, for all $i \in V$, $\pi_i' \in \Delta(S_i)$, where the existence is guaranteed by the work (Cai et al., 2016). Here we denote the mixed strategies of all players other than $i$ by $\pi_{-i}$ and write $u_i(\pi_i, \pi_{-i}) = u_i(\pi)$. To measure how close a strategy $\pi \in \Delta(S)$ is to an NE, we introduce

$$\texttt{NE-Gap}(\pi) = \max_{i \in V} \left[ \max_{\pi_i' \in \Delta(S_i)} u_i(\pi_i', \pi_{-i}) - u_i(\pi) \right],$$

which measures the largest possible gain in the expected utility when players deviate from its strategy unilaterally. A mixed strategy profile $\pi$ is called an $\epsilon$-approximate Nash equilibrium ($\epsilon$-NE) when $\texttt{NE-Gap}(\pi) \le \epsilon$, which ensures that $u_i(\pi_i', \pi_{-i}) \le u_i(\pi_i, \pi_{-i}) + \epsilon$, for all $i \in V$, $\pi_i' \in \Delta(S_i)$.

**Quantal response equilibrium (QRE).** The quantal response equilibrium (QRE), proposed by McKelvey & Palfrey (1995), generalizes the classical notion of NE under uncertain payoffs or bounded rationality, while balancing exploration and exploitation. A mixed strategy profile $\pi_\tau^\star = (\pi_{1,\tau}^\star, \cdots, \pi_{n,\tau}^\star)$ is a QRE when each player assigns its probability of action according to the expected utility of every action in a Boltzmann fashion, i.e., for all $i \in V$,

$$\pi_{i,\tau}^\star(k) = \frac{\exp([A_i \pi_\tau^\star]_k / \tau)}{\sum_{k \in S_i} \exp([A_i \pi_\tau^\star]_k / \tau)}, \qquad k \in S_i, \tag{4}$$

where $\tau > 0$ is the regularization parameter or temperature. Equivalently, this amounts to maximizing an entropy-regularized utility of each player (Mertikopoulos & Sandholm, 2016), i.e., $u_{i,\tau}(\pi_i', \pi_{-i,\tau}^\star) \le u_{i,\tau}(\pi_{i,\tau}^\star, \pi_{-i,\tau}^\star)$ for all $i \in V$, $\pi_i' \in \Delta(S_i)$. Here, the entropy-regularized utility function $u_i : S \to \mathbb{R}$ of player $i$ is given by

$$u_{i,\tau}(\pi) = u_i(\pi) + \tau \mathcal{H}(\pi_i), \tag{5}$$

where $\mathcal{H}(\pi_i) = -\pi_i^\top \log \pi_i$ denotes the Shannon entropy of $\pi_i$. In Leonardos et al. (2021), it is shown that a unique QRE exists in a zero-sum polymatrix game. Similarly, we can measure the proximity of a strategy $\pi$ to a QRE by

$$\texttt{QRE-Gap}_\tau(\pi) = \max_{i \in V} \left[ \max_{\pi_i' \in \Delta(S_i)} u_{i,\tau}(\pi_i', \pi_{-i}) - u_{i,\tau}(\pi) \right]. \tag{6}$$

A mixed strategy profile $\pi$ is called an $\epsilon$-QRE when $\texttt{QRE-Gap}_\tau(\pi) \le \epsilon$. According to the straightforward relationship $\texttt{NE-Gap}(\pi) \le \texttt{QRE-Gap}_\tau(\pi) + \tau \log S_{\max}$, it follows immediately that we can link an $\epsilon/2$-QRE to $\epsilon$-NE by setting $\tau = \frac{\epsilon}{2 \log S_{\max}}$. This facilitates the translation of convergence to the QRE to one regarding the NE by appropriately setting the regularization parameter $\tau$.

---

**Algorithm 1** Entropy-regularized OMWU, agent $i$

---

1: Initialize $\pi_i^{(0)} = \overline{\pi}_i^{(0)}$ as uniform distribution. Learning rates $\eta$, and $\overline{\eta}$ (optional).
2: **for** $t = 0, 1, 2, \ldots$ **do**
3:     Receive payoff vector $A_i \overline{\pi}^{(\kappa_i^{(t)})}$.
4:     When $t \geq 1$, update $\pi_i$ according to

$$\pi_i^{(t)}(k) \propto \pi_i^{(t-1)}(k)^{1-\eta\tau} \exp(\eta[A_i \overline{\pi}^{(\kappa_i^{(t)})}]_k), \qquad \forall k \in S_i.$$

5:     Update $\overline{\pi}_i$ according to the single-timescale rule

$$\overline{\pi}_i^{(t+1)}(k) \propto \pi_i^{(t)}(k)^{1-\eta\tau} \exp(\eta[A_i \overline{\pi}^{(\kappa_i^{(t)})}]_k), \qquad \forall k \in S_i. \tag{9}$$

    or the two-timescale rule

$$\overline{\pi}_i^{(t+1)}(k) \propto \pi_i^{(t)}(k)^{1-\overline{\eta}\tau} \exp(\overline{\eta}[A_i \overline{\pi}^{(\kappa_i^{(t)})}]_k), \qquad \forall k \in S_i. \tag{10}$$

6: **end for**

---

## 3    PERFORMANCE GUARANTEES OF SINGLE-TIMESCALE OMWU

In this section, we present and study the entropy-regularized OMWU method (Cen et al., 2021) for finding the QRE of zero-sum polymatrix games. Whilst the method is originally proposed for finding QRE in a two-player zero-sum game, the update rule naturally generalizes to the multi-player setting as

$$\pi_i^{(t+1)}(k) \propto \pi_i^{(t)}(k)^{1-\eta\tau} \exp(\eta[A_i \overline{\pi}^{(t+1)}]_k), \qquad \forall k \in S_i, \tag{7}$$

where $\eta > 0$ is the learning rate and $\overline{\pi}^{(t+1)}$ serves as a prediction for $\pi^{(t+1)}$ via an extrapolation step

$$\overline{\pi}_i^{(t+1)}(k) \propto \pi_i^{(t)}(k)^{1-\eta\tau} \exp(\eta[A_i \overline{\pi}^{(t)}]_k), \qquad \forall k \in S_i. \tag{8}$$

In the asynchronous setting, however, each agent $i$ receives a delayed payoff vector $A_i \overline{\pi}^{(\kappa_i^{(t)})}$ instead of $A_i \overline{\pi}^{(t)}$ in the $t$-th iteration, where $\kappa_i^{(t)} = \max\{t - \gamma_i^{(t)}, 0\}$, with $\gamma_i^{(t)} \geq 0$ representing the length of delay. The detailed procedure is outlined in Algorithm 1 using the single-timescale rule (9) for extrapolation.

### 3.1   PERFORMANCE GUARANTEES WITHOUT DELAYS

We first present our theorem concerning the last-iterate convergence of single-timescale OMWU for finding the QRE in the synchronous setting, i.e. $\gamma_i^{(t)} = 0$ for all $i \in V$ and $t \geq 0$. For any $\pi, \pi' \in V$, let $\mathsf{KL}(\pi \,\|\, \pi') = \sum_{i \in V} \mathsf{KL}(\pi_i \,\|\, \pi_i')$.

**Theorem 1** (Last-iterate convergence without delays). *Suppose that the learning rate $\eta$ of single-timescale OMWU in Algorithm 1 obeys $0 < \eta \leq \min\left\{\frac{1}{2\tau}, \frac{1}{4d_{\max}\|A\|_\infty}\right\}$, then for any $T \geq 0$, the iterates $\pi^{(T)}$ and $\overline{\pi}^{(Ts)}$ converge at a linear rate according to*

$$\mathsf{KL}(\pi_\tau^\star \,\|\, \pi^{(T)}) \leq (1 - \eta\tau)^T \mathsf{KL}(\pi_\tau^\star \,\|\, \pi^{(0)}), \quad \mathsf{KL}(\pi_\tau^\star \,\|\, \overline{\pi}^{(T+1)}) \leq 2(1 - \eta\tau)^T \mathsf{KL}(\pi_\tau^\star \,\|\, \pi^{(0)}). \tag{11a}$$

*Furthermore, the QRE-gap also converges linearly according to*

$$QRE\text{-}Gap_\tau(\overline{\pi}^{(T)}) \leq \left(\eta^{-1} + 2\tau^{-1}d_{\max}^2\|A\|_\infty^2\right)(1 - \eta\tau)^{T-1}\mathsf{KL}(\pi_\tau^\star \,\|\, \pi^{(0)}). \tag{11b}$$

Theorem 1 demonstrates that as long as the learning rate $\eta$ is sufficiently small, the last iterate of single-timescale OMWU converges to the QRE at a linear rate. Compared with prior works for finding approximate equilibrium for zero-sum polymatrix games, our approach features a closed-form multiplicative update and a fast linear last-iterate convergence. Some remarks are in order.

- *Linear convergence to the QRE.* Theorem 1 implies an iteration complexity of $\widetilde{\mathcal{O}}\left(\frac{1}{\eta\tau} \log \frac{1}{\epsilon}\right)$ for finding an $\epsilon$-QRE in a last-iterate manner, which leads to an iteration complexity of

$\widetilde{\mathcal{O}}\left(\left(\frac{d_{\max}\|A\|_{\infty}}{\tau}+1\right)\log\frac{1}{\epsilon}\right)$ by optimizing the learning rate in Theorem 1.The result is especially appealing as it avoids direct dependency on the number of agents $n$ as well as the size of action spaces (up to logarithmic factors), suggesting that learning in competitive multi-agent games can be made quite scalable as long as the interactions among the agents are sparse (so that the maximum degree of the graph $d_{\max}$ is much smaller than the number of agents $n$).

- *Last-iterate convergence to $\epsilon$-NE.* By setting $\tau$ appropriately, we end up with an iteration complexity of $\widetilde{\mathcal{O}}\left(\frac{d_{\max}\|A\|_{\infty}}{\epsilon}\right)$ for achieving last-iterate convergence to an $\epsilon$-NE, which outperforms the best existing last-iterate rate of $\widetilde{\mathcal{O}}\left(n\|A\|_{\infty}/\epsilon^2\right)$ from Leonardos et al. (2021) by at least a factor of $n/(d_{\max}\epsilon)$.

**Remark 1.** *Our results trivially extend to the setting of weighted zero-sum polymatrix games (Leonardos et al., 2021), which amounts to adopting different learning rates $\{\eta_i\}_{i\in V}$ at each player. In this case, the iteration complexity becomes $\widetilde{\mathcal{O}}\left(\max_{i\in V}\frac{1}{\eta_i\tau}\log\frac{1}{\epsilon}\right)$. In addition, our convergence result readily translates to a bound on individual regret as detailed in Appendix C.*

### 3.2 PERFORMANCE GUARANTEES UNDER RANDOM DELAYS

We continue to examine single-timescale OMWU in the more challenging asynchronous setting. In particularly, we show that the last iterate of single-timescale OMWU continues to converge linearly to the QRE at a slower rate, as long as the delays satisfy some mild statistical assumptions given below.

**Assumption 1** (Random delays). *Assume that for all $i\in V$, $t\geq 0$, the delay $\gamma_i^{(t)}$ is independently generated and satisfies*

$$\mathbb{E}_{\gamma_i^{(t)}\geq\ell}\left[\gamma_i^{(t)}\right]:=\mathbb{E}\left[\gamma_i^{(t)}\mid\gamma_i^{(t)}\geq\ell\right]\leq E(\ell),\qquad\forall\ell=0,1,\dots. \tag{12}$$

*Additionally, there exists some constant $\zeta>1$, such that $L\triangleq\sum_{\ell=0}^{\infty}\zeta^{\ell}E(\ell)<\infty$.*

We remark that Assumption 1 is a rather mild condition that applies to typical delay distributions, such as the Poisson distribution (Zhang et al., 2020), as well as distributions with bounded support (Recht et al., 2011; Liu et al., 2014; Assran et al., 2020). Roughly speaking, Assumption 1 implies that the probability of the delay decays exponentially with its length, where $\zeta^{-1}$ approximately indicates the decay rate. We have the following theorem.

**Theorem 2** (Last-iterate convergence with random delays). *Under Assumption 1, suppose that the regulari-zation parameter $\tau<\min\{1,d_{\max}\|A\|_{\infty}\}$ and the learning rate $\eta$ of single-timescale OMWU in Algorithm 1 obeys*

$$0<\eta\leq\min\left\{\frac{\tau}{24d_{\max}^2\|A\|_{\infty}^2(L+1)},\frac{\zeta-1}{\tau\zeta}\right\}, \tag{13}$$

*then for any $T\geq 1$, the iterates $\pi^{(T)}$ and $\overline{\pi}^{(T)}$ converges to $\pi_{\tau}^{\star}$ at the rate*

$$\max\left\{\mathbb{E}\left[\mathsf{KL}\big(\pi_{\tau}^{\star}\|\pi^{(T)}\big)\right],\frac{1}{2}\mathbb{E}\left[\mathsf{KL}\big(\pi_{\tau}^{\star}\|\overline{\pi}^{(T)}\big)\right]\right\}\leq(1-\eta\tau)^T\,\mathsf{KL}\big(\pi_{\tau}^{\star}\|\pi^{(0)}\big). \tag{14a}$$

*Furthermore, the QRE-gap also converges linearly according to*

$$\mathbb{E}\left[QRE\text{-}Gap_{\tau}(\overline{\pi}^{(T)})\right]\leq 4\eta^{-1}(1-\eta\tau)^T\mathsf{KL}\big(\pi_{\tau}^{\star}\|\pi^{(0)}\big). \tag{14b}$$

Theorem 2 suggests that the iteration complexity to $\epsilon$-QRE is no more than $\widetilde{\mathcal{O}}\left(\max\left\{d_{\max}^2\|A\|_{\infty}^2(L+1),\frac{\zeta}{\zeta-1}\right\}\frac{1}{\tau^2}\log\frac{1}{\epsilon}\right)$ after optimizing the learning rate, whose range is more limited compared with the requirement in Theorem 1without delays. In particular, the range of the learning rate is proportional to the regularization parameter $\tau$, an issue we shall try to address by resorting to two-timescale learning rates in OMWU. To facilitate further understanding, we showcase the iteration complexity for finding $\epsilon$-QRE/NE under two typical scenarios: bounded delay and Poisson delay.

- *Bounded random delay.* When the delays are bounded above by some maximum delay $\gamma$, Assumption 1 is met with $\zeta = 1 + \gamma^{-1}$ and $L = e\gamma(\gamma + 1)$. Plugging into Theorem 2 yields an iteration complexity of $\widetilde{\mathcal{O}}\left(\frac{d_{\mathsf{max}}^2 \|A\|_\infty^2 (\gamma+1)^2}{\tau^2} \log \frac{1}{\epsilon}\right)$ for finding an $\epsilon$-QRE, or $\widetilde{\mathcal{O}}\left(\frac{d_{\mathsf{max}}^2 \|A\|_\infty^2 (\gamma+1)^2}{\epsilon^2}\right)$ for finding an $\epsilon$-NE, which increases quadratically as the maximum delay increases. Note that these rates are worse than those without delays (cf. Theorem 1).

- *Poisson delay.* When the delays follow the Poisson distribution with parameter $1/\overline{T}$, it suffices to set $\zeta = 1 + \overline{T}^{-1}$ and $L = e\overline{T}(1 + \overline{T})$ Assumption 1. This leads to an iteration complexity of $\widetilde{\mathcal{O}}\left(\frac{d_{\mathsf{max}}^2 \|A\|_\infty^2 \overline{T}^2}{\tau^2} \log \frac{1}{\epsilon}\right)$ for finding an $\epsilon$-QRE, or $\widetilde{\mathcal{O}}\left(\frac{d_{\mathsf{max}}^2 \|A\|_\infty^2 \overline{T}^2}{\epsilon^2}\right)$ for finding an $\epsilon$-NE, which is similar to the bounded random delay case.

# 4 PERFORMANCE GUARANTEES OF TWO-TIMESCALE OMWU

While Theorem 2 demonstrates provable convergence of single-timescale OMWU with random delays, it remains unclear whether the update rule can be better motivated in more general asynchronous settings, and whether the convergence can be further ensured under adversarial delays. Indeed, theoretical insights from previous literature (Mokhtari et al., 2020; Cen et al., 2021) suggest the critical role of $\overline{\pi}^{(t)}$ as a predictive surrogate for $\pi^{(t)}$ in enabling fast convergence, which no longer holds when $\overline{\pi}^{(t)}$ is replaced by a delayed feedback from $\overline{\pi}^{(\kappa_i^{(t)})}$. To this end, we propose to replace the extrapolation update (9) with one equipped with a different learning rate:

$$\overline{\pi}_i^{(t+1)}(k) \propto \pi_i^{(t)}(k)^{1-\overline{\eta}\tau} \exp(\overline{\eta}[A_i \overline{\pi}^{(\kappa_i^{(t)})}]_k), \qquad \forall k \in S_i, \tag{15}$$

which adopts a larger learning rate $\overline{\eta} > \eta$ to counteract the delay. Intuitively, a choice of $\overline{\eta} \approx (\gamma_i^{(t)} + 1)\eta$ would allow $\overline{\pi}^{(\kappa_i^{(t)})}$ to approximate $\pi^{(t)}$ by taking the intermediate updates $\{\pi^{(l)} : \kappa_i^{(t)} \le l < t\}$ into consideration. We refer to this update rule as the two-timescale entropy-regularized OMWU, whose detailed procedure is again outlined in Algorithm 1 using (10) for extrapolation.

## 4.1 PERFORMANCE GUARANTEES UNDER CONSTANT AND KNOWN DELAYS

To highlight the potential benefit of learning rate separation, we start by studying the convergence of two-timescale OMWU in the asynchronous setting with constant and known delays, which has been studied in (Weinberger & Ordentlich, 2002; Flaspohler et al., 2021; Meng et al., 2022). We have the following theorem, which reveals a a faster linear convergence to the QRE by using a delay-aware two-timescale learning rate design.

**Theorem 3** (Last-iterate convergence with fixed delays). *Suppose that the delays $\gamma_i^{(t)} = \gamma$ are fixed and known. Suppose that the learning rate $\eta$ of two-timescale OMWU in Algorithm 1 satisfies*

$$\eta \le \min\left\{\frac{1}{2\tau(\gamma+1)}, \frac{1}{5d_{\mathsf{max}} \|A\|_\infty (\gamma+1)^2}\right\}$$

*and $\overline{\eta}$ is determined by $1 - \overline{\eta}\tau = (1 - \eta\tau)^{(\gamma+1)}$, then the last iterate $\pi^{(t)}$ and $\overline{\pi}^{(t)}$ converge to the QRE at a linear rate: for $T \ge \gamma$,*

$$\max\left\{\mathsf{KL}\big(\pi_\tau^\star \,\|\, \pi^{(T+1)}\big), \frac{1}{2}\mathsf{KL}\big(\pi_\tau^\star \,\|\, \overline{\pi}^{(T-\gamma+1)}\big)\right\} \le (1 - \eta\tau)^{T+1}\mathsf{KL}\big(\pi_\tau^\star \,\|\, \pi^{(0)}\big) + (1 - \eta\tau)^{T+1-\gamma}.$$

*In addition, the QRE-gap converges linearly according to*

$$QRE\text{-}Gap_\tau(\overline{\pi}^{(T-\gamma+1)}) \le 2\max\left\{\frac{d_{\mathsf{max}}^2 \|A\|_\infty^2}{\tau}, \frac{1}{\eta}\right\}\Big((1 - \eta\tau)^{T+1}\mathsf{KL}\big(\pi_\tau^\star \,\|\, \pi^{(0)}\big) + (1 - \eta\tau)^{T+1-\gamma}\Big).$$

By optimizing the learning rate $\eta$, Theorem 3 implies that two-timescale OMWU takes at most $\widetilde{\mathcal{O}}\left(\frac{d_{\mathsf{max}} \|A\|_\infty (\gamma+1)^2}{\tau} \log \frac{1}{\epsilon}\right)$ iterations to find an $\epsilon$-QRE in a last-iterate manner, which translates to an iteration complexity of $\widetilde{\mathcal{O}}\left(\frac{d_{\mathsf{max}} \|A\|_\infty (\gamma+1)^2}{\epsilon}\right)$ for finding an $\epsilon$-NE. This significantly improves over the iteration complexity of $\widetilde{\mathcal{O}}\big(d_{\mathsf{max}}^2 \|A\|_\infty^2 (\gamma+1)^2/\epsilon^2\big)$ for single-timescale OMWU, verifying the positive role of adopting two-timescale learning rate in enabling faster convergence.

## 4.2 PERFORMANCE GUARANTEES WITH PERMUTED BOUNDED DELAYS

The above result requires the exact information of the delay, which may not always be available. Motivated by the need to address arbitrary or even adversarial delays, we consider a more realistic scenario, where the payoff sequence arrives in a permuted order (Agarwal & Duchi, 2011) constrained by a maximum bounded delay (McMahan & Streeter, 2014; Wan et al., 2022).

**Assumption 2** (Bounded delay). *For any $i \in V$ and $t > 0$, it holds that $\gamma_i^{(t)} \leq \gamma$.*

**Assumption 3** (Permuted feedback). *For any $t > 0$, the payoff vector at the $t$-th iteration is received by agent $i$ only once. The payoff at the $0$-th iteration can be used multiple times.*

The following theorem unveils the convergence of two-timescale OMWU to the QRE in an average sense under permutated bounded delays.

**Theorem 4** (Average-iterate convergence under permutated delays). *Under Assumption 2 and 3, suppose that the learning rate $\eta$ of two-timescale OMWU in Algorithm 1 satisfies $\eta \leq \min\left\{\frac{1}{2\tau(\gamma+1)}, \frac{1}{28 d_{\max}\|A\|_\infty(\gamma+1)^{5/2}}\right\}$, and $\overline{\eta}$ is determined by $1 - \overline{\eta}\tau = (1 - \eta\tau)^{(\gamma+1)}$, then for $T > 2\gamma$, it holds that*

$$\frac{1}{T-2\gamma} \max\left\{ \sum_{t=2\gamma}^{T-1} \mathsf{KL}\big(\pi_\tau^\star \,\|\, \pi^{(t+1)}\big), \frac{1}{3} \sum_{t=2\gamma}^{T-1} \mathsf{KL}\big(\pi_\tau^\star \,\|\, \overline{\pi}^{(t-\gamma+1)}\big) \right\}$$

$$\leq \frac{1}{\eta\tau(T-2\gamma)} \left( \mathsf{KL}\big(\pi_{i,\tau}^\star \,\|\, \pi_i^{(0)}\big) + n \right) + \frac{24n\gamma \log S_{\max}}{T-2\gamma}. \tag{16}$$

*Furthermore, the average QRE-gap can be bounded by*

$$\frac{1}{T-2\gamma} \sum_{t=2\gamma}^{T-1} QRE\text{-}Gap_\tau\big(\overline{\pi}^{(t+1)}\big)$$

$$\leq \frac{1}{T-2\gamma} \max\left\{ \frac{3 d_{\max}^2 \|A\|_\infty^2}{2\tau}, \tau \right\} \left( \frac{1}{\eta\tau}\big(\mathsf{KL}\big(\pi_{i,\tau}^\star \,\|\, \pi_i^{(0)}\big) + n\big) + 36n\gamma \log S_{\max} \right).$$

Theorem 4 guarantees that the best iterate among $\{\overline{\pi}^{(t)}\}_{2\gamma < t \leq T}$ is an $\epsilon$-QRE as long as $T$ is on the order of $\widetilde{\mathcal{O}}\left(\frac{n d_{\max}^3 \|A\|_\infty^3 (\gamma+1)^{5/2}}{\tau^2 \epsilon}\right)$, which translates to an iteration complexity of $\widetilde{\mathcal{O}}\left(\frac{n d_{\max}^3 \|A\|_\infty^3 (\gamma+1)^{5/2}}{\epsilon^3}\right)$ for finding an $\epsilon$-NE. While the rate seems slower than the previous theorems, Theorem 4 holds under arguably the weakest delay assumptions, where it can be even adversarially bounded. We remark that the result in (16) also guarantees the convergence of the last iterate $\overline{\pi}^{(t)}$ to the QRE asymptotically, although without a finite-time rate. This is in sharp contrast to typical average-iterate analysis that only applies to $\frac{1}{T}\sum_{t=1}^T \overline{\pi}^{(t)}$ without implications on the convergence of the last iterate $\overline{\pi}^{(t)}$.

**Remark 2.** *The analysis in this section can be generalized to more commonly-used delay models where the reward information is not assumed to be observed once per round (Quanrud & Khashabi, 2015; Joulani et al., 2013), i.e., in every round an agent may observe multiple reward feedbacks from previous iterations or receive no information. This can be achieved by storing reward feedbacks in a buffer memory and picking one for policy update every round in a First-In-First-Out manner.*

## 5 NUMERICAL EXPERIMENTS

In this section, we verify our theoretical findings by investigating the performance of both single-timescale and two-timescale OMWU on randomly generated zero-sum entropy-regularized polymatrix games with $n = 10$, $|S_i| = 10, i \in V$ and $\tau = 0.1$. For each $(i, j) \in E$, we set $A_{ij} = -A_{ji}^\top$ with entries of $A_{ij}$ independently sampled from the uniform distribution over $[-1, 1]$. All the results are averaged over five independent runs.

In Fig. 1 (a), we compare the performance of single-timescale OMWU in both synchronous and asynchronous settings, with delay uniformly sampled from $\{0, 1, \ldots, 10\}$. We adopt the optimal learning rate $\eta$ from $\{0.1, 0.05, 0.02, 0.01, \ldots\}$ that yields the highest accuracy. The method

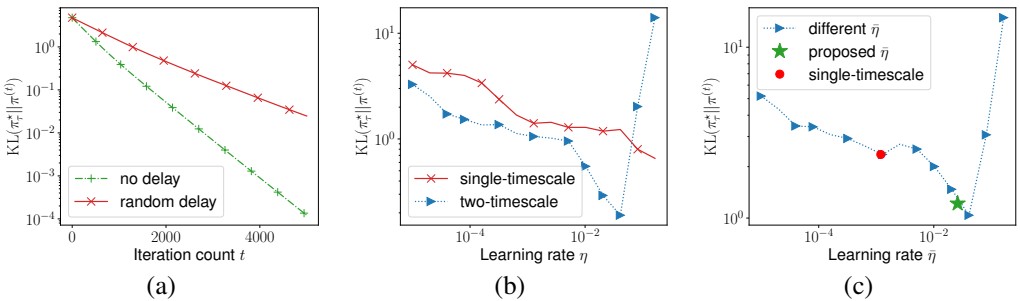

Figure 1: $\mathsf{KL}\big(\pi_\tau^\star \,\|\, \pi^{(t)}\big)$ of single-timescale and two-timescale OMWU with respect to different values of learning rate and delay. (a): performance of single-timescale OMWU in the synchronous setting and asynchronous setting. (b) & (c): performance of the two methods after 5000 iterations under various choices of $\eta$ and $\bar\eta$, with $\bar\eta$ fixed to $\bar\eta = \tau^{-1}(1 - (1 - \eta\tau)^{\gamma+1})$ in (b) and $\eta$ fixed to 0.001 in (c).

achieves linear convergence in both cases, yet the convergence rate is slowed down by delayed feedbacks in the asynchronous setting. Fig. 1 (b) and (c) compare the effect of different choices of learning rates $\eta, \bar\eta$ on the performance of the proposed methods, where the feedback is permuted with bounded delay $\gamma = 25$ (cf. Assumptions 2 and 3). In general, two-timescale OMWU outperforms single-timescale OMWU given appropriate choices of learning rate $\eta$. On the other hand, (c) demonstrates that the choice of $\bar\eta = \tau^{-1}(1 - (1 - \eta\tau)^{\gamma+1})$ suggested by the theory (marked with star) indeed leads to near-optimal performance of two-timescale OMWU.

Figure 2 shows $\mathsf{KL}\big(\pi_\tau^\star \,\|\, \pi^{(t)}\big)$ with respect to the number of iterations of single-timescale and two-timescale OMWU under different asynchronous scenarios, with optimal choices of $\eta$ and $\tau = 0.1$. In particular, two-timescale OMWU adopts the extrapolation learning rate suggested by theory $\bar\eta = \tau^{-1}(1 - (1 - \eta\tau)^{\gamma+1})$. While both methods yield linear convergence to the QRE, two-timescale method outperforms its single-timescale counterpart in the case with constant and known delay and the case where the feedback is permuted with bounded delay, which verifies our theory.

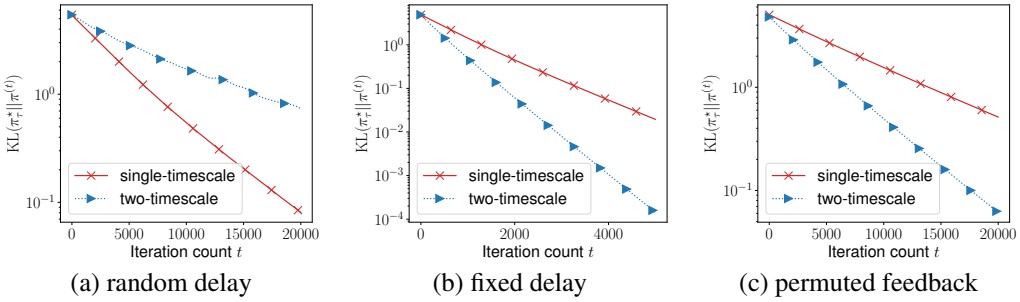

Figure 2: $\mathsf{KL}\big(\pi_\tau^\star \,\|\, \pi^{(t)}\big)$ with respect to iteration count $t$ of single-timescale and two-timescale OMWU under various asynchronous settings. (a): random delays bounded by $\gamma = 25$. (b): constant delays $\gamma = 50$. (c): permuted feedback with delay bounded by $\gamma = 25$.

## 6  CONCLUSION

This paper studies asynchronous gradient play in zero-sum polymatrix games, by investigating the convergence behaviors of entropy-regularized OMWU with delayed feedbacks under two different schedules of the learning rates. We demonstrate that single-timescale OMWU enjoys a linear last-iterate convergence to the QRE even under mild statistical delays. However, the presence of the delay noticeably limits the allowable range of learning rates and slows down the convergence. To mitigate the impact, we further show that the method benefits from adopting a two-timescale learning rate in a delay-aware manner, which achieves a faster last-iterate convergence when the delay is fixed and known, and continues to converge provably even when the delays are arbitrarily bounded in an average-iterate manner. We believe our work lays the foundation for further understandings of delayed feedback in games under symmetric and independent learning.

ACKNOWLEDGEMENT

S. Cen and Y. Chi are supported in part by the grants ONR N00014-19-1-2404, NSF CCF-1901199, CCF-2106778 and CNS-2148212. S. Cen is also gratefully supported by Wei Shen and Xuehong Zhang Presidential Fellowship, and Nicholas Minnici Dean's Graduate Fellowship in Electrical and Computer Engineering at Carnegie Mellon University. R. Ao is supported by the Elite Undergraduate Training Program of School of Mathematical Sciences at Peking University.

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

# Appendix

## Table of Contents

## A    FURTHER RELATED WORKS

**Learning in two-player zero-sum matrix games.**    Freund & Schapire (1999) proved that Multiplicative Weights Update (MWU) method achieve an average-iterate convergence rate of $\mathcal{O}(1/\sqrt{T})$ through the lens of regret analysis. Daskalakis et al. (2011) is the first to achieve an optimal convergence rate of $\mathcal{O}(1/T)$ with the excessive gap technique of Nesterov (Nesterov, 2005a;b). Rakhlin & Sridharan (2013) achieved the same rate with OMD, which is more commonly referred to as OMWU when entropy regularization is in use for the mirror descent update rule. In terms of last-iterate convergence, Daskalakis & Panageas (2018) established asymptotic last-iterate convergence for OMWU assuming the uniqueness of NE. Wei et al. (2021) improved upon the analysis under the same assumption by showing a problem-dependent linear rate of convergence, which is extended to a class of extensive-form games (Lee et al., 2021). Cen et al. (2021) showed that entropy-regularized OMWU converges linearly to the QRE of two-player zero-sum matrix game, which translates to an iteration complexity of $\widetilde{\mathcal{O}}(1/T)$ for finding an $\epsilon$-NE, without assuming its uniqueness; the linear convergence to the QRE continues to hold with smooth value updates (Cen et al., 2022). Sokota et al. (2022) showed that linear convergence to QRE can be achieved without resorting to optimistic update rules, e.g., using entropy-regularized MWU, albeit with a more restrictive learning rate. It is worth pointing out that the idea of learning rate separation has been explored for equilibrium finding in two-player zero-sum games with instant feedback (Fasoulakis et al., 2022) and online learning with delayed feedback (Hsieh et al., 2020), but lacks study in an asynchronous multi-player game setting.

**Asynchronous optimization.** Asynchronous and decentralized optimization algorithms have been extensively studied since the proposal of Bertsekas & Tsitsiklis (1989), where a number of agents seek to find an approximate global optimizer of a common loss function, by performing iterative gradient-based methods in a collaborative manner. Typical approaches including parallelizing the computation of gradient with regard to data (Tong et al., 2020), or parallelizing the model updates by imposing coordinate update rules (Nesterov, 2012; Liu et al., 2014; Liu & Wright, 2015). Delayed gradient (feedback) is also common in these scenarios due to the existence of other agents updating the model. Moreover, the zero-sum polymatrix setting considered in this work is inherently non-collaborative by requiring every agent to maximize its own utility function and compete with other agents, and leads to substantially difference analysis techniques.

## B  PROOF FOR SINGLE-TIMESCALE OMWU (SECTION 3)

Before delving into the main proof, we first record a useful lemma pertaining to a basic property of zero-sum polymatrix games; the proof is deferred to Appendix E.1. For $i \in V$, we denote by $\mathcal{N}_i = \{j : (i, j) \in E\}$ the neighbors of agent $i$ in the graph $(V, E)$. For notational simplicity, we denote by $x \overset{1}{=} y$ the equivalence between two vectors $x$ and $y$ up to a global shift, i.e.,

$$x = y + c \cdot \mathbf{1} \tag{17}$$

for some constant $c \in \mathbb{R}$, where $\mathbf{1}$ is the all-one vector.

**Lemma 1.** *For any zero-sum polymatrix game $\mathcal{G}$, it holds that for $\pi, \pi' \in \Delta(S)$ that*

$$\sum_{i \in V} \left[ u_i(\pi_i, \pi'_{-i}) + u_i(\pi'_i, \pi_{-i}) \right] = 0. \tag{18}$$

*Or equivalently, $\sum_{i \in V} \left[ \pi_i^\top A_i \pi' + (\pi'_i)^\top A_i \pi \right] = 0$. It follows that*

$$\sum_{i \in V} \left\langle \pi_i - \pi'_i, A_i(\pi - \pi') \right\rangle = \sum_{i \in V} \left[ u_i(\pi) + u_i(\pi') \right] - \sum_{i \in V} \left[ \pi_i^\top A_i \pi' + (\pi'_i)^\top A_i \pi \right] = 0.$$

### B.1  PROOF OF THEOREM 1

We start with the following lemma that characterizes the iterates of OMWU, which generalizes Cen et al. (2021, Lemma 1) for zero-sum two-player games to zero-sum polymatrix games. The proof can be found in Appendix E.2.

**Lemma 2.** *The iterates of OMWU based on the update rule (9) satisfy*

$$\left\langle \log \pi^{(t+1)} - (1 - \eta\tau) \log \pi^{(t)} - \eta\tau \log \pi_\tau^\star, \overline{\pi}^{(t+1)} - \pi_\tau^\star \right\rangle = 0.$$

To continue, by the definition of KL divergence, we have

$$\begin{aligned}
&\left\langle \log \pi^{(t+1)} - (1 - \eta\tau) \log \pi^{(t)} - \eta\tau \log \pi_\tau^\star, \overline{\pi}^{(t+1)} \right\rangle \\
&= \left\langle \log \overline{\pi}^{(t+1)} - (1 - \eta\tau) \log \pi^{(t)} - \eta\tau \log \pi_\tau^\star, \overline{\pi}^{(t+1)} \right\rangle \\
&\quad - \left\langle \log \overline{\pi}^{(t+1)} - \log \pi^{(t+1)}, \pi^{(t+1)} \right\rangle - \left\langle \log \overline{\pi}^{(t+1)} - \log \pi^{(t+1)}, \overline{\pi}^{(t+1)} - \pi^{(t+1)} \right\rangle \\
&= (1 - \eta\tau)\mathsf{KL}\left(\overline{\pi}^{(t+1)} \,\|\, \pi^{(t)}\right) + \eta\tau\mathsf{KL}\left(\overline{\pi}^{(t+1)} \,\|\, \pi_\tau^\star\right) + \mathsf{KL}\left(\pi^{(t+1)} \,\|\, \overline{\pi}^{(t+1)}\right) \\
&\quad - \left\langle \log \overline{\pi}^{(t+1)} - \log \pi^{(t+1)}, \overline{\pi}^{(t+1)} - \pi^{(t+1)} \right\rangle.
\end{aligned}$$

In addition,

$$-\left\langle \log \pi^{(t+1)} - (1 - \eta\tau) \log \pi^{(t)} - \eta\tau \log \pi_\tau^\star, \pi_\tau^\star \right\rangle = \mathsf{KL}\left(\pi_\tau^\star \,\|\, \pi^{(t+1)}\right) - (1 - \eta\tau)\mathsf{KL}\left(\pi_\tau^\star \,\|\, \pi^{(t)}\right).$$

Summing up the above two relations, in view of Lemma 2, it holds that

$$\begin{aligned}
\mathsf{KL}\left(\pi_\tau^\star \,\|\, \pi^{(t+1)}\right) = {}&(1 - \eta\tau)\mathsf{KL}\left(\pi_\tau^\star \,\|\, \pi^{(t)}\right) - (1 - \eta\tau)\mathsf{KL}\left(\overline{\pi}^{(t+1)} \,\|\, \pi^{(t)}\right) - \mathsf{KL}\left(\pi^{(t+1)} \,\|\, \overline{\pi}^{(t+1)}\right) \\
&+ \left\langle \log \overline{\pi}^{(t+1)} - \log \pi^{(t+1)}, \overline{\pi}^{(t+1)} - \pi^{(t+1)} \right\rangle - \eta\tau\mathsf{KL}\left(\overline{\pi}^{(t+1)} \,\|\, \pi_\tau^\star\right).
\end{aligned} \tag{19}$$

We now proceed to bound the terms of interest one by one.

**Bounding** $\mathsf{KL}\big(\pi_\tau^\star \,\|\, \pi^{(t)}\big)$**.** We aim to control the right-hand-side (RHS) of (19). Based on the update rule of $\overline{\pi}_i^{(t+1)}$ in Algorithm 1, we have

$$\log \overline{\pi}_i^{(t+1)} - \log \pi_i^{(t+1)} \overset{\mathbf{1}}{=} \eta A_i(\overline{\pi}^{(t)} - \overline{\pi}^{(t+1)}) \tag{20}$$
$$\overset{\mathbf{1}}{=} \eta A_i(\overline{\pi}^{(t)} - \pi^{(t)}) + \eta A_i(\pi^{(t)} - \overline{\pi}^{(t+1)}).$$

It follows that

$$\big\langle \log \overline{\pi}_i^{(t+1)} - \log \pi_i^{(t+1)}, \, \overline{\pi}_i^{(t+1)} - \pi_i^{(t+1)} \big\rangle$$
$$= \eta \sum_{j \in \mathcal{N}_i} (\overline{\pi}_i^{(t+1)} - \pi_i^{(t+1)})^\top A_{ij}(\overline{\pi}_j^{(t)} - \pi_j^{(t)}) + \eta \sum_{j \in \mathcal{N}_i} (\overline{\pi}_i^{(t+1)} - \pi_i^{(t+1)})^\top A_{ij}(\pi_j^{(t)} - \overline{\pi}_j^{(t+1)})$$
$$\leq \eta \sum_{j \in \mathcal{N}_i} \|A_{ij}\|_\infty \big\|\overline{\pi}_i^{(t+1)} - \pi_i^{(t+1)}\big\|_1 \big\|\pi_j^{(t)} - \overline{\pi}_j^{(t)}\big\|_1 + \eta \sum_{j \in \mathcal{N}_i} \|A_{ij}\|_\infty \big\|\overline{\pi}_i^{(t+1)} - \pi_i^{(t+1)}\big\|_1 \big\|\pi_j^{(t)} - \overline{\pi}_j^{(t+1)}\big\|_1$$
$$\leq \frac{\eta}{2} \|A\|_\infty \sum_{j \in \mathcal{N}_i} \left( \big\|\overline{\pi}_j^{(t)} - \pi_j^{(t)}\big\|_1^2 + \big\|\overline{\pi}_j^{(t+1)} - \pi_j^{(t)}\big\|_1^2 + 2\big\|\overline{\pi}_i^{(t+1)} - \pi_i^{(t+1)}\big\|_1^2 \right)$$
$$\leq \eta \|A\|_\infty \sum_{j \in \mathcal{N}_i} \left( \mathsf{KL}\big(\pi_j^{(t)} \,\|\, \overline{\pi}_j^{(t)}\big) + \mathsf{KL}\big(\overline{\pi}_j^{(t+1)} \,\|\, \pi_j^{(t)}\big) + 2\mathsf{KL}\big(\pi_i^{(t+1)} \,\|\, \overline{\pi}_i^{(t+1)}\big) \right), \tag{21}$$

where the last line follows from Pinsker's inequality. Summing the inequality over $i \in V$, we get

$$\big\langle \log \overline{\pi}^{(t+1)} - \log \pi^{(t+1)}, \, \overline{\pi}^{(t+1)} - \pi^{(t+1)} \big\rangle$$
$$\leq \eta d_{\mathsf{max}} \|A\|_\infty \left( \mathsf{KL}\big(\pi^{(t)} \,\|\, \overline{\pi}^{(t)}\big) + \mathsf{KL}\big(\overline{\pi}^{(t+1)} \,\|\, \pi^{(t)}\big) + 2\mathsf{KL}\big(\pi^{(t+1)} \,\|\, \overline{\pi}^{(t+1)}\big) \right).$$

Plugging the above inequality back into (19) yields

$$\mathsf{KL}\big(\pi_\tau^\star \,\|\, \pi^{(t+1)}\big) \leq (1 - \eta\tau)\mathsf{KL}\big(\pi_\tau^\star \,\|\, \pi^{(t)}\big) - (1 - \eta\tau - \eta d_{\mathsf{max}} \|A\|_\infty) \, \mathsf{KL}\big(\overline{\pi}^{(t+1)} \,\|\, \pi^{(t)}\big)$$
$$- (1 - 2\eta d_{\mathsf{max}} \|A\|_\infty)\mathsf{KL}\big(\pi^{(t+1)} \,\|\, \overline{\pi}^{(t+1)}\big) + \eta d_{\mathsf{max}} \|A\|_\infty \, \mathsf{KL}\big(\pi^{(t)} \,\|\, \overline{\pi}^{(t)}\big)$$
$$- \eta\tau\mathsf{KL}\big(\overline{\pi}^{(t+1)} \,\|\, \pi_\tau^\star\big). \tag{22}$$

With the choice of the learning rate

$$0 < \eta \leq \min \left\{ \frac{1}{2\tau}, \, \frac{1}{4d_{\mathsf{max}} \|A\|_\infty} \right\},$$

it holds that $1 - \eta\tau - \eta d_{\mathsf{max}} \|A\|_\infty > 0$ and

$$\eta d_{\mathsf{max}} \|A\|_\infty \leq \frac{1}{4} \leq (1 - \eta\tau)(1 - 2\eta d_{\mathsf{max}} \|A\|_\infty). \tag{23}$$

This allows us to further relax (22) by

$$\mathsf{KL}\big(\pi_\tau^\star \,\|\, \pi^{(t+1)}\big) + (1 - 2\eta d_{\mathsf{max}} \|A\|_\infty)\mathsf{KL}\big(\pi^{(t+1)} \,\|\, \overline{\pi}^{(t+1)}\big)$$
$$\leq (1 - \eta\tau)\mathsf{KL}\big(\pi_\tau^\star \,\|\, \pi^{(t)}\big) + \eta d_{\mathsf{max}} \|A\|_\infty \, \mathsf{KL}\big(\pi^{(t)} \,\|\, \overline{\pi}^{(t)}\big)$$
$$\leq (1 - \eta\tau) \left( \mathsf{KL}\big(\pi_\tau^\star \,\|\, \pi^{(t)}\big) + (1 - 2\eta d_{\mathsf{max}} \|A\|_\infty)\mathsf{KL}\big(\pi^{(t)} \,\|\, \overline{\pi}^{(t)}\big) \right).$$

Let us now introduce the potential function of iterates

$$L^{(t)} := \mathsf{KL}\big(\pi_\tau^\star \,\|\, \pi^{(t)}\big) + (1 - 2\eta d_{\mathsf{max}} \|A\|_\infty)\mathsf{KL}\big(\pi^{(t)} \,\|\, \overline{\pi}^{(t)}\big),$$

which allows us to simply the previous inequality as

$$L^{(t+1)} \leq (1 - \eta\tau)L^{(t)} \leq (1 - \eta\tau)^{t+1} L^{(0)} = (1 - \eta\tau)^{t+1}\mathsf{KL}\big(\pi_\tau^\star \,\|\, \pi^{(0)}\big), \tag{24}$$

where the last equality follows from the definition $\overline{\pi}^{(0)} = \pi^{(0)}$. Hence, we have

$$\mathsf{KL}\big(\pi_\tau^\star \,\|\, \pi^{(t)}\big) \leq L^{(t)} \leq (1 - \eta\tau)^t \mathsf{KL}\big(\pi_\tau^\star \,\|\, \pi^{(0)}\big).$$

**Bounding** $\mathsf{KL}\big(\pi_\tau^\star \,\|\, \overline{\pi}^{(t+1)}\big)$**.** Following similar approaches to (21), we can bound

$$
\begin{aligned}
&- \big\langle \pi_{i,\tau}^\star - \overline{\pi}_i^{(t+1)}, \, \log \overline{\pi}_i^{(t+1)} - \log \pi_i^{(t+1)} \big\rangle \\
&= \eta(\overline{\pi}_i^{(t+1)} - \pi_{i,\tau}^\star)^\top A_i(\overline{\pi}^{(t)} - \pi^{(t)}) + \eta(\overline{\pi}_i^{(t+1)} - \pi_{i,\tau}^\star)^\top A_i(\pi^{(t)} - \overline{\pi}^{(t+1)}) \\
&\leq \eta \,\|A\|_\infty \sum_{j \in \mathcal{N}_i} \Big( \mathsf{KL}\big(\pi_j^{(t)} \,\|\, \overline{\pi}_j^{(t)}\big) + \mathsf{KL}\big(\overline{\pi}_j^{(t+1)} \,\|\, \pi_j^{(t)}\big) + 2\mathsf{KL}\big(\pi_{i,\tau}^\star \,\|\, \overline{\pi}_i^{(t+1)}\big) \Big).
\end{aligned}
\tag{25}
$$

Summing the inequality over $i \in V$ leads to

$$
\begin{aligned}
&- \big\langle \pi_\tau^\star - \overline{\pi}^{(t+1)}, \, \log \overline{\pi}^{(t+1)} - \log \pi^{(t+1)} \big\rangle \\
&\leq \eta d_{\mathsf{max}} \|A\|_\infty \big[ \mathsf{KL}\big(\pi^{(t)} \,\|\, \overline{\pi}^{(t)}\big) + \mathsf{KL}\big(\overline{\pi}^{(t+1)} \,\|\, \pi^{(t)}\big) + 2\mathsf{KL}\big(\pi_\tau^\star \,\|\, \overline{\pi}^{(t+1)}\big) \big].
\end{aligned}
$$

On the other hand, by the definition of KL divergence, we have

$$
\mathsf{KL}\big(\pi_\tau^\star \,\|\, \overline{\pi}^{(t+1)}\big) = \mathsf{KL}\big(\pi_\tau^\star \,\|\, \pi^{(t+1)}\big) - \mathsf{KL}\big(\overline{\pi}^{(t+1)} \,\|\, \pi^{(t+1)}\big) - \big\langle \pi_\tau^\star - \overline{\pi}^{(t+1)}, \, \log \overline{\pi}^{(t+1)} - \log \pi^{(t+1)} \big\rangle.
\tag{26}
$$

Combining the above two inequalities, we get

$$
\begin{aligned}
&(1 - 2\eta d_{\mathsf{max}} \|A\|_\infty)\mathsf{KL}\big(\pi_\tau^\star \,\|\, \overline{\pi}^{(t+1)}\big) \\
&\leq \mathsf{KL}\big(\pi_\tau^\star \,\|\, \pi^{(t+1)}\big) + \eta d_{\mathsf{max}} \|A\|_\infty \Big( \mathsf{KL}\big(\pi^{(t)} \,\|\, \overline{\pi}^{(t)}\big) + \mathsf{KL}\big(\overline{\pi}^{(t+1)} \,\|\, \pi^{(t)}\big) \Big).
\end{aligned}
$$

Plugging the above inequality back into (22), we have

$$
\begin{aligned}
&(1 - 2\eta d_{\mathsf{max}} \|A\|_\infty)\mathsf{KL}\big(\pi_\tau^\star \,\|\, \overline{\pi}^{(t+1)}\big) \\
&\leq (1 - \eta\tau)\mathsf{KL}\big(\pi_\tau^\star \,\|\, \pi^{(t)}\big) - (1 - \eta\tau - 2d_{\mathsf{max}}\eta \|A\|_\infty)\mathsf{KL}\big(\overline{\pi}^{(t+1)} \,\|\, \pi^{(t)}\big) - \eta\tau\mathsf{KL}\big(\overline{\pi}^{(t+1)} \,\|\, \pi_\tau^\star\big) \\
&\quad - (1 - 2\eta d_{\mathsf{max}} \|A\|_\infty)\mathsf{KL}\big(\pi^{(t+1)} \,\|\, \overline{\pi}^{(t+1)}\big) + 2\eta d_{\mathsf{max}} \|A\|_\infty \, \mathsf{KL}\big(\pi^{(t)} \,\|\, \overline{\pi}^{(t)}\big) \\
&\leq (1 - \eta\tau)\mathsf{KL}\big(\pi_\tau^\star \,\|\, \pi^{(t)}\big) + 2\eta d_{\mathsf{max}} \|A\|_\infty \, \mathsf{KL}\big(\pi^{(t)} \,\|\, \overline{\pi}^{(t)}\big) \\
&\leq \mathsf{KL}\big(\pi_\tau^\star \,\|\, \pi^{(t)}\big) + (1 - 2\eta d_{\mathsf{max}} \|A\|_\infty)\mathsf{KL}\big(\pi^{(t)} \,\|\, \overline{\pi}^{(t)}\big) = L^{(t)},
\end{aligned}
$$

where the second and third inequalities follow from the choice of the learning rate, and the last line follows from the definition of the potential function $L^{(t)}$. Then the result follows from (24) as

$$
\frac{1}{2}\mathsf{KL}\big(\pi_\tau^\star \,\|\, \overline{\pi}^{(t+1)}\big) \leq (1 - 2\eta d_{\mathsf{max}} \|A\|_\infty)\mathsf{KL}\big(\pi_\tau^\star \,\|\, \overline{\pi}^{(t+1)}\big) \leq L^{(t)} \leq (1 - \eta\tau)^t \mathsf{KL}\big(\pi_\tau^\star \,\|\, \pi^{(0)}\big).
$$

**Bounding the QRE-Gap.** Finally, we bound the QRE-gap, which can be linked to the KL divergence using the following lemma. The proof can be found in Appendix E.3.

**Lemma 3.** *For any $\pi \in \Delta(S)$ and QRE $\pi_\tau^\star \in \Delta(S)$, it holds that*

$$
QRE\text{-}Gap_\tau(\pi) \leq \tau\mathsf{KL}\big(\pi \,\|\, \pi_\tau^\star\big) + \frac{d_{\mathsf{max}}^2 \|A\|_\infty^2}{\tau}\mathsf{KL}\big(\pi_\tau^\star \,\|\, \pi\big).
$$

Lemma 3 tells us

$$
\texttt{QRE-Gap}_\tau\big(\overline{\pi}^{(t)}\big) \leq \tau\mathsf{KL}\big(\overline{\pi}^{(t)} \,\|\, \pi_\tau^\star\big) + \frac{d_{\mathsf{max}}^2 \|A\|_\infty^2}{\tau}\mathsf{KL}\big(\pi_\tau^\star \,\|\, \overline{\pi}^{(t)}\big).
\tag{27}
$$

With $\mathsf{KL}\big(\pi_\tau^\star \,\|\, \overline{\pi}^{(t)}\big)$ controlled in the above, we still need to control $\mathsf{KL}\big(\overline{\pi}^{(t)} \,\|\, \pi_\tau^\star\big)$. From (22), it follows that

$$
\tau\mathsf{KL}\big(\overline{\pi}^{(t)} \,\|\, \pi_\tau^\star\big) \leq \eta^{-1}(1 - \eta\tau)L^{(t-1)} \leq \eta^{-1}(1 - \eta\tau)^t L^{(0)} = \eta^{-1}(1 - \eta\tau)^t \mathsf{KL}\big(\pi_\tau^\star \,\|\, \pi^{(0)}\big).
$$

Plugging them back to (27), we arrive at

$$
\texttt{QRE-Gap}_\tau\big(\overline{\pi}^{(t)}\big) \leq \big(\eta^{-1} + 2\tau^{-1}d_{\mathsf{max}}^2\|A\|_\infty^2\big)(1 - \eta\tau)^{t-1}\mathsf{KL}\big(\pi_\tau^\star \,\|\, \pi^{(0)}\big).
$$

### B.2 PROOF OF THEOREM 2

We begin with bounding the KL divergence $\mathsf{KL}\big(\pi_\tau^\star \,\|\, \pi^{(t)}\big)$ and then move to bound the QRE-gap by linking it to the KL divergence.

**Bounding the term** $\mathsf{KL}\big(\pi_\tau^\star \,\|\, \pi^{(t)}\big)$. We start with the following equation

$$
\begin{aligned}
(1 - \eta\tau)\mathsf{KL}\big(\pi_{i,\tau}^\star \,\|\, \pi_i^{(t)}\big) = {} & (1 - \eta\tau)\mathsf{KL}\big(\overline{\pi}_i^{(t+1)} \,\|\, \pi_i^{(t)}\big) + \eta\tau\mathsf{KL}\big(\overline{\pi}_i^{(t+1)} \,\|\, \pi_{i,\tau}^\star\big) + \mathsf{KL}\big(\pi_i^{(t+1)} \,\|\, \overline{\pi}_i^{(t+1)}\big) \\
& + \mathsf{KL}\big(\pi_{i,\tau}^\star \,\|\, \pi_i^{(t+1)}\big) - \big\langle \log \overline{\pi}_i^{(t+1)} - \log \pi_i^{(t+1)}, \, \overline{\pi}_i^{(t+1)} - \pi_i^{(t+1)} \big\rangle \\
& + \eta(\overline{\pi}_i^{(t+1)} - \pi_{i,\tau}^\star)^\top A_i(\overline{\pi}^{(\kappa_i^{(t)})} - \pi_\tau^\star)
\end{aligned}
\tag{28}
$$

where its proof follows a similar deduction as (19). Our first target is to bound the last two terms on the RHS of (28) with

$$
\eta\tau\mathsf{KL}\big(\overline{\pi}_i^{(t+1)} \,\|\, \pi_{i,\tau}^\star\big) + \mathsf{KL}\big(\pi_i^{(t+1)} \,\|\, \overline{\pi}_i^{(t+1)}\big) + (1 - \eta\tau)\mathsf{KL}\big(\overline{\pi}_i^{(t+1)} \,\|\, \pi_i^{(t)}\big).
$$

Let us introduce the potential function of iterates

$$
\Psi_i^{(l)} := \mathsf{KL}\big(\overline{\pi}_i^{(l+1)} \,\|\, \pi_i^{(l)}\big) + \mathsf{KL}\big(\pi_i^{(l)} \,\|\, \overline{\pi}_i^{(l)}\big), \quad \Psi^{(l)} = \sum_{i \in V} \Psi_i^{(l)} = \mathsf{KL}\big(\overline{\pi}^{(l+1)} \,\|\, \pi^{(l)}\big) + \mathsf{KL}\big(\pi^{(l)} \,\|\, \overline{\pi}^{(l)}\big),
$$

which will be used repetitively in the rest of this proof. For notational simplicity, let $\Psi_i^{(l)} = 0$ when $l < 0$.

**Step 1: bounding** $\big\langle \log \overline{\pi}_i^{(t+1)} - \log \pi_i^{(t+1)}, \, \overline{\pi}_i^{(t+1)} - \pi_i^{(t+1)} \big\rangle$. Following a similar argument as (21), we get

$$
\begin{aligned}
& \big\langle \log \overline{\pi}_i^{(t+1)} - \log \pi_i^{(t+1)}, \, \overline{\pi}_i^{(t+1)} - \pi_i^{(t+1)} \big\rangle \\
& = \eta \sum_{j \in \mathcal{N}_i} (\overline{\pi}_i^{(t+1)} - \pi_i^{(t+1)})^\top A_{ij}(\overline{\pi}_j^{(\kappa_i^{(t+1)})} - \overline{\pi}_j^{(\kappa_i^{(t)})}) \\
& \le \eta d_{\mathsf{max}} \|A\|_\infty \, \mathsf{KL}\big(\pi_i^{(t+1)} \,\|\, \overline{\pi}_i^{(t+1)}\big) + \frac{\eta \|A\|_\infty}{2} \sum_{j \in \mathcal{N}_i} \big\| \overline{\pi}_j^{(\kappa_i^{(t+1)})} - \overline{\pi}_j^{(\kappa_i^{(t)})} \big\|_1^2.
\end{aligned}
\tag{29}
$$

To control the term $\big\| \overline{\pi}_j^{(\kappa_i^{(t+1)})} - \overline{\pi}_j^{(\kappa_i^{(t)})} \big\|_1^2$, when $t = 0$, we have

$$
\big\| \overline{\pi}_j^{(\kappa_i^{(t+1)})} - \overline{\pi}_j^{(\kappa_i^{(t)})} \big\|_1^2 = \big\| \overline{\pi}_j^{(\kappa_i^{(t+1)})} - \overline{\pi}_j^{(0)} \big\|_1^2 \le \big\| \overline{\pi}_j^{(1)} - \pi_j^{(0)} \big\|_1^2 \le 2\Psi_j^{(0)}
\tag{30}
$$

by Pinsker's inequality. For $t \ge 1$, consider the decomposition

$$
\overline{\pi}_j^{(t)} - \overline{\pi}_j^{(t-k)} = \sum_{l=t-k}^{t-1} \left( \overline{\pi}_j^{(l+1)} - \overline{\pi}_j^{(l)} \right), \quad \forall 1 \le k \le t,
$$

it then follows that

$$
\begin{aligned}
\big\| \overline{\pi}_j^{(t)} - \overline{\pi}_j^{(t-k)} \big\|_1^2 & \le k \sum_{l=t-k}^{t-1} \big\| \overline{\pi}_j^{(l+1)} - \overline{\pi}_j^{(l)} \big\|_1^2 \\
& \le 2k \sum_{l=t-k}^{t-1} \left( \big\| \overline{\pi}_j^{(l+1)} - \pi_j^{(l)} \big\|_1^2 + \big\| \pi_j^{(l)} - \overline{\pi}_j^{(l)} \big\|_1^2 \right) \\
& \le 4k \sum_{l=t-k}^{t-1} \Psi_j^{(l)},
\end{aligned}
\tag{31}
$$

where the last line applies Pinsker's inequality. Depending on whether $\gamma_i^{(t+1)} > 0$, we proceed to bound the terms $\big\| \overline{\pi}_j^{(\kappa_i^{(t+1)})} - \overline{\pi}_j^{(\kappa_i^{(t)})} \big\|_1^2$ in (29) considering the following two cases based on (31).

- $\gamma_i^{(t+1)} = 0$. Then

$$
\big\| \overline{\pi}_j^{(\kappa_i^{(t+1)})} - \overline{\pi}_j^{(\kappa_i^{(t)})} \big\|_1^2 \le 2\big\| \overline{\pi}_j^{(t+1)} - \overline{\pi}_j^{(t)} \big\|_1^2 + 2\big\| \overline{\pi}_j^{(t)} - \overline{\pi}_j^{(\kappa_i^{(t)})} \big\|_1^2
$$

$$\leq 8\Psi_j^{(t)} + 8\gamma_i^{(t)} \sum_{l=t-\gamma_i^{(t)}}^{t-1} \Psi_j^{(l)},$$

where the last step uses (31) and

$$\left\|\overline{\pi}_j^{(t+1)} - \overline{\pi}_j^{(t)}\right\|_1^2 \leq 2\left(\left\|\overline{\pi}_j^{(t+1)} - \pi_j^{(t)}\right\|_1^2 + \left\|\pi_j^{(t)} - \overline{\pi}_j^{(t)}\right\|_1^2\right) \leq 4\Psi_j^{(t)}$$

via again Pinsker's inequality.

- $\gamma_i^{(t+1)} > 0$. Then it follows similarly that

$$\left\|\overline{\pi}_j^{(\kappa_i^{(t+1)})} - \overline{\pi}_j^{(\kappa_i^{(t)})}\right\|_1^2 \leq \sum_{l=t+1-\gamma_i^{(t+1)}}^{t-1} \left\|\overline{\pi}_j^{(l+1)} - \overline{\pi}_j^{(l)}\right\|_1^2 + \sum_{l=t-\gamma_i^{(t)}}^{t-1} \left\|\overline{\pi}_j^{(l+1)} - \overline{\pi}_j^{(l)}\right\|_1^2$$

$$\leq 4\gamma_i^{(t+1)} \sum_{l=t-\gamma_i^{(t+1)}}^{t-1} \Psi_j^{(l)} + 4\gamma_i^{(t)} \sum_{l=t-\gamma_i^{(t)}}^{t-1} \Psi_j^{(l)}.$$

Combining the above two bounds together, we get

$$\left\|\overline{\pi}_j^{(\kappa_i^{(t+1)})} - \overline{\pi}_j^{(\kappa_i^{(t)})}\right\|_1^2 \leq 8\Psi_j^{(t)} + 8\gamma_i^{(t)} \sum_{l=t-\gamma_i^{(t)}}^{t-1} \Psi_j^{(l)} + 4\gamma_i^{(t+1)} \sum_{l=t-\gamma_i^{(t+1)}}^{t-1} \Psi_j^{(l)} \qquad (32)$$

when $t > 0$. In view of (30) when $t = 0$, the above bound (32) holds for all $t \geq 0$. Plugging the above inequality into (29) yields

$$\left\langle \log \overline{\pi}_i^{(t+1)} - \log \pi_i^{(t+1)}, \overline{\pi}_i^{(t+1)} - \pi_i^{(t+1)} \right\rangle$$

$$\leq 2\eta \|A\|_\infty \sum_{j\in\mathcal{N}_i} \sum_{l=t-\gamma_i^{(t+1)}}^{t-1} \gamma_i^{(t+1)} \Psi_j^{(l)} + 4\eta \|A\|_\infty \sum_{j\in\mathcal{N}_i} \sum_{l=t-\gamma_i^{(t)}}^{t-1} \gamma_i^{(t)} \Psi_j^{(l)}$$

$$+ 4\eta \|A\|_\infty \sum_{j\in\mathcal{N}_i} \Psi_j^{(t)} + \eta d_{\mathsf{max}} \|A\|_\infty \, \mathsf{KL}\big(\pi_i^{(t+1)} \,\|\, \overline{\pi}_i^{(t+1)}\big). \qquad (33)$$

**Step 2: bounding** $(\overline{\pi}_i^{(t+1)} - \pi_{i,\tau}^\star)^\top A_i(\overline{\pi}^{(\kappa_i^{(t+1)})} - \pi_\tau^\star)$. Let us begin with the following decomposition

$$(\overline{\pi}_i^{(t+1)} - \pi_{i,\tau}^\star)^\top A_i(\overline{\pi}^{(\kappa_i^{(t+1)})} - \pi_\tau^\star) = (\overline{\pi}_i^{(t+1)} - \pi_{i,\tau}^\star)^\top A_i(\overline{\pi}^{(t+1)} - \pi_\tau^\star)$$

$$+ (\overline{\pi}_i^{(t+1)} - \pi_{i,\tau}^\star)^\top A_i(\overline{\pi}^{(\kappa_i^{(t+1)})} - \overline{\pi}^{(t+1)}), \qquad (34)$$

where the second term in the RHS of (34) can be bounded by

$$(\overline{\pi}_i^{(t+1)} - \pi_{i,\tau}^\star)^\top A_i(\overline{\pi}^{(\kappa_i^{(t+1)})} - \overline{\pi}^{(t+1)})$$

$$= \sum_{j\in\mathcal{N}_i} (\overline{\pi}_i^{(t+1)} - \pi_{i,\tau}^\star)^\top A_{ij}(\overline{\pi}_j^{(\kappa_i^{(t+1)})} - \overline{\pi}_j^{(t+1)})$$

$$\leq \|A\|_\infty \sum_{j\in\mathcal{N}_i} \left\|\overline{\pi}_i^{(t+1)} - \pi_{i,\tau}^\star\right\|_1 \left\|\overline{\pi}_j^{(\kappa_i^{(t+1)})} - \overline{\pi}_j^{(t+1)}\right\|_1$$

$$\leq \frac{1}{2}\|A\|_\infty \sum_{j\in\mathcal{N}_i} \left(\frac{\tau}{d_{\mathsf{max}}\|A\|_\infty}\left\|\overline{\pi}_i^{(t+1)} - \pi_{i,\tau}^\star\right\|_1^2 + \frac{d_{\mathsf{max}}\|A\|_\infty}{\tau}\left\|\overline{\pi}_j^{(\kappa_i^{(t+1)})} - \overline{\pi}_j^{(t+1)}\right\|_1^2\right)$$

$$\leq \tau\mathsf{KL}\big(\overline{\pi}_i^{(t+1)} \,\|\, \pi_{i,\tau}^\star\big) + \frac{d_{\mathsf{max}}\|A\|_\infty^2}{2\tau} \sum_{j\in\mathcal{N}_i} \left\|\overline{\pi}_j^{(\kappa_i^{(t+1)})} - \overline{\pi}_j^{(t+1)}\right\|_1^2.$$

Following similar deduction of (32) for the second term, we attain

$$(\overline{\pi}_i^{(t+1)} - \pi_{i,\tau}^\star)^\top A_i(\overline{\pi}^{(\kappa_i^{(t+1)})} - \overline{\pi}^{(t+1)})$$

$$\leq \tau \mathsf{KL}\big(\overline{\pi}_i^{(t+1)} \,\|\, \pi_{i,\tau}^\star\big) + \frac{4d_{\max}\|A\|_\infty^2}{\tau} \sum_{j \in \mathcal{N}_i} \Big(\Psi_j^{(t)} + \sum_{l=t-\gamma_i^{(t+1)}}^{t-1} \gamma_i^{(t+1)}\Psi_j^{(l)}\Big).$$

Plugging the above inequality back to (34) results in

$$\begin{aligned}
(\overline{\pi}_i^{(t+1)} - \pi_{i,\tau}^\star)^\top A_i(\overline{\pi}^{(\kappa_i^{(t+1)})} - \pi_\tau^\star) &\leq (\overline{\pi}_i^{(t+1)} - \pi_{i,\tau}^\star)^\top A_i(\overline{\pi}^{(t+1)} - \pi_\tau^\star) \\
&\quad + \tau \mathsf{KL}\big(\overline{\pi}_i^{(t+1)} \,\|\, \pi_{i,\tau}^\star\big) + \frac{4d_{\max}\|A\|_\infty^2}{\tau} \sum_{j \in \mathcal{N}_i} \Big(\Psi_j^{(t)} + \sum_{l=t-\gamma_i^{(t+1)}}^{t-1} \gamma_i^{(t+1)}\Psi_j^{(l)}\Big).
\end{aligned}$$

$$(35)$$

**Step 3: combining the bounds.** For simplicity, we introduce the short-hand notation

$$c_\tau = 1 + \frac{d_{\max}\|A\|_\infty}{\tau} \qquad \text{and} \qquad c_A = d_{\max}\|A\|_\infty. \tag{36}$$

Combining (33) and (35) into (28), and summing over $i \in V$ gives

$$\begin{aligned}
(1 &- \eta\tau)\mathsf{KL}\big(\pi_\tau^\star \,\|\, \pi^{(t)}\big) \\
&\geq (1 - \eta\tau)\mathsf{KL}\big(\overline{\pi}^{(t+1)} \,\|\, \pi^{(t)}\big) + (1 - 2\eta c_A)\mathsf{KL}\big(\pi^{(t+1)} \,\|\, \overline{\pi}^{(t+1)}\big) + \mathsf{KL}\big(\pi_\tau^\star \,\|\, \pi^{(t+1)}\big) \\
&\quad - 4\eta\|A\|_\infty \sum_{i \in V} \sum_{j \in \mathcal{N}_i} \Big( \sum_{l=t-\gamma_i^{(t+1)}}^{t-1} c_\tau \gamma_i^{(t+1)}\Psi_j^{(l)} + \sum_{l=t-\gamma_i^{(t)}}^{t-1} \gamma_i^{(t)}\Psi_j^{(l)} + c_\tau \Psi_j^{(t)} \Big) \\
&\geq \mathsf{KL}\big(\pi_\tau^\star \,\|\, \pi^{(t+1)}\big) + (1 - 4\eta c_A(c_\tau + 1))\,\Psi^{(t)} \\
&\quad - 4\eta\|A\|_\infty \sum_{i \in V} \sum_{j \in \mathcal{N}_i} \Big( c_\tau \sum_{l=t-\gamma_i^{(t+1)}}^{t-1} \gamma_i^{(t+1)}\Psi_j^{(l)} + \sum_{l=t-\gamma_i^{(t)}}^{t-1} \gamma_i^{(t)}\Psi_j^{(l)} \Big),
\end{aligned}$$

$$(37)$$

where we make use of the fact

$$\sum_{i \in V}(\overline{\pi}_i^{(t+1)} - \pi_{i,\tau}^\star)^\top A_i(\overline{\pi}^{(t+1)} - \pi_\tau^\star) = 0$$

from Lemma 1 in the first inequality, and the second inequality uses the relation

$$\sum_{i \in V} \sum_{j \in \mathcal{N}_i} \Psi_j^{(t)} = \sum_{i \in V} d_i \Psi_i^{(t)} \leq d_{\max}\Psi^{(t)}.$$

**Step 4: finishing up via averaging the delay.** We now evaluate the expectation of $\mathsf{KL}\big(\pi_\tau^\star \,\|\, \pi^{(t+1)}\big)$. Recall that we use subscript $\mathbb{E}_{\gamma^{(t)}}[\cdot]$ to represent the conditional expectation given $\gamma^{(t)} = \{\gamma_i^{(t)}\}_{i \in V}$. We shall first control the conditional expectation of the last term in (37). Observing that $\overline{\pi}_j^{(l+1)}, \pi_j^{(l)}$ are independent of $\gamma_i^{(t)}$ for $j \in \mathcal{N}_i$ and $l \leq t - 1$. Using the definition of $E(t - l)$, we have

$$\begin{aligned}
\sum_{i \in V} \sum_{j \in \mathcal{N}_i} \mathbb{E}_{\gamma^{(t)}}\Big[\gamma_i^{(t)} \sum_{l=t-\gamma_i^{(t)}}^{t-1} \Psi_j^{(l)}\Big] &= \sum_{i \in V} \sum_{l=0}^{t-1} \sum_{j \in \mathcal{N}_i} \mathbb{E}_{t-l \leq \gamma_i^{(t)}}\Big[\gamma_i^{(t)}\Psi_j^{(l)}\Big] \\
&\leq \sum_{l=0}^{t-1} E(t - l)\sum_{i \in V} \sum_{j \in \mathcal{N}_i} \Psi_j^{(l)} \\
&= \sum_{l=0}^{t-1} E(t - l) \sum_{i \in V} \sum_{j \in \mathcal{N}_i} \Psi_i^{(l)} \\
&\leq d_{\max} \sum_{l=0}^{t-1} E(t - l) \sum_{i \in V} \Psi_i^{(l)} = d_{\max} \sum_{l=0}^{t-1} E(t - l)\Psi^{(l)}, \quad (38)
\end{aligned}$$

where the second line follows from the definition of $E(t-l)$ in Assumption 1. Applying a similar argument to bound $\sum_{i\in V}\sum_{j\in\mathcal{N}_i}\mathbb{E}_{\gamma^{(t+1)}}\left[\gamma_i^{(t+1)}\sum_{l=t-\gamma_i^{(t+1)}}^{t-1}\Psi_j^{(l)}\right]$, and taking expectation of $\gamma^{(t)},\gamma^{(t+1)}$ on both sides of (37), we get

$$(1-\eta\tau)\mathbb{E}_{\gamma^{(t)}}\left[\mathsf{KL}\big(\pi_\tau^\star\,\|\,\pi^{(t)}\big)\right]\geq\mathbb{E}_{\gamma^{(t)},\gamma^{(t+1)}}\left[\mathsf{KL}\big(\pi_\tau^\star\,\|\,\pi^{(t+1)}\big)+(1-4c_A(c_\tau+1))\Psi^{(t)}\right]$$
$$-4\eta c_A(c_\tau+1)\sum_{l=0}^{t-1}E(t-l)\Psi^{(l)}.$$

Taking expectation on both sides over all the delays yields

$$(1-\eta\tau)\mathbb{E}\left[\mathsf{KL}\big(\pi_\tau^\star\,\|\,\pi^{(t)}\big)\right]$$
$$\geq\mathbb{E}\left[\mathsf{KL}\big(\pi_\tau^\star\,\|\,\pi^{(t+1)}\big)\right]+\mathbb{E}\Big[\underbrace{(1-4\eta c_A(c_\tau+1))\,\Psi^{(t)}-4\eta c_A(c_\tau+1)\sum_{l=0}^{t-1}E(t-l)\Psi^{(l)}}_{=:U^{(t)}}\Big]. \tag{39}$$

Telescoping over $t=0,1,\ldots,T$, we get

$$(1-\eta\tau)^{T+1}\mathsf{KL}\big(\pi_\tau^\star\,\|\,\pi^{(0)}\big)\geq\mathbb{E}\left[\mathsf{KL}\big(\pi_\tau^\star\,\|\,\pi^{(T+1)}\big)\right]+\sum_{t=0}^{T}(1-\eta\tau)^{T-t}\mathbb{E}\left[U^{(t)}\right], \tag{40}$$

which leads to the desired bound if

$$\sum_{t=0}^{t}(1-\eta\tau)^{T-t}\mathbb{E}\left[U^{(t)}\right]\geq 0. \tag{41}$$

**Proof of (41).** To begin, notice that with the choice of the learning rate

$$0<\eta\leq\min\left\{\frac{\tau}{24d_{\mathsf{max}}^2\,\|A\|_\infty^2\,(L+1)},\ \frac{\zeta-1}{\zeta\tau}\right\},$$

it follows that

$$\frac{1}{1-\eta\tau}\leq\zeta \tag{42a}$$

and

$$4\eta c_A(c_\tau+1)(L+1)<4\frac{\tau}{24d_{\mathsf{max}}^2\,\|A\|_\infty^2\,(L+1)}d_{\mathsf{max}}\,\|A\|_\infty\left(2+\frac{d_{\mathsf{max}}\,\|A\|_\infty}{\tau}\right)(L+1)$$
$$=\frac{\tau}{6d_{\mathsf{max}}\,\|A\|_\infty}\left(2+\frac{d_{\mathsf{max}}\,\|A\|_\infty}{\tau}\right)=\frac{\tau}{3d_{\mathsf{max}}\,\|A\|_\infty}+\frac{1}{6}\leq\frac{1}{2} \tag{42b}$$

as $\tau\leq d_{\mathsf{max}}\,\|A\|_\infty$. Both of these relations will be useful in our follow-up analysis.

Now, taking the definition of $U^{(t)}$ (cf. (39)), we have

$$\sum_{t=0}^{T}(1-\eta\tau)^{T-t}U^{(t)}=\sum_{t=0}^{T}(1-\eta\tau)^{T-t}\left[(1-4\eta c_A(c_\tau+1))\,\Psi^{(t)}-4\eta c_A(c_\tau+1)\sum_{l=0}^{t-1}E(t-l)\Psi^{(l)}\right],$$

where the second half of the RHS can be further controlled via

$$\sum_{t=0}^{T}(1-\eta\tau)^{T-t}\sum_{l=0}^{t-1}E(t-l)\Psi^{(l)}=\sum_{t=0}^{T}\Psi^{(t)}\sum_{l=t+1}^{T}(1-\eta\tau)^{T-l}E(l-t)$$
$$\leq\sum_{t=0}^{T}\Psi^{(t)}\sum_{l'=0}^{T-t}(1-\eta\tau)^{T-(t+l')}E(l')$$

$$= \sum_{t=0}^{T}(1-\eta\tau)^{T-t}\Psi^{(t)}\sum_{l'=0}^{T-t}(1-\eta\tau)^{-l'}E(l')$$

$$\leq \sum_{t=0}^{T}(1-\eta\tau)^{T-t}\Psi^{(t)}\sum_{l=0}^{\infty}\zeta^{l}E(l)$$

$$= \sum_{t=0}^{T}(1-\eta\tau)^{T-t}L\Psi^{(t)},$$

where the first line follows by changing the order of summation, the second line follows from the change of variable $l' = l - t$, and the last line follows from (42a) and the definition of $L$ in Assumption 1. Plugging the above relation back leads to

$$\sum_{t=0}^{T}(1-\eta\tau)^{T-t}U^{(t)} \geq \sum_{t=0}^{T}(1-\eta\tau)^{T-t}\left[(1-4\eta c_A(c_\tau+1)) - 4\eta c_A(c_\tau+1)L\right]\Psi^{(t)}$$

$$\geq \sum_{t=0}^{T}\frac{1}{2}(1-\eta\tau)^{T-t}\Psi^{(t)} \geq 0, \tag{43}$$

where the second line results from (42b).

**Bounding the term** $\mathsf{KL}\!\left(\pi_\tau^\star \,\|\, \overline{\pi}^{(t+1)}\right)$**.** With a similar deduction of (19), we get

$$(1-\eta\tau)\mathsf{KL}\!\left(\pi_\tau^\star \,\|\, \pi^{(t)}\right) + \eta\sum_{i\in V}(\overline{\pi}_i^{(t+1)} - \pi_\tau^\star)^\top A_i(\overline{\pi}^{(\kappa_i^{(t)})} - \pi_\tau^\star)$$

$$= \mathsf{KL}\!\left(\pi_\tau^\star \,\|\, \overline{\pi}^{(t+1)}\right) + (1-\eta\tau)\mathsf{KL}\!\left(\overline{\pi}^{(t+1)} \,\|\, \pi^{(t)}\right) + \eta\tau\mathsf{KL}\!\left(\overline{\pi}^{(t+1)} \,\|\, \pi_\tau^\star\right). \tag{44}$$

Following the similar argument of (35), we have

$$(\overline{\pi}_i^{(t+1)} - \pi_{i,\tau}^\star)^\top A_i(\overline{\pi}^{(\kappa_i^{(t)})} - \pi_\tau^\star) \leq (\overline{\pi}_i^{(t+1)} - \pi_{i,\tau}^\star)^\top A_i(\overline{\pi}^{(t+1)} - \pi_\tau^\star)$$

$$+ \frac{\tau}{2}\mathsf{KL}\!\left(\overline{\pi}_i^{(t+1)} \,\|\, \pi_{i,\tau}^\star\right) + \frac{8d_{\mathsf{max}}\|A\|_\infty^2}{\tau}\sum_{j\in\mathcal{N}_i}\left(\Psi_j^{(t)} + \sum_{l=t-\gamma_i^{(t)}}^{t-1}\gamma_i^{(t)}\Psi_j^{(l)}\right).$$

Summing over $i \in V$ and plugging into (44) yields

$$(1-\eta\tau)\mathsf{KL}\!\left(\pi_\tau^\star \,\|\, \pi^{(t)}\right) + \frac{8\eta d_{\mathsf{max}}\|A\|_\infty^2}{\tau}\sum_{(i,j)\in E}\left(\Psi_j^{(t)} + \sum_{l=t-\gamma_i^{(t)}}^{t-1}\gamma_i^{(t)}\Psi_j^{(l)}\right)$$

$$\geq \mathsf{KL}\!\left(\pi_\tau^\star \,\|\, \overline{\pi}^{(t+1)}\right) + (1-\eta\tau)\mathsf{KL}\!\left(\overline{\pi}^{(t+1)} \,\|\, \pi^{(t)}\right) + \frac{\eta\tau}{2}\mathsf{KL}\!\left(\overline{\pi}^{(t+1)} \,\|\, \pi_\tau^\star\right)$$

$$\geq \mathsf{KL}\!\left(\pi_\tau^\star \,\|\, \overline{\pi}^{(t+1)}\right) + \frac{\eta\tau}{2}\mathsf{KL}\!\left(\overline{\pi}^{(t+1)} \,\|\, \pi_\tau^\star\right).$$

Taking expectation on both sides over all delays and using (38) leads to

$$(1-\eta\tau)\mathbb{E}\left[\mathsf{KL}\!\left(\pi_\tau^\star \,\|\, \pi^{(t)}\right)\right] + \frac{8\eta d_{\mathsf{max}}^2\|A\|_\infty^2}{\tau}\mathbb{E}\left[\Psi^{(t)} + \sum_{l=0}^{t-1}E(t-l)\Psi^{(l)}\right]$$

$$\geq \mathbb{E}\left[\mathsf{KL}\!\left(\pi_\tau^\star \,\|\, \overline{\pi}^{(t+1)}\right)\right] + \frac{\eta\tau}{2}\mathbb{E}\left[\mathsf{KL}\!\left(\overline{\pi}^{(t+1)} \,\|\, \pi_\tau^\star\right)\right]. \tag{45}$$

Notice that with the choice of the learning rate

$$0 < \eta \leq \min\left\{\frac{\tau}{24d_{\mathsf{max}}^2\|A\|_\infty^2(L+1)}, \frac{\zeta-1}{\zeta\tau}\right\},$$

we have

$$\frac{8(L+1)\eta d_{\mathsf{max}}^2\|A\|_\infty^2}{\tau} \leq \frac{1}{2}$$

and

$$(1 - \eta\tau)^{t+1} \mathsf{KL}\big(\pi_\tau^\star \,\|\, \pi^{(0)}\big) \geq \frac{1}{2} \sum_{l=0}^{t} (1 - \eta\tau)^{t-l} \mathbb{E}\left[\Psi^{(l)}\right]$$

by combining (43) and (40). It follows that

$$\mathbb{E}\left[\Psi^{(t)}\right] \leq 2(1 - \eta\tau)^{t+1} \mathsf{KL}\big(\pi_\tau^\star \,\|\, \pi^{(0)}\big)$$

and

$$\mathbb{E}\left[\sum_{l=0}^{t-1} E(t-l)\Psi^{(l)}\right] \overset{\text{(i)}}{\leq} \mathbb{E}\left[\sum_{l=0}^{t-1}(1-\eta\tau)^{t-l}\Psi^{(l)} \cdot E(t-l)\zeta^{t-l}\right]$$

$$\leq \mathbb{E}\left[\sum_{l=0}^{t-1}(1-\eta\tau)^{t-l}\Psi^{(l)} \sum_{l=0}^{t-1} E(t-l)\zeta^{t-l}\right]$$

$$\overset{\text{(ii)}}{\leq} 2L(1-\eta\tau)^{t+1}\mathsf{KL}\big(\pi_\tau^\star \,\|\, \pi^{(0)}\big),$$

where (i) is by the bound $(1-\eta\tau)^{-1} \leq \zeta$ and (ii) uses the definition of $L$ in Assumption 1. Plugging the above inequalities into (45) leads to

$$(1 - \eta\tau)\mathbb{E}\left[\mathsf{KL}\big(\pi_\tau^\star \,\|\, \pi^{(t)}\big)\right] + (1 - \eta\tau)^{t+1}\mathsf{KL}\big(\pi_\tau^\star \,\|\, \pi^{(0)}\big)$$

$$\geq \mathbb{E}\left[\mathsf{KL}\big(\pi_\tau^\star \,\|\, \overline{\pi}^{(t+1)}\big)\right] + \frac{\eta\tau}{2}\mathbb{E}\left[\mathsf{KL}\big(\overline{\pi}^{(t+1)} \,\|\, \pi_\tau^\star\big)\right].$$

Then from (40) we have

$$\mathbb{E}\left[\mathsf{KL}\big(\pi_\tau^\star \,\|\, \overline{\pi}^{(t+1)}\big)\right] \leq \mathbb{E}\left[\mathsf{KL}\big(\pi_\tau^\star \,\|\, \overline{\pi}^{(t+1)}\big)\right] + \frac{\eta\tau}{2}\mathbb{E}\left[\mathsf{KL}\big(\overline{\pi}^{(t+1)} \,\|\, \pi_\tau^\star\big)\right]$$

$$\leq (1-\eta\tau)^{t+1}\mathsf{KL}\big(\pi_\tau^\star \,\|\, \pi^{(0)}\big) + (1-\eta\tau)^{t+1}\mathsf{KL}\big(\pi_\tau^\star \,\|\, \pi^{(0)}\big)$$

$$= 2(1-\eta\tau)^{t+1}\mathsf{KL}\big(\pi_\tau^\star \,\|\, \pi^{(0)}\big). \tag{46}$$

**Bounding the QRE-Gap.** Combining (27) and (46), we have

$$\mathbb{E}\left[\mathtt{QRE\text{-}Gap}_\tau(\overline{\pi}^{(t+1)})\right] \leq \tau\mathbb{E}\left[\mathsf{KL}\big(\overline{\pi}^{(t+1)} \,\|\, \pi_\tau^\star\big)\right] + \frac{d_{\mathsf{max}}^2 \|A\|_\infty^2}{\tau}\mathbb{E}\left[\mathsf{KL}\big(\pi_\tau^\star \,\|\, \overline{\pi}^{(t+1)}\big)\right]$$

$$\leq \frac{2}{\eta}\left(\frac{\eta\tau}{2}\mathbb{E}\left[\mathsf{KL}\big(\overline{\pi}^{(t+1)} \,\|\, \pi_\tau^\star\big)\right] + \mathbb{E}\left[\mathsf{KL}\big(\pi_\tau^\star \,\|\, \overline{\pi}^{(t+1)}\big)\right]\right)$$

$$\leq \frac{4(1-\eta\tau)^{t+1}}{\eta}\mathsf{KL}\big(\pi_\tau^\star \,\|\, \pi^{(0)}\big),$$

where the second line uses the learning rate bound

$$\frac{2}{\eta} > \frac{24 d_{\mathsf{max}}^2 \|A\|_\infty^2 (L+1)}{\tau} > \frac{d_{\mathsf{max}}^2 \|A\|_\infty^2}{\tau}.$$

## C  REGRET ANALYSIS OF SINGLE-TIMESCALE OMWU

For completeness, we also provide the regret analysis of single-timescale OMWU in both synchronous and asynchronous settings, which might be of independent interest. To begin, for $\tau \geq 0$, the regret for each player $i \in V$ is defined as

$$\mathsf{Regret}_{i,\tau}(T) = \max_{\pi_i \in \Delta(S_i)} \sum_{t=1}^{T} u_{i,\tau}(\pi_i, \overline{\pi}_{-i}^{(t)}) - \sum_{t=1}^{T} u_{i,\tau}(\overline{\pi}^{(t)}), \tag{47}$$

which measures the performance gap compared to the optimal fixed strategy in hindsight for player $i$, when the rest of the players follow the strategies derived from Algorithm 1.

**Synchronous setting.** We begin with the following no-regret guarantee of single-timescale OMWU in the synchronous setting.

**Theorem 5** (No-regret without delays). *Suppose all players $i \in V$ follow single-timescale OMWU in Algorithm 1 with the initialization $\pi_i^{(0)} = \frac{1}{|S_i|} \mathbf{1}$ and the learning rate obeys $0 < \eta \leq \frac{1}{4 d_{\max} \|A\|_\infty + 4\tau}$. Then, for $T \geq 1$, it holds that*

$$\mathsf{Regret}_{i,\tau}(T) \leq \frac{1}{\eta} \log |S_i| + 16\eta \deg_i \|A\|_\infty^2 \sum_{k \in V} \log |S_k|.$$

By optimizing the learning rate $\eta$, Theorem 5 suggests that the regret is bounded by

$$\max_{i \in V} \mathsf{Regret}_{i,\tau}(T) \lesssim \widetilde{\mathcal{O}} \left( \|A\|_\infty \sqrt{n d_{\max}} \right)$$

up to logarithmic factors. Compared with the OMD method for multi-agent games in Anagnostides et al. (2022), which only provided the regret bound for $\tau = 0$, our bound is more general by allowing entropy regularization. Moreover, our bound is tighter by a factor of $n/d_{\max}$ by exploiting the graph connectivity pattern, which is significant for large sparse graphs.

**Asynchronous setting.** We next move to the asynchronous case, and show that single-timescale OMWU continues to enjoy no-regret learning as long as the delays have finite second-order moments.

**Assumption 4** (Random delays). *Recall the definition of $E(\ell)$ in (12). There exists some constant $\sigma > 0$, such that $\mathbb{E}\left[ (\gamma_i^{(t)})^2 \right] \leq \sum_{\ell=0}^\infty E(\ell) \leq \sigma^2$, for all $t \geq 0$ and $i \in V$.*

Clearly Assumption 4 is weaker than Assumption 1, since it only requires the second-order moments to be finite, instead of an exponential decay of $\gamma_i^{(t)}$. We have the following theorem.

**Theorem 6** (No-regret with random delays). *Under Assumption 4, suppose all players $i \in V$ follow single-timescale OMWU in Algorithm 1 with the initialization $\pi_i^{(0)} = \frac{1}{|S_i|} \mathbf{1}$, the regularization parameter $\tau < \min\{1, \|A\|_\infty\}$, and the learning rate obeys $0 < \eta \leq \frac{\tau}{24 d_{\max}^2 \|A\|_\infty^2 (\sigma^2 + 1)}$. Then, for $T \geq 1$, it holds that*

$$\mathbb{E}\left[ \mathsf{Regret}_{i,\tau}(T) \right] \leq \frac{1}{\eta} \log |S_i| + 8 d_{\max} \|A\|_\infty \left( \frac{d_{\max} \|A\|_\infty}{\tau} + 2 \right) (\sigma^2 + 1) \sum_{i \in V} \log |S_i|. \quad (48)$$

Theorem 6 guarantees that the iterate among $\{\overline{\pi}^{(t)}\}_{t \geq 1}$ enjoys a regret bound on the order of

$$\max_{i \in V} \mathbb{E}\left[ \mathsf{Regret}_{i,\tau}(T) \right] \lesssim \widetilde{\mathcal{O}} \left( \frac{\sigma^2 n d_{\max} \|A\|_\infty^2}{\tau} \right)$$

by optimizing the learning rate $\eta$.

## C.1 PROOF OF THEOREM 5

Recall the expression of the regret

$$\mathsf{Regret}_{i,\tau}(T) = \max_{\pi_i \in \Delta(S_i)} \mathsf{Regret}_{i,\tau}(\pi_i, T),$$

where

$$\begin{aligned}
\mathsf{Regret}_{i,\tau}(\pi_i, T) &:= \sum_{t=1}^T u_{i,\tau}(\pi_i, \overline{\pi}_{-i}^{(t)}) - \sum_{t=1}^T u_{i,\tau}(\overline{\pi}^{(t)}) \\
&= \sum_{t=0}^T \left( \left\langle \pi_i - \overline{\pi}_i^{(t+1)}, A_i \overline{\pi}^{(t+1)} \right\rangle + \tau \mathcal{H}(\pi_i) - \tau \mathcal{H}(\overline{\pi}_i^{(t+1)}) \right). \quad (49)
\end{aligned}$$

Therefore, it is sufficient to bound $\mathsf{Regret}_{i,\tau}(\pi_i, T)$ for any $\pi_i \in \Delta(S_i)$. To begin, we record the following useful lemma whose proof can be found in Appendix E.4.

**Lemma 4.** *For any $\tau \geq 0$ and $T \geq 0$, we have*

$$\sum_{i \in V} \mathsf{Regret}_{i,\tau}(T+1) = \sum_{i \in V} \max_{\pi_i \in \Delta(S_i)} \sum_{t=0}^{T} \left( \left\langle \pi_i - \overline{\pi}_i^{(t+1)}, A_i \overline{\pi}^{(t+1)} \right\rangle + \tau \mathcal{H}(\pi_i) - \tau \mathcal{H}(\overline{\pi}_i^{(t+1)}) \right) \geq 0.$$

Let us now proceed with the following regret decomposition

$$\left\langle \pi_i - \overline{\pi}_i^{(t+1)}, A_i \overline{\pi}^{(t+1)} \right\rangle + \tau \mathcal{H}(\pi_i) - \tau \mathcal{H}(\overline{\pi}_i^{(t+1)})$$

$$= \left\langle \pi_i - \pi_i^{(t+1)}, A_i \overline{\pi}^{(t+1)} \right\rangle + \tau \mathcal{H}(\pi_i) + \left\langle \pi_i^{(t+1)} - \overline{\pi}_i^{(t+1)}, A_i \overline{\pi}^{(t)} \right\rangle - \tau \mathcal{H}(\overline{\pi}_i^{(t+1)})$$

$$\qquad - \left\langle \overline{\pi}_i^{(t+1)} - \pi_i^{(t+1)}, A_i \overline{\pi}^{(t+1)} - A_i \overline{\pi}^{(t)} \right\rangle. \tag{50}$$

We proceed to bound each term on the RHS of (50).

- To begin, note that $\log \pi_i^{(t+1)} \overset{\mathbf{1}}{=} (1 - \eta\tau) \log \pi_i^{(t)} + \eta A_i \overline{\pi}^{(t+1)}$. The first term in (50) can then be written as

$$\left\langle \pi_i - \pi_i^{(t+1)}, A_i \overline{\pi}^{(t+1)} \right\rangle + \tau \mathcal{H}(\pi_i)$$

$$= \frac{1}{\eta} \left\langle \log \pi_i^{(t+1)} - \log \pi_i^{(t)}, \pi_i - \pi_i^{(t+1)} \right\rangle + \tau \left\langle \log \pi_i^{(t)}, \pi_i - \pi_i^{(t+1)} \right\rangle - \tau \left\langle \log \pi_i, \pi_i \right\rangle$$

$$= \left( \frac{1}{\eta} - \tau \right) \mathsf{KL}\left( \pi_i \,\|\, \pi_i^{(t)} \right) - \frac{1}{\eta} \left( \mathsf{KL}\left( \pi_i \,\|\, \pi_i^{(t+1)} \right) + \mathsf{KL}\left( \pi_i^{(t+1)} \,\|\, \pi_i^{(t)} \right) \right) - \tau \left\langle \log \pi_i^{(t)}, \pi_i^{(t+1)} \right\rangle, \tag{51}$$

where the second step is derived from the definition of KL divergence.

- Similarly, the second term in (50) has the form

$$\left\langle \pi_i^{(t+1)} - \overline{\pi}_i^{(t+1)}, A_i \overline{\pi}^{(t)} \right\rangle - \tau \mathcal{H}(\overline{\pi}_i^{(t+1)})$$

$$= \frac{1}{\eta} \left( \mathsf{KL}\left( \pi_i^{(t+1)} \,\|\, \pi_i^{(t)} \right) - \mathsf{KL}\left( \pi_i^{(t+1)} \,\|\, \overline{\pi}_i^{(t+1)} \right) \right) - \left( \frac{1}{\eta} - \tau \right) \mathsf{KL}\left( \overline{\pi}_i^{(t+1)} \,\|\, \pi_i^{(t)} \right)$$

$$\qquad + \tau \left\langle \log \pi_i^{(t)}, \pi_i^{(t+1)} \right\rangle. \tag{52}$$

- Moving to the third term on the RHS of (50), we first make the following claim, which shall be proven at the end of this proof:

$$\left\| \overline{\pi}_i^{(t+1)} - \pi_i^{(t+1)} \right\|_1 \leq \eta \left\| A_i \overline{\pi}^{(t+1)} - A_i \overline{\pi}^{(t)} \right\|_\infty. \tag{53}$$

With (53) in place, we have

$$-\left\langle \pi_i^{(t+1)} - \overline{\pi}_i^{(t+1)}, A_i \overline{\pi}^{(t+1)} - A_i \overline{\pi}^{(t)} \right\rangle \leq \sum_{j \in \mathcal{N}_i} \|A\|_\infty \left\| \pi_i^{(t+1)} - \overline{\pi}_i^{(t+1)} \right\|_1 \left\| \overline{\pi}_j^{(t+1)} - \overline{\pi}_j^{(t)} \right\|_1$$

$$\leq \eta \sum_{j \in \mathcal{N}_i} \|A\|_\infty \left\| A_i(\overline{\pi}^{(t+1)} - \overline{\pi}^{(t)}) \right\|_\infty \left\| \overline{\pi}_j^{(t+1)} - \overline{\pi}_j^{(t)} \right\|_1$$

$$\leq \eta \|A\|_\infty^2 \left( \sum_{j \in \mathcal{N}_i} \left\| \overline{\pi}_j^{(t+1)} - \overline{\pi}_j^{(t)} \right\|_1 \right)^2.$$

The latter term can be further bounded by

$$\left( \sum_{j \in \mathcal{N}_i} \left\| \overline{\pi}_j^{(t+1)} - \overline{\pi}_j^{(t)} \right\|_1 \right)^2 \leq \deg_i \sum_{j \in \mathcal{N}_i} \left\| \overline{\pi}_j^{(t+1)} - \pi_j^{(t)} + \pi_j^{(t)} - \overline{\pi}_j^{(t)} \right\|_1^2$$

$$\leq 2 \deg_i \sum_{j \in \mathcal{N}_i} \left( \left\| \overline{\pi}_j^{(t+1)} - \pi_j^{(t)} \right\|_1^2 + \left\| \pi_j^{(t)} - \overline{\pi}_j^{(t)} \right\|_1^2 \right)$$

$$\leq 4 \deg_i \sum_{j \in \mathcal{N}_i} \left( \mathsf{KL}\left( \overline{\pi}_j^{(t+1)} \,\|\, \pi_j^{(t)} \right) + \mathsf{KL}\left( \pi_j^{(t)} \,\|\, \overline{\pi}_j^{(t)} \right) \right),$$

where it follows respectively from Cauchy-Schwarz inequality, $\|a + b\|_1^2 \leq 2(\|a\|_1^2 + \|b\|_1^2)$, and Pinsker's inequality. Plugging this into the previous inequality leads to

$$-\langle \pi_i^{(t+1)} - \overline{\pi}_i^{(t+1)}, A_i \overline{\pi}^{(t+1)} - A_i \overline{\pi}^{(t)} \rangle \leq 4\eta \deg_i \|A\|_\infty^2 \sum_{j \in \mathcal{N}_i} \left( \mathsf{KL}\big(\overline{\pi}_j^{(t+1)} \,\|\, \pi_j^{(t)}\big) + \mathsf{KL}\big(\pi_j^{(t)} \,\|\, \overline{\pi}_j^{(t)}\big) \right).$$
(54)

Plugging (51), (52), and (54) into (50) yields

$$\langle \pi_i - \overline{\pi}_i^{(t+1)}, A_i \overline{\pi}^{(t+1)} \rangle + \tau \mathcal{H}(\pi_i) - \tau \mathcal{H}(\overline{\pi}_i^{(t+1)})$$
$$\leq \frac{1}{\eta} \left( \mathsf{KL}\big(\pi_i \,\|\, \pi_i^{(t)}\big) - \mathsf{KL}\big(\pi_i \,\|\, \pi_i^{(t+1)}\big) \right) - \frac{1}{\eta} \mathsf{KL}\big(\pi_i^{(t+1)} \,\|\, \overline{\pi}_i^{(t+1)}\big) - \left( \frac{1}{\eta} - \tau \right) \mathsf{KL}\big(\overline{\pi}_i^{(t+1)} \,\|\, \pi_i^{(t)}\big)$$
$$- \tau \mathsf{KL}\big(\pi_i \,\|\, \pi_i^{(t)}\big) + 4\eta \deg_i \|A\|_\infty^2 \sum_{j \in \mathcal{N}_i} \left( \mathsf{KL}\big(\overline{\pi}_j^{(t+1)} \,\|\, \pi_j^{(t)}\big) + \mathsf{KL}\big(\pi_j^{(t)} \,\|\, \overline{\pi}_j^{(t)}\big) \right).$$

Telescoping the sum over $t = 0, 1, \ldots, T$ leads to

$$\mathsf{Regret}_{i,\tau}\big(\pi_i, T+1\big) \leq \frac{1}{\eta} \mathsf{KL}\big(\pi_i \,\|\, \pi_i^{(0)}\big) - \frac{1}{\eta} \sum_{t=0}^{T} \mathsf{KL}\big(\pi_i^{(t+1)} \,\|\, \overline{\pi}_i^{(t+1)}\big) - \left( \frac{1}{\eta} - \tau \right) \sum_{t=0}^{T} \mathsf{KL}\big(\overline{\pi}_i^{(t+1)} \,\|\, \pi_i^{(t)}\big)$$
$$+ 4\eta \deg_i \|A\|_\infty^2 \sum_{t=0}^{T} \sum_{j \in \mathcal{N}_i} \left( \mathsf{KL}\big(\overline{\pi}_j^{(t+1)} \,\|\, \pi_j^{(t)}\big) + \mathsf{KL}\big(\pi_j^{(t)} \,\|\, \overline{\pi}_j^{(t)}\big) \right) \quad (55)$$
$$\leq \frac{1}{\eta} \log |S_i| + 4\eta \deg_i \|A\|_\infty^2 \sum_{t=0}^{T} \left( \mathsf{KL}\big(\overline{\pi}^{(t+1)} \,\|\, \pi^{(t)}\big) + \mathsf{KL}\big(\pi^{(t)} \,\|\, \overline{\pi}^{(t)}\big) \right),$$
(56)

where the last line follows from the fact that $\mathsf{KL}\big(\pi_i \,\|\, \pi_i^{(0)}\big) \leq \log |S_i|$ and $1/\eta > \tau$. The proof is thus complete if we can establish

$$\sum_{t=0}^{T} \mathsf{KL}\big(\pi^{(t)} \,\|\, \overline{\pi}^{(t)}\big) + \sum_{t=0}^{T} \mathsf{KL}\big(\overline{\pi}^{(t+1)} \,\|\, \pi^{(t)}\big) \leq 4 \sum_{k \in V} \log |S_k|.$$
(57)

Therefore, it remains to establish (53) and (57), which shall be completed as follows.

**Proof of (53).** By the update rules of $\overline{\pi}_i^{(t+1)}$ and $\pi_i^{(t+1)}$, from (20) we can deduce that

$$\langle \log \overline{\pi}_i^{(t+1)} - \log \pi_i^{(t+1)}, \overline{\pi}_i^{(t+1)} - \pi_i^{(t+1)} \rangle = \eta \langle A_i(\overline{\pi}^{(t)} - \overline{\pi}^{(t+1)}), \overline{\pi}_i^{(t+1)} - \pi_i^{(t+1)} \rangle. \quad (58)$$

By Pinsker's inequality, we have

$$\langle \log \overline{\pi}_i^{(t+1)} - \log \pi_i^{(t+1)}, \overline{\pi}_i^{(t+1)} - \pi_i^{(t+1)} \rangle \geq \left\| \overline{\pi}_i^{(t+1)} - \pi_i^{(t+1)} \right\|_1^2.$$

In addition,

$$\langle A_i(\overline{\pi}^{(t)} - \overline{\pi}^{(t+1)}), \overline{\pi}_i^{(t+1)} - \pi_i^{(t+1)} \rangle \leq \left\| \overline{\pi}_i^{(t+1)} - \pi_i^{(t+1)} \right\|_1 \left\| A_i(\overline{\pi}^{(t)} - \overline{\pi}^{(t+1)}) \right\|_\infty.$$

Plugging the above two relations into (58) then leads to (53).

**Proof of (57).** Summing (55) over $i \in V$ gives

$$\sum_{i \in V} \mathsf{Regret}_{i,\tau}\big(\pi_i, T+1\big) \leq \frac{1}{\eta} \mathsf{KL}\big(\pi \,\|\, \pi^{(0)}\big) - \frac{1}{\eta} \sum_{t=0}^{T} \mathsf{KL}\big(\pi^{(t+1)} \,\|\, \overline{\pi}^{(t+1)}\big) - \left( \frac{1}{\eta} - \tau \right) \sum_{t=0}^{T} \mathsf{KL}\big(\overline{\pi}^{(t+1)} \,\|\, \pi^{(t)}\big)$$
$$+ 4\eta d_{\mathsf{max}}^2 \|A\|_\infty^2 \sum_{t=0}^{T} \left( \mathsf{KL}\big(\overline{\pi}^{(t+1)} \,\|\, \pi^{(t)}\big) + \mathsf{KL}\big(\pi^{(t)} \,\|\, \overline{\pi}^{(t)}\big) \right)$$
$$\leq \frac{1}{\eta} \left( \mathsf{KL}\big(\pi \,\|\, \pi^{(0)}\big) + \mathsf{KL}\big(\pi^{(0)} \,\|\, \overline{\pi}^{(0)}\big) \right) - \left( \frac{1}{\eta} - \tau - 4\eta d_{\mathsf{max}}^2 \|A\|_\infty^2 \right) \sum_{t=0}^{T} \mathsf{KL}\big(\overline{\pi}^{(t+1)} \,\|\, \pi^{(t)}\big)$$

$$-\left(\frac{1}{\eta} - 4\eta d_{\max}^2 \|A\|_\infty^2\right) \sum_{t=0}^{T} \mathsf{KL}\big(\pi^{(t)} \,\|\, \overline{\pi}^{(t)}\big)$$

$$\leq \frac{1}{\eta} \sum_{k \in V} \log |S_k| - \frac{1}{4\eta} \sum_{t=0}^{T} \mathsf{KL}\big(\overline{\pi}^{(t+1)} \,\|\, \pi^{(t)}\big) - \frac{1}{4\eta} \sum_{t=0}^{T} \mathsf{KL}\big(\pi^{(t)} \,\|\, \overline{\pi}^{(t)}\big),$$

(59)

where the last line follows from $\pi^{(0)} = \overline{\pi}^{(0)}$, $\mathsf{KL}\big(\pi \,\|\, \pi^{(0)}\big) \leq \sum_{k \in V} \log |S_k|$ for any $\pi$ since $\pi^{(0)}$ is a uniform distribution, as well as the choice of the learning rate such that

$$4\eta d_{\max}^2 \|A\|_\infty^2 \leq \frac{1}{4\eta} \qquad \text{and} \qquad \tau \leq \frac{1}{2\eta}.$$

Taking supremum over $\pi$ on both sides of (59) and applying Lemma 4 gives (57) as advertised.

## C.2  PROOF OF THEOREM 6

Similar to the proof of Theorem 5 in Appendix C.1, it suffices to bound $\mathsf{Regret}_{i,\tau}\big(\pi_i, T\big)$ for any $\pi_i \in \Delta(S_i)$, where

$$\mathsf{Regret}_{i,\tau}\big(\pi_i, T\big) := \sum_{t=1}^{T} u_{i,\tau}(\pi_i, \overline{\pi}_{-i}^{(t)}) - \sum_{t=1}^{T} u_{i,\tau}(\overline{\pi}^{(t)})$$

$$= \sum_{t=0}^{T} \left( \big\langle \pi_i - \overline{\pi}_i^{(t+1)}, A_i \overline{\pi}^{(t+1)} \big\rangle + \tau \mathcal{H}(\pi_i) - \tau \mathcal{H}(\overline{\pi}_i^{(t+1)}) \right).$$

(60)

Consider the following decomposition:

$$\big\langle \pi_i - \overline{\pi}_i^{(t+1)}, A_i \overline{\pi}^{(t+1)} \big\rangle + \tau \mathcal{H}(\pi_i) - \tau \mathcal{H}(\overline{\pi}_i^{(t+1)})$$

$$= \big\langle \pi_i - \overline{\pi}_i^{(t+1)}, A_i \overline{\pi}^{(\kappa_i^{(t+1)})} \big\rangle + \big\langle \pi_i - \overline{\pi}_i^{(t+1)}, A_i \overline{\pi}^{(t+1)} - A_i \overline{\pi}^{(\kappa_i^{(t+1)})} \big\rangle + \tau \mathcal{H}(\pi_i) - \tau \mathcal{H}(\overline{\pi}_i^{(t+1)})$$

$$= \big\langle \pi_i - \pi_i^{(t+1)}, A_i \overline{\pi}^{(\kappa_i^{(t+1)})} \big\rangle + \big\langle \pi_i^{(t+1)} - \overline{\pi}_i^{(t+1)}, A_i \overline{\pi}^{(\kappa_i^{(t)})} \big\rangle$$

$$\quad - \big\langle \overline{\pi}_i^{(t+1)} - \pi_i^{(t+1)}, A_i \overline{\pi}^{(\kappa_i^{(t+1)})} - A_i \overline{\pi}^{(\kappa_i^{(t)})} \big\rangle$$

$$\quad + \big\langle \pi_i - \overline{\pi}_i^{(t+1)}, A_i \overline{\pi}^{(t+1)} - A_i \overline{\pi}^{(\kappa_i^{(t+1)})} \big\rangle.$$

(61)

We now bound each term on the RHS of (61). For simplicity, we reuse the short-hand notation in (36).

- To begin with, note that $\pi_i^{(t+1)} = (1 - \eta\tau)\pi_i^{(t)} + \eta A_i \overline{\pi}^{(\kappa_i^{(t+1)})} + c_i \mathbf{1}$ for some normalization constant $c_i$. Thus we have

$$\big\langle \pi_i - \pi_i^{(t+1)}, A_i \overline{\pi}^{(\kappa_i^{(t+1)})} \big\rangle$$

$$= \frac{1}{\eta} \big\langle \log \pi_i^{(t+1)} - \log \pi_i^{(t)}, \pi_i - \pi_i^{(t+1)} \big\rangle + \tau \big\langle \log \pi_i^{(t)}, \pi_i - \pi_i^{(t+1)} \big\rangle$$

$$= \frac{1}{\eta} \left( \mathsf{KL}\big(\pi_i \,\|\, \pi_i^{(t)}\big) - \mathsf{KL}\big(\pi_i \,\|\, \pi_i^{(t+1)}\big) - \mathsf{KL}\big(\pi_i^{(t+1)} \,\|\, \pi_i^{(t)}\big) \right) + \tau \big\langle \log \pi_i^{(t)}, \pi_i - \pi_i^{(t+1)} \big\rangle,$$

(62)

where the second step is derived from the definition of KL-divergence.

- Similarly, it holds that

$$\big\langle \pi_i^{(t+1)} - \overline{\pi}_i^{(t+1)}, A_i \overline{\pi}^{(\kappa_i^{(t)})} \big\rangle$$

$$= \frac{1}{\eta} \left( \mathsf{KL}\big(\pi_i^{(t+1)} \,\|\, \pi_i^{(t)}\big) - \mathsf{KL}\big(\pi_i^{(t+1)} \,\|\, \overline{\pi}_i^{(t+1)}\big) - \mathsf{KL}\big(\overline{\pi}_i^{(t+1)} \,\|\, \pi_i^{(t)}\big) \right) + \tau \big\langle \log \pi_i^{(t)}, \pi_i^{(t+1)} - \overline{\pi}_i^{(t+1)} \big\rangle.$$

(63)

- For the term
$$-\big\langle \pi_i^{(t+1)} - \overline{\pi}_i^{(t+1)},\, A_i\overline{\pi}^{(\kappa_i^{(t+1)})} - A_i\overline{\pi}^{(\kappa_i^{(t)})}\big\rangle$$
in (61), following the deduction of (33), we get

$$
-\big\langle \pi_i^{(t+1)} - \overline{\pi}_i^{(t+1)},\, A_i\overline{\pi}^{(\kappa_i^{(t+1)})} - A_i\overline{\pi}^{(\kappa_i^{(t)})}\big\rangle
$$
$$
\leq 2\,\|A\|_\infty \sum_{j\in\mathcal{N}_i} \left( \Psi_j^{(t)} + \sum_{l=t-\gamma_i^{(t+1)}}^{t-1} \gamma_i^{(t+1)}\Psi_j^{(l)} + \sum_{s=t-\gamma_i^{(t)}}^{t-1} \gamma_i^{(t)}\Psi_j^{(l)} \right) + 2c_A\mathsf{KL}\big(\pi_i^{(t+1)} \,\|\, \overline{\pi}_i^{(t+1)}\big).
$$
$$\tag{64}$$

- For the last term in (61), it similarly follows that

$$
\big\langle \pi_i - \overline{\pi}_i^{(t+1)},\, A_i\overline{\pi}^{(t+1)} - A_i\overline{\pi}^{(\kappa_i^{(t+1)})}\big\rangle
$$
$$
= \big\langle \pi_i - \pi_i^{(t)},\, A_i\overline{\pi}^{(t+1)} - A_i\overline{\pi}^{(\kappa_i^{(t+1)})}\big\rangle + \big\langle \pi_i^{(t)} - \overline{\pi}_i^{(t+1)},\, A_i\overline{\pi}^{(t+1)} - A_i\overline{\pi}^{(\kappa_i^{(t+1)})}\big\rangle
$$
$$
\leq 2c_A\mathsf{KL}\big(\overline{\pi}_i^{(t+1)} \,\|\, \pi_i^{(t)}\big) + 4c_\tau\,\|A\|_\infty \sum_{j\in\mathcal{N}_i} \left( \Psi_j^{(t)} + \sum_{l=t-\gamma_i^{(t+1)}}^{t-1} \gamma_i^{(t+1)}\Psi_j^{(l)} \right).
$$
$$\tag{65}$$

Plugging (62) (63) (64) (65) into (61) yields

$$
\big\langle \pi_i - \overline{\pi}_i^{(t+1)},\, A_i\overline{\pi}^{(t+1)}\big\rangle + \tau\left( \mathcal{H}(\pi_i) - \mathcal{H}(\overline{\pi}_i^{(t+1)}) \right)
$$
$$
\leq \frac{1}{\eta}\left( \mathsf{KL}\big(\pi_i \,\|\, \pi_i^{(t)}\big) - \mathsf{KL}\big(\pi_i \,\|\, \pi_i^{(t+1)}\big) \right) - \left( \frac{1}{\eta} - 2c_A \right)\Psi_i^{(t)}
$$
$$
+ 4c_\tau\,\|A\|_\infty \sum_{j\in\mathcal{N}_i} \Psi_j^{(t)} + 2\gamma_i^{(t)}\,\|A\|_\infty \sum_{j\in\mathcal{N}_i} \sum_{l=t-\gamma_i^{(t)}}^{t-1} \Psi_j^{(l)}
$$
$$
+ 4c_\tau\,\|A\|_\infty \gamma_i^{(t+1)} \sum_{j\in\mathcal{N}_i} \sum_{l=t-\gamma_i^{(t+1)}}^{t-1} \Psi_j^{(l)} + \tau\big\langle \log\pi_i^{(t)},\, \pi_i - \overline{\pi}_i^{(t)}\big\rangle + \tau\left( \mathcal{H}(\pi_i) - \mathcal{H}(\overline{\pi}_i^{(t+1)}) \right).
$$

Note that

$$
\mathcal{H}(\pi_i) - \mathcal{H}(\overline{\pi}_i^{(t+1)}) + \big\langle \log\pi_i^{(t)},\, \pi_i - \overline{\pi}_i^{(t+1)}\big\rangle = -\big\langle \log\pi_i - \log\pi_i^{(t)},\, \pi_i\big\rangle + \big\langle \log\overline{\pi}_i^{(t+1)} - \log\pi_i^{(t)},\, \overline{\pi}_i^{(t+1)}\big\rangle
$$
$$
= \mathsf{KL}\big(\overline{\pi}_i^{(t+1)} \,\|\, \pi_i^{(t)}\big) - \mathsf{KL}\big(\pi_i \,\|\, \pi_i^{(t)}\big).
$$

Then we have

$$
\big\langle \pi_i - \overline{\pi}_i^{(t+1)},\, A_i\overline{\pi}^{(t+1)}\big\rangle + \tau\left( \mathcal{H}(\pi_i) - \mathcal{H}(\overline{\pi}_i^{(t+1)}) \right)
$$
$$
\leq \frac{1}{\eta}\left( \mathsf{KL}\big(\pi_i \,\|\, \pi_i^{(t)}\big) - \mathsf{KL}\big(\pi_i \,\|\, \pi_i^{(t+1)}\big) \right) - \left( \frac{1}{\eta} - 2c_A - \tau \right)\Psi_i^{(t)}
$$
$$
+ 4c_\tau\,\|A\|_\infty \sum_{j\in\mathcal{N}_i} \Psi_j^{(t)} + 2\gamma_i^{(t)}\,\|A\|_\infty \sum_{j\in\mathcal{N}_i} \sum_{l=t-\gamma_i^{(t)}}^{t-1} \Psi_j^{(l)} + 4c_\tau\,\|A\|_\infty \gamma_i^{(t+1)} \sum_{j\in\mathcal{N}_i} \sum_{l=t-\gamma_i^{(t+1)}}^{t-1} \Psi_j^{(l)}.
$$
$$\tag{66}$$

Since the learning rate satisfies

$$
\frac{1}{\eta} \geq \frac{24d_{\mathsf{max}}^2\,\|A\|_\infty^2}{\tau}(\sigma^2 + 1) \geq 2d_{\mathsf{max}}\,\|A\|_\infty + \tau = 2c_A + \tau,
$$

taking expectation on both sides of (66) leads to

$$
\mathbb{E}\left[ \big\langle \pi_i - \overline{\pi}_i^{(t+1)},\, A_i\overline{\pi}^{(t+1)}\big\rangle + \tau\left( \mathcal{H}(\pi_i) - \mathcal{H}(\overline{\pi}_i^{(t+1)}) \right) \right]
$$
$$
\leq \frac{1}{\eta}\mathbb{E}\left[ \mathsf{KL}\big(\pi_i \,\|\, \pi_i^{(t)}\big) - \mathsf{KL}\big(\pi_i \,\|\, \pi_i^{(t+1)}\big) \right] - \left( \frac{1}{\eta} - 2c_A - \tau \right)\mathbb{E}\left[ \Psi_i^{(t)} \right]
$$

$$+ 4c_\tau \|A\|_\infty \, \mathbb{E}\left[\Psi^{(t)}\right] + 4c_\tau \|A\|_\infty \, \mathbb{E}\left[\sum_{l=0}^{t-1} E(t-l)\Psi^{(l)}\right]$$

$$\leq \frac{1}{\eta}\mathbb{E}\left[\mathsf{KL}\big(\pi_i \,\|\, \pi_i^{(t)}\big) - \mathsf{KL}\big(\pi_i \,\|\, \pi_i^{(t+1)}\big)\right] + 4c_\tau \|A\|_\infty \, \mathbb{E}\left[\Psi^{(t)}\right] + 4c_\tau \|A\|_\infty \, \mathbb{E}\left[\sum_{l=0}^{t-1} E(t-l)\Psi^{(l)}\right],$$

$$(67)$$

where we use the fact $\sum_{j\in\mathcal{N}_i} \Psi_j^{(l)} \leq \Psi^{(l)}$ and the definition of $E(t-l)$. Since $\sum_{l=0}^\infty E(l) \leq \sigma^2$ by definition in Assumption 4, summing (67) over $t = 0, 1, \ldots, T$ yields

$$\mathbb{E}\left[\mathsf{Regret}_{i,\tau}\big(T+1\big)\right] \leq \frac{1}{\eta}\mathbb{E}\left[\mathsf{KL}\big(\pi_i \,\|\, \pi_i^{(0)}\big)\right] + 4c_\tau \|A\|_\infty \, (\sigma^2+1)\mathbb{E}\left[\sum_{t=0}^{T}\Psi^{(t)}\right]$$

$$\leq \frac{1}{\eta}\log|S_i| + 4c_\tau \|A\|_\infty \, (\sigma^2+1)\mathbb{E}\left[\sum_{t=0}^{T}\Psi^{(t)}\right]. \qquad (68)$$

It remains to establish

$$\mathbb{E}\left[\sum_{t=0}^{T}\Psi^{(t)}\right] \leq 2\sum_{i\in V}\log|S_i|. \qquad (69)$$

**Proof of (69).** Summing (66) over $i \in V$ gives

$$\sum_{i\in V}\big\langle \pi_i - \overline{\pi}_i^{(t+1)}, \, A_i\overline{\pi}^{(t+1)}\big\rangle + \tau\left(\mathcal{H}(\pi_i) - \mathcal{H}(\overline{\pi}_i^{(t+1)})\right)$$

$$\leq \frac{1}{\eta}\left(\mathsf{KL}\big(\pi \,\|\, \pi^{(t)}\big) - \mathsf{KL}\big(\pi \,\|\, \pi^{(t+1)}\big)\right) - \left(\frac{1}{\eta} - 2c_A - \tau\right)\Psi^{(t)}$$

$$+ 4c_\tau \|A\|_\infty \sum_{i\in V}\sum_{j\in\mathcal{N}_i}\Psi_j^{(t)} + 2\gamma_i^{(t)}\|A\|_\infty \sum_{i\in V}\sum_{j\in\mathcal{N}_i}\sum_{l=t-\gamma_i^{(t)}}^{t-1}\Psi_j^{(l)} + 4c_\tau \|A\|_\infty \sum_{i\in V}\gamma_i^{(t+1)}\sum_{j\in\mathcal{N}_i}\sum_{l=t-\gamma_i^{(t+1)}}^{t-1}\Psi_j^{(l)}.$$

Taking expectation of on both sides and using (38) leads to

$$\mathbb{E}\left[\sum_{i\in V}\big\langle \pi_i - \overline{\pi}_i^{(t+1)}, \, A_i\overline{\pi}^{(t+1)}\big\rangle + \tau\left(\mathcal{H}(\pi_i) - \mathcal{H}(\overline{\pi}_i^{(t+1)})\right)\right]$$

$$\leq \frac{1}{\eta}\mathbb{E}\left[\mathsf{KL}\big(\pi \,\|\, \pi^{(t)}\big) - \mathsf{KL}\big(\pi \,\|\, \pi^{(t+1)}\big)\right] - \left(\frac{1}{\eta} - 4c_A(c_\tau+1) - \tau\right)\mathbb{E}\left[\Psi^{(t)}\right]$$

$$+ 4c_A(c_\tau+1)\mathbb{E}\left[\sum_{l=0}^{t-1}E(t-l)\Psi^{(l)}\right] - \frac{\tau}{2}\mathbb{E}\left[\mathsf{KL}\big(\pi \,\|\, \pi^{(t)}\big)\right].$$

Summing over $t = 0, 1, \ldots, T$ yields

$$\mathbb{E}\left[\sum_{i\in V}\mathsf{Regret}_{i,\tau}\big(\pi, T+1\big)\right] \leq \frac{1}{\eta}\mathbb{E}\left[\mathsf{KL}\big(\pi \,\|\, \pi^{(0)}\big)\right] - \left(\frac{1}{\eta} - 4c_A(c_\tau+1)(\sigma^2+1)\right)\mathbb{E}\left[\sum_{t=0}^{T}\Psi^{(t)}\right]$$

$$\leq \frac{1}{\eta}\mathbb{E}\left[\mathsf{KL}\big(\pi \,\|\, \pi^{(0)}\big)\right] - \frac{1}{2\eta}\mathbb{E}\left[\sum_{t=0}^{T}\Psi^{(t)}\right], \qquad (70)$$

where the second line follows from

$$4c_A(c_\tau+1)(\sigma^2+1)\eta \leq 4d_{\mathsf{max}}\|A\|_\infty\left(2 + \frac{d_{\mathsf{max}}\|A\|_\infty}{\tau}\right)(\sigma^2+1)\frac{\tau}{24d_{\mathsf{max}}^2\|A\|_\infty^2(\sigma^2+1)} < \frac{1}{2}$$

due to $\tau \leq d_{\mathsf{max}}\|A\|_\infty$ and $\eta \leq \frac{\tau}{24d_{\mathsf{max}}^2\|A\|_\infty^2(\sigma^2+1)}$. Taking supremum with respect to $\pi$ on both sides, in view of Lemma 4, we arrive at the advertised bound (69).

# D PROOF FOR TWO-TIMESCALE OMWU (SECTION 4)

## D.1 PROOF OF THEOREM 3

**Bounding** $\mathsf{KL}\big(\pi_\tau^\star \,\|\, \pi^{(t)}\big)$. For notational convenience, we set $\pi^{(t)} = \overline{\pi}^{(t)} = \pi^{(0)}$ for $t < 0$. The following lemma parallels Lemma 2 by focusing on delayed feedbacks. The proof is postponed to Appendix E.5.

**Lemma 5.** *Assuming constant delays* $\gamma_i^{(t)} = \gamma$, *the iterates of OMWU based on the update rule* (10) *satisfy*

$$\big\langle \log \pi^{(t+1)} - (1 - \eta\tau) \log \pi^{(t)} - \eta\tau \log \pi_\tau^\star, \, \overline{\pi}^{(t-\gamma+1)} - \pi_\tau^\star \big\rangle = 0.$$

By following a similar argument in (19), we conclude that

$$\begin{aligned}
\mathsf{KL}\big(\pi_\tau^\star \,\|\, \pi^{(t+1)}\big) = \,&(1 - \eta\tau)\mathsf{KL}\big(\pi_\tau^\star \,\|\, \pi^{(t)}\big) - (1 - \eta\tau)\mathsf{KL}\big(\overline{\pi}^{(t-\gamma+1)} \,\|\, \pi^{(t)}\big) - \mathsf{KL}\big(\pi^{(t+1)} \,\|\, \overline{\pi}^{(t-\gamma+1)}\big) \\
&+ \big\langle \log \overline{\pi}^{(t-\gamma+1)} - \log \pi^{(t+1)}, \, \overline{\pi}^{(t-\gamma+1)} - \pi^{(t+1)} \big\rangle - \eta\tau\mathsf{KL}\big(\overline{\pi}^{(t-\gamma+1)} \,\|\, \pi_\tau^\star\big).
\end{aligned} \tag{71}$$

It boils down to control the term $-\big\langle \log \overline{\pi}^{(t-\gamma+1)} - \log \pi^{(t+1)}, \, \overline{\pi}^{(t-\gamma+1)} - \pi^{(t+1)} \big\rangle$. When $t \geq \gamma$, by taking logarithm on the both sides of the update rules (7) and (10), we have

$$\log \overline{\pi}_i^{(t-\gamma+1)} \overset{1}{=} (1 - \overline{\eta}\tau) \log \pi_i^{(t-\gamma)} + \overline{\eta} A_i \overline{\pi}^{(t-2\gamma)}$$

and

$$\begin{aligned}
\log \pi_i^{(t+1)} &\overset{1}{=} (1 - \eta\tau) \log \pi_i^{(t)} + \eta A_i \overline{\pi}^{(t-\gamma+1)} \\
&\overset{1}{=} (1 - \eta\tau)^{\gamma+1} \log \pi_i^{(t-\gamma)} + \eta \sum_{l=0}^{\gamma} (1 - \eta\tau)^l A_i \overline{\pi}^{(t-\gamma-l+1)}.
\end{aligned}$$

Subtracting the above equalities and taking inner product with $\overline{\pi}_i^{(t-\gamma+1)} - \pi_i^{(t+1)}$ gives

$$\begin{aligned}
&\big\langle \log \overline{\pi}_i^{(t-\gamma+1)} - \log \pi_i^{(t+1)}, \, \overline{\pi}_i^{(t-\gamma+1)} - \pi_i^{(t+1)} \big\rangle \\
&= \eta \sum_{l=0}^{\gamma} (1 - \eta\tau)^l \big\langle \overline{\pi}_i^{(t-\gamma+1)} - \pi_i^{(t+1)}, \, A_i(\overline{\pi}^{(t-2\gamma)} - \overline{\pi}^{(t-\gamma-l+1)}) \big\rangle,
\end{aligned}$$

where the $\log \pi_i^{(t-\gamma)}$ terms cancel out due to the choice $1 - \overline{\eta}\tau = (1 - \eta\tau)^{\gamma+1}$. Summing over $i \in V$,

$$\begin{aligned}
&\big\langle \log \overline{\pi}^{(t-\gamma+1)} - \log \pi^{(t+1)}, \, \overline{\pi}^{(t-\gamma+1)} - \pi^{(t+1)} \big\rangle \\
&= \eta \sum_{i \in V} \sum_{l=0}^{\gamma} (1 - \eta\tau)^l \big\langle \overline{\pi}_i^{(t-\gamma+1)} - \pi_i^{(t+1)}, \, A_i(\overline{\pi}^{(t-2\gamma)} - \overline{\pi}^{(t-\gamma-l+1)}) \big\rangle \\
&\leq \eta \,\|A\|_\infty \sum_{(i,j) \in E} \sum_{l=0}^{\gamma} (1 - \eta\tau)^l \big\|\overline{\pi}_i^{(t-\gamma+1)} - \pi_i^{(t+1)}\big\|_1 \big\|\overline{\pi}_j^{(t-2\gamma)} - \overline{\pi}_j^{(t-\gamma-l+1)}\big\|_1.
\end{aligned} \tag{72}$$

Using the triangle inequality, we can bound $\big\|\overline{\pi}^{(t-2\gamma)} - \overline{\pi}^{(t-\gamma-l+1)}\big\|_1$ as

$$\begin{aligned}
\big\|\overline{\pi}^{(t-2\gamma)} - \overline{\pi}^{(t-\gamma-l+1)}\big\|_1 &\leq \sum_{l_1=t-\gamma}^{t-l} \big\|\overline{\pi}_i^{(l_1-\gamma)} - \overline{\pi}_j^{(l_1-\gamma+1)}\big\|_1 \\
&\leq \sum_{l_1=t-\gamma}^{t-l} \Big( \big\|\overline{\pi}_i^{(l_1-\gamma)} - \pi_i^{(l_1)}\big\|_1 + \big\|\overline{\pi}_j^{(l_1-\gamma+1)} - \pi_j^{(l_1)}\big\|_1 \Big).
\end{aligned}$$

Substitution of the bound into (72) yields

$$\big\langle \log \overline{\pi}^{(t-\gamma+1)} - \log \pi^{(t+1)}, \, \overline{\pi}^{(t-\gamma+1)} - \pi^{(t+1)} \big\rangle$$

$$\leq \eta \|A\|_\infty \sum_{(i,j)\in E} \sum_{l=0}^{\gamma} (1-\eta\tau)^l \sum_{l_1=t-\gamma}^{t-l} \left\|\overline{\pi}_i^{(t-\gamma+1)} - \pi_i^{(t+1)}\right\|_1 \left(\left\|\overline{\pi}_j^{(l_1-\gamma)} - \pi_j^{(l_1)}\right\|_1 + \left\|\overline{\pi}_j^{(l_1-\gamma+1)} - \pi_j^{(l_1)}\right\|_1\right)$$

$$= \eta \|A\|_\infty \sum_{(i,j)\in E} \sum_{l_1=t-\gamma}^{t} \sum_{l=0}^{t-l_1} (1-\eta\tau)^l \left\|\overline{\pi}_i^{(t-\gamma+1)} - \pi_i^{(t+1)}\right\|_1 \left(\left\|\overline{\pi}_j^{(l_1-\gamma)} - \pi_j^{(l_1)}\right\|_1 + \left\|\overline{\pi}_j^{(l_1-\gamma+1)} - \pi_j^{(l_1)}\right\|_1\right)$$

$$\leq \frac{1}{2}\eta \|A\|_\infty \sum_{(i,j)\in E} \left[2 \sum_{l_1=t-\gamma}^{t} \sum_{l=0}^{t-l_1} (1-\eta\tau)^l \left\|\overline{\pi}_i^{(t-\gamma+1)} - \pi_i^{(t+1)}\right\|_1^2 \right.$$

$$\left. + \sum_{l_1=t-\gamma}^{t} \sum_{l=0}^{t-l_1} (1-\eta\tau)^l \left(\left\|\overline{\pi}_j^{(l_1-\gamma)} - \pi_j^{(l_1)}\right\|_1^2 + \left\|\overline{\pi}_j^{(l_1-\gamma+1)} - \pi_j^{(l_1)}\right\|_1^2\right)\right]$$

$$\leq \eta d_{\mathsf{max}} \|A\|_\infty \left[2(\gamma+1)^2 \mathsf{KL}\big(\pi^{(t+1)} \,\|\, \overline{\pi}^{(t-\gamma+1)}\big) \right.$$

$$\left. + \sum_{l_1=t-\gamma}^{t} \sum_{l=0}^{t-l_1} (1-\eta\tau)^l \left(\mathsf{KL}\big(\pi^{(l_1)} \,\|\, \overline{\pi}^{(l_1-\gamma)}\big) + \mathsf{KL}\big(\overline{\pi}^{(l_1-\gamma+1)} \,\|\, \pi^{(l_1)}\big)\right)\right]. \qquad (73)$$

Plugging the above inequality into (71) and recursively applying the inequality gives

$$\mathsf{KL}\big(\pi_\tau^\star \,\|\, \pi^{(t+1)}\big) + \mathsf{KL}\big(\pi^{(t+1)} \,\|\, \overline{\pi}^{(t-\gamma+1)}\big) + \eta\tau \mathsf{KL}\big(\overline{\pi}^{(t-\gamma+1)} \,\|\, \pi_\tau^\star\big)$$

$$\leq (1-\eta\tau)^{t+1-\gamma} \mathsf{KL}\big(\pi_\tau^\star \,\|\, \pi^{(\gamma)}\big) - \sum_{l_1=\gamma}^{t} (1-\eta\tau)^{t-l_1} \left(\mathsf{KL}\big(\pi^{(l_1+1)} \,\|\, \overline{\pi}^{(l_1-\gamma+1)}\big) + (1-\eta\tau)\mathsf{KL}\big(\overline{\pi}^{(l_1-\gamma+1)} \,\|\, \pi^{(l_1)}\big)\right)$$

$$+ \eta d_{\mathsf{max}} \|A\|_\infty \left[2(\gamma+1)^2 \sum_{l_1=\gamma}^{t} (1-\eta\tau)^{t-l_1} \mathsf{KL}\big(\pi^{(l_1+1)} \,\|\, \overline{\pi}^{(l_1-\gamma+1)}\big) \right.$$

$$\left. + \sum_{t_2=\gamma}^{t} (1-\eta\tau)^{t-l_2} \sum_{l_1=l_2-\gamma}^{l_2} \sum_{l=0}^{l_2-l_1} (1-\eta\tau)^l \left(\mathsf{KL}\big(\pi^{(l_1)} \,\|\, \overline{\pi}^{(l_1-\gamma)}\big) + \mathsf{KL}\big(\overline{\pi}^{(l_1-\gamma+1)} \,\|\, \pi^{(l_1)}\big)\right)\right]$$

$$\overset{\text{(i)}}{\leq} (1-\eta\tau)^{t+1-\gamma} \mathsf{KL}\big(\pi_\tau^\star \,\|\, \pi^{(\gamma)}\big) - \sum_{l_1=\gamma}^{t} (1-\eta\tau)^{t-l_1} \left(\mathsf{KL}\big(\pi^{(l_1+1)} \,\|\, \overline{\pi}^{(l_1-\gamma+1)}\big) + (1-\eta\tau)\mathsf{KL}\big(\overline{\pi}^{(l_1-\gamma+1)} \,\|\, \pi^{(l_1)}\big)\right)$$

$$+ 2(\gamma+1)^2 \eta d_{\mathsf{max}} \|A\|_\infty \sum_{l_1=\gamma}^{t} (1-\eta\tau)^{t-l_1} \mathsf{KL}\big(\pi^{(l_1+1)} \,\|\, \overline{\pi}^{(l_1-\gamma+1)}\big)$$

$$+ 2(\gamma+1)^2 \eta d_{\mathsf{max}} \|A\|_\infty \sum_{l_1=0}^{t} (1-\eta\tau)^{t-l_1} \left(\mathsf{KL}\big(\pi^{(l_1+1)} \,\|\, \overline{\pi}^{(l_1-\gamma+1)}\big) + (1-\eta\tau)\mathsf{KL}\big(\overline{\pi}^{(l_1-\gamma+1)} \,\|\, \pi^{(l_1)}\big)\right)$$

$$\overset{\text{(ii)}}{\leq} (1-\eta\tau)^{t+1-\gamma} \mathsf{KL}\big(\pi_\tau^\star \,\|\, \pi^{(\gamma)}\big)$$

$$+ 2(\gamma+1)^2 \eta d_{\mathsf{max}} \|A\|_\infty \sum_{l_1=0}^{\gamma-1} (1-\eta\tau)^{t-l_1} \left(\mathsf{KL}\big(\pi^{(l_1+1)} \,\|\, \overline{\pi}^{(l_1-\gamma+1)}\big) + (1-\eta\tau)\mathsf{KL}\big(\overline{\pi}^{(l_1-\gamma+1)} \,\|\, \pi^{(l_1)}\big)\right),$$

$$(74)$$

where (i) results from basic calculation

$$\sum_{t_2=\gamma}^{t} (1-\eta\tau)^{t-l_2} \sum_{l_1=l_2-\gamma}^{l_2} \sum_{l=0}^{l_2-l_1} (1-\eta\tau)^l \left(\mathsf{KL}\big(\pi^{(l_1)} \,\|\, \overline{\pi}^{(l_1-\gamma)}\big) + \mathsf{KL}\big(\overline{\pi}^{(l_1-\gamma+1)} \,\|\, \pi^{(l_1)}\big)\right)$$

$$= \sum_{l_1=0}^{t} (1-\eta\tau)^{t-l_1} \sum_{l_2=l_1}^{l_1+\gamma} \sum_{l=0}^{l_2-l_1} (1-\eta\tau)^{l_1-l_2+l} \left(\mathsf{KL}\big(\pi^{(l_1)} \,\|\, \overline{\pi}^{(l_1-\gamma)}\big) + \mathsf{KL}\big(\overline{\pi}^{(l_1-\gamma+1)} \,\|\, \pi^{(l_1)}\big)\right)$$

$$= \sum_{l_1=0}^{t} (1-\eta\tau)^{t-l_1} \sum_{l'=0}^{\gamma} \sum_{l=0}^{l'} (1-\eta\tau)^{l-l'} \left(\mathsf{KL}\big(\pi^{(l_1)} \,\|\, \overline{\pi}^{(l_1-\gamma)}\big) + \mathsf{KL}\big(\overline{\pi}^{(l_1-\gamma+1)} \,\|\, \pi^{(l_1)}\big)\right)$$

$$\leq \sum_{l_1=0}^{t} (1-\eta\tau)^{t-l_1} (\gamma+1)^2 \left(1 - \frac{1}{2(\gamma+1)}\right)^{-(\gamma+1)} (1-\eta\tau)\left(\mathsf{KL}\big(\pi^{(l_1)} \,\|\, \overline{\pi}^{(l_1-\gamma)}\big) + \mathsf{KL}\big(\overline{\pi}^{(l_1-\gamma+1)} \,\|\, \pi^{(l_1)}\big)\right)$$

$$\leq 2(\gamma+1)^2 \sum_{l_1=0}^{t} (1-\eta\tau)^{t-l_1} (1-\eta\tau)\left(\mathsf{KL}\big(\pi^{(l_1)} \,\|\, \overline{\pi}^{(l_1-\gamma)}\big) + \mathsf{KL}\big(\overline{\pi}^{(l_1-\gamma+1)} \,\|\, \pi^{(l_1)}\big)\right)$$

$$\leq 2(\gamma+1)^2 \sum_{l_1=0}^{t} (1-\eta\tau)^{t-l_1} \left(\mathsf{KL}\big(\pi^{(l_1+1)} \,\|\, \overline{\pi}^{(l_1-\gamma+1)}\big) + (1-\eta\tau)\mathsf{KL}\big(\overline{\pi}^{(l_1-\gamma+1)} \,\|\, \pi^{(l_1)}\big)\right)$$

and (ii) is due to $\eta \leq \min\left\{\frac{1}{2\tau(\gamma+1)}, \frac{1}{5d_{\max}\|A\|_\infty(\gamma+1)^2}\right\}$. To proceed, we introduce the following lemma concerning the error $\mathsf{KL}\big(\pi_\tau^\star \,\|\, \pi^{(\gamma)}\big)$, with the proof postponed to Appendix E.6.

**Lemma 6.** *With constant delays $\gamma_i^{(t)} = \gamma$, the iterates of OMWU based on the update rule* (10) *satisfy*

$$\mathsf{KL}\big(\pi_\tau^\star \,\|\, \pi^{(\gamma)}\big)$$

$$\leq (1-\eta\tau)^\gamma \mathsf{KL}\big(\pi_\tau^\star \,\|\, \pi^{(0)}\big) - \sum_{l_1=0}^{\gamma-1} (1-\eta\tau)^{\gamma-1-l_1} \left(\mathsf{KL}\big(\pi^{(l_1+1)} \,\|\, \overline{\pi}^{(l_1-\gamma+1)}\big) + (1-\eta\tau)\mathsf{KL}\big(\overline{\pi}^{(l_1-\gamma+1)} \,\|\, \pi^{(l_1)}\big)\right)$$

$$\qquad + 2\eta\gamma^2 d_{\max}\|A\|_\infty.$$

With the lemma above in mind, we can continue to bound (74) by

$$\mathsf{KL}\big(\pi_\tau^\star \,\|\, \pi^{(t+1)}\big)$$

$$\leq (1-\eta\tau)^{t+1} \mathsf{KL}\big(\pi_\tau^\star \,\|\, \pi^{(0)}\big) + 2(1-\eta\tau)^{t+1-\gamma}\eta\gamma^2 d_{\max}\|A\|_\infty$$

$$\qquad - \sum_{l_1=0}^{\gamma-1} (1-\eta\tau)^{t-l_1} \left(\mathsf{KL}\big(\pi^{(l_1+1)} \,\|\, \overline{\pi}^{(l_1-\gamma+1)}\big) + (1-\eta\tau)\mathsf{KL}\big(\overline{\pi}^{(l_1-\gamma+1)} \,\|\, \pi^{(l_1)}\big)\right)$$

$$\qquad + 2(\gamma+1)^2\eta d_{\max}\|A\|_\infty \sum_{l_1=0}^{\gamma-1} (1-\eta\tau)^{t-l_1} \left(\mathsf{KL}\big(\pi^{(l_1+1)} \,\|\, \overline{\pi}^{(l_1-\gamma+1)}\big) + (1-\eta\tau)\mathsf{KL}\big(\overline{\pi}^{(l_1-\gamma+1)} \,\|\, \pi^{(l_1)}\big)\right)$$

$$\leq (1-\eta\tau)^{t+1} \mathsf{KL}\big(\pi_\tau^\star \,\|\, \pi^{(0)}\big) + (1-\eta\tau)^{t+1-\gamma}.$$

**Bounding** $\mathsf{KL}\big(\pi_\tau^\star \,\|\, \overline{\pi}^{(t-\gamma+1)}\big)$. By definition of KL divergence, we have

$$\mathsf{KL}\big(\pi_\tau^\star \,\|\, \overline{\pi}^{(t-\gamma+1)}\big)$$

$$= \mathsf{KL}\big(\pi_\tau^\star \,\|\, \pi^{(t+1)}\big) + \big\langle \pi_\tau^\star, \log\pi^{(t+1)} - \log\overline{\pi}^{(t-\gamma+1)} \big\rangle$$

$$= \mathsf{KL}\big(\pi_\tau^\star \,\|\, \pi^{(t+1)}\big) + \mathsf{KL}\big(\pi^{(t+1)} \,\|\, \overline{\pi}^{(t-\gamma+1)}\big) + \big\langle \pi_\tau^\star - \pi^{(t+1)}, \log\pi^{(t+1)} - \log\overline{\pi}^{(t-\gamma+1)} \big\rangle. \quad (75)$$

It remains to control the term $\big\langle \pi_\tau^\star - \pi^{(t+1)}, \log\pi^{(t+1)} - \log\overline{\pi}^{(t-\gamma+1)} \big\rangle$. By following a similar argument in (73), we have

$$\big\langle \pi_\tau^\star - \pi^{(t+1)}, \log\pi^{(t+1)} - \log\overline{\pi}^{(t-\gamma+1)} \big\rangle$$

$$= \eta \sum_{i\in V} \sum_{l=0}^{\gamma} (1-\eta\tau)^l \big\langle \pi_{i,\tau}^\star - \pi_i^{(t+1)}, \, A_i(\overline{\pi}^{(t-2\gamma)} - \overline{\pi}^{(t-\gamma-l+1)}) \big\rangle$$

$$\leq \eta\|A\|_\infty \sum_{(i,j)\in E} \sum_{l=0}^{\gamma} (1-\eta\tau)^l \big\|\pi_{i,\tau}^\star - \pi_i^{(t+1)}\big\|_1 \big\|\overline{\pi}_j^{(t-2\gamma)} - \overline{\pi}_j^{(t-\gamma-l+1)}\big\|_1$$

$$\leq \eta\|A\|_\infty \sum_{(i,j)\in E} \sum_{l=0}^{\gamma} (1-\eta\tau)^l \sum_{l_1=t-\gamma}^{t-l} \big\|\pi_{i,\tau}^\star - \pi_i^{(t+1)}\big\|_1 \left(\big\|\overline{\pi}_j^{(l_1-\gamma)} - \pi_j^{(l_1)}\big\|_1 + \big\|\overline{\pi}_j^{(l_1-\gamma+1)} - \pi_j^{(l_1)}\big\|_1\right)$$

$$= \eta\|A\|_\infty \sum_{(i,j)\in E} \sum_{l_1=t-\gamma}^{t} \sum_{l=0}^{t-l_1} (1-\eta\tau)^l \big\|\pi_{i,\tau}^\star - \pi_i^{(t+1)}\big\|_1 \left(\big\|\overline{\pi}_j^{(l_1-\gamma)} - \pi_j^{(l_1)}\big\|_1 + \big\|\overline{\pi}_j^{(l_1-\gamma+1)} - \pi_j^{(l_1)}\big\|_1\right)$$

$$\leq \eta d_{\max} \|A\|_\infty \left[ 2(\gamma+1)^2 \mathsf{KL}\big(\pi_\tau^\star \,\|\, \pi^{(t+1)}\big) \right.$$

$$\left. + \sum_{l_1=t-\gamma}^{t} \sum_{l=0}^{t-l_1} (1-\eta\tau)^l \left( \mathsf{KL}\big(\pi^{(l_1)} \,\|\, \overline{\pi}^{(l_1-\gamma)}\big) + \mathsf{KL}\big(\overline{\pi}^{(l_1-\gamma+1)} \,\|\, \pi^{(l_1)}\big) \right) \right].$$

Substitution of the above inequality into (75) yields

$$\mathsf{KL}\big(\pi_\tau^\star \,\|\, \overline{\pi}^{(t-\gamma+1)}\big) + \eta\tau\mathsf{KL}\big(\overline{\pi}^{(t-\gamma+1)} \,\|\, \pi_\tau^\star\big)$$

$$= (1 + 2(\gamma+1)^2 \eta d_{\max}) \mathsf{KL}\big(\pi_\tau^\star \,\|\, \pi^{(t+1)}\big) + \mathsf{KL}\big(\pi^{(t+1)} \,\|\, \overline{\pi}^{(t-\gamma+1)}\big) + \eta\tau\mathsf{KL}\big(\overline{\pi}^{(t-\gamma+1)} \,\|\, \pi_\tau^\star\big)$$

$$+ \eta d_{\max} \|A\|_\infty \sum_{l_1=t-\gamma}^{t} \sum_{l=0}^{t-l_1} (1-\eta\tau)^l \left( \mathsf{KL}\big(\pi^{(l_1)} \,\|\, \overline{\pi}^{(l_1-\gamma)}\big) + \mathsf{KL}\big(\overline{\pi}^{(l_1-\gamma+1)} \,\|\, \pi^{(l_1)}\big) \right)$$

$$\overset{(i)}{\leq} 2\Big( \mathsf{KL}\big(\pi_\tau^\star \,\|\, \pi^{(t+1)}\big) + \mathsf{KL}\big(\pi^{(t+1)} \,\|\, \overline{\pi}^{(t-\gamma+1)}\big) + \eta\tau\mathsf{KL}\big(\overline{\pi}^{(t-\gamma+1)} \,\|\, \pi_\tau^\star\big) \Big)$$

$$+ 2(\gamma+1)\eta d_{\max} \|A\|_\infty \sum_{l_1=0}^{t} (1-\eta\tau)^{t-l_1} \left( \mathsf{KL}\big(\pi^{(l_1+1)} \,\|\, \overline{\pi}^{(l_1-\gamma+1)}\big) + (1-\eta\tau)\mathsf{KL}\big(\overline{\pi}^{(l_1-\gamma+1)} \,\|\, \pi^{(l_1)}\big) \right)$$

$$\overset{(ii)}{\leq} 2(1-\eta\tau)^{t+1-\gamma} \mathsf{KL}\big(\pi_\tau^\star \,\|\, \pi^{(\gamma)}\big) - 2\sum_{l_1=\gamma}^{t} (1-\eta\tau)^{t-l_1} \left( \mathsf{KL}\big(\pi^{(l_1+1)} \,\|\, \overline{\pi}^{(l_1-\gamma+1)}\big) + (1-\eta\tau)\mathsf{KL}\big(\overline{\pi}^{(l_1-\gamma+1)} \,\|\, \pi^{(l_1)}\big) \right)$$

$$+ 4(\gamma+1)^2 \eta d_{\max} \|A\|_\infty \sum_{l_1=\gamma}^{t} (1-\eta\tau)^{t-l_1} \mathsf{KL}\big(\pi^{(l_1+1)} \,\|\, \overline{\pi}^{(l_1-\gamma+1)}\big)$$

$$+ 6(\gamma+1)^2 \eta d_{\max} \|A\|_\infty \sum_{l_1=0}^{t} (1-\eta\tau)^{t-l_1} \left( \mathsf{KL}\big(\pi^{(l_1+1)} \,\|\, \overline{\pi}^{(l_1-\gamma+1)}\big) + (1-\eta\tau)\mathsf{KL}\big(\overline{\pi}^{(l_1-\gamma+1)} \,\|\, \pi^{(l_1)}\big) \right)$$

$$\leq 2(1-\eta\tau)^{t+1-\gamma} \mathsf{KL}\big(\pi_\tau^\star \,\|\, \pi^{(\gamma)}\big)$$

$$+ 6(\gamma+1)^2 \eta d_{\max} \|A\|_\infty \sum_{l_1=0}^{\gamma-1} (1-\eta\tau)^{t-l_1} \left( \mathsf{KL}\big(\pi^{(l_1+1)} \,\|\, \overline{\pi}^{(l_1-\gamma+1)}\big) + (1-\eta\tau)\mathsf{KL}\big(\overline{\pi}^{(l_1-\gamma+1)} \,\|\, \pi^{(l_1)}\big) \right),$$

where (i) results from

$$\sum_{l_1=t-\gamma}^{t} \sum_{l=0}^{t-l_1} (1-\eta\tau)^l \left( \mathsf{KL}\big(\pi^{(l_1)} \,\|\, \overline{\pi}^{(l_1-\gamma)}\big) + \mathsf{KL}\big(\overline{\pi}^{(l_1-\gamma+1)} \,\|\, \pi^{(l_1)}\big) \right)$$

$$= \sum_{l_1=t-\gamma}^{t} (1-\eta\tau)^{t-l_1} \sum_{l=0}^{t-l_1} (1-\eta\tau)^{l+l_1-t} \left( \mathsf{KL}\big(\pi^{(l_1)} \,\|\, \overline{\pi}^{(l_1-\gamma)}\big) + \mathsf{KL}\big(\overline{\pi}^{(l_1-\gamma+1)} \,\|\, \pi^{(l_1)}\big) \right)$$

$$\leq \sum_{l_1=t-\gamma}^{t} (1-\eta\tau)^{t-l_1} (\gamma+1)(1-\eta\tau)^{-(\gamma+1)}(1-\eta\tau) \left( \mathsf{KL}\big(\pi^{(l_1)} \,\|\, \overline{\pi}^{(l_1-\gamma)}\big) + \mathsf{KL}\big(\overline{\pi}^{(l_1-\gamma+1)} \,\|\, \pi^{(l_1)}\big) \right)$$

$$\leq 2(\gamma+1) \sum_{l_1=0}^{t} (1-\eta\tau)^{t-l_1} \left( \mathsf{KL}\big(\pi^{(l_1+1)} \,\|\, \overline{\pi}^{(l_1-\gamma+1)}\big) + (1-\eta\tau)\mathsf{KL}\big(\overline{\pi}^{(l_1-\gamma+1)} \,\|\, \pi^{(l_1)}\big) \right).$$

and (ii) is due to the bound established in (74). Finally, applying Lemma 6 yields

$$\mathsf{KL}\big(\pi_\tau^\star \,\|\, \overline{\pi}^{(t-\gamma+1)}\big) + \eta\tau\mathsf{KL}\big(\overline{\pi}^{(t-\gamma+1)} \,\|\, \pi_\tau^\star\big)$$

$$\leq 2(1-\eta\tau)^{t+1} \mathsf{KL}\big(\pi_\tau^\star \,\|\, \pi^{(0)}\big) + 4(1-\eta\tau)^{t+1-\gamma} \eta\gamma^2 d_{\max} \|A\|_\infty$$

$$- 2\sum_{l_1=0}^{\gamma-1} (1-\eta\tau)^{t-l_1} \left( \mathsf{KL}\big(\pi^{(l_1+1)} \,\|\, \overline{\pi}^{(l_1-\gamma+1)}\big) + (1-\eta\tau)\mathsf{KL}\big(\overline{\pi}^{(l_1-\gamma+1)} \,\|\, \pi^{(l_1)}\big) \right)$$

$$+ 6(\gamma+1)^2 \eta d_{\max} \|A\|_\infty \sum_{l_1=0}^{\gamma-1} (1-\eta\tau)^{t-l_1} \left( \mathsf{KL}\big(\pi^{(l_1+1)} \,\|\, \overline{\pi}^{(l_1-\gamma+1)}\big) + (1-\eta\tau)\mathsf{KL}\big(\overline{\pi}^{(l_1-\gamma+1)} \,\|\, \pi^{(l_1)}\big) \right)$$

$$\leq 2(1 - \eta\tau)^{t+1} \mathsf{KL}\big(\pi_\tau^\star \,\|\, \pi^{(0)}\big) + 2(1 - \eta\tau)^{t+1-\gamma}. \tag{76}$$

**Bounding the QRE gap.** With Lemma 3, we have

$$\begin{aligned}
\mathtt{QRE\text{-}Gap}_\tau\big(\overline{\pi}^{(t-\gamma+1)}\big) &\leq \frac{d_{\mathsf{max}}^2 \|A\|_\infty^2}{\tau} \mathsf{KL}\big(\pi_\tau^\star \,\|\, \overline{\pi}^{(t-\gamma+1)}\big) + \tau \mathsf{KL}\big(\overline{\pi}^{(t-\gamma+1)} \,\|\, \pi_\tau^\star\big) \\
&\leq \max\Big\{ \frac{d_{\mathsf{max}}^2 \|A\|_\infty^2}{\tau}, \frac{1}{\eta} \Big\} \Big( \mathsf{KL}\big(\pi_\tau^\star \,\|\, \overline{\pi}^{(t-\gamma+1)}\big) + \eta\tau \mathsf{KL}\big(\overline{\pi}^{(t-\gamma+1)} \,\|\, \pi_\tau^\star\big) \Big) \\
&\leq 2 \max\Big\{ \frac{d_{\mathsf{max}}^2 \|A\|_\infty^2}{\tau}, \frac{1}{\eta} \Big\} \Big( (1 - \eta\tau)^{t+1} \mathsf{KL}\big(\pi_\tau^\star \,\|\, \pi^{(0)}\big) + (1 - \eta\tau)^{t+1-\gamma} \Big),
\end{aligned}$$

where the last step results from (76).

### D.2 PROOF OF THEOREM 4

**Bounding the term** $\mathsf{KL}\big(\pi_\tau^\star \,\|\, \pi^{(t)}\big)$**.** Recall that the update rule of $\pi_i^{(t)}(k)$ is given by

$$\pi_i^{(t)}(k) \propto \pi_i^{(t-1)}(k)^{1-\eta\tau} \exp(\eta[A_i \overline{\pi}^{(\kappa_i^{(t)})}]_k). \tag{77}$$

We introduce an auxiliary variable $\widetilde{\pi}_i^{(t)}$:

$$\widetilde{\pi}_i^{(t)}(k) \propto \pi_i^{(t-1)}(k)^{1-\widetilde{\eta}_i^{(t)}\tau} \exp\Big( \widetilde{\eta}_i^{(t)} [A_i \overline{\pi}^{(\kappa_i^{(t)})}]_k \Big), \tag{78}$$

which can be viewed as a conceptual alternative update of $\pi_i^{(t)}$ with a different step size $\widetilde{\eta}_i^{(t)} > 0$ satisfying

$$(1 - \widetilde{\eta}_i^{(t)}\tau)(1 - \eta\tau)^{t-\kappa_i^{(t)}} = 1 - \overline{\eta}\tau$$

or equivalently

$$1 - \widetilde{\eta}_i^{(t)}\tau = (1 - \eta\tau)^{\gamma+1-t+\kappa_i^{(t)}}.$$

It directly follows that $\widetilde{\eta}_i^{(t)} \geq \eta$. Since $\kappa_i^{(t)} \leq t$, we have $1 - \widetilde{\eta}_i^{(t)}\tau \geq 1 - (\gamma + 1 - t + \kappa_i^{(t)})\eta\tau \geq 1 - (\gamma + 1)\eta\tau$, which implies $\widetilde{\eta}_i^{(t)} \leq (\gamma + 1)\eta$. For notational convenience, we set $\widetilde{\pi}_i^{(t)} = \pi^{(0)}$, $\widetilde{\eta}_i^{(t)} = \eta$ and $\kappa_i^{(t)} = 0$ when $t \leq 0$. The following lemma establishes a one-step analysis, with the proof postponed to Appendix E.7.

**Lemma 7.** *When $t \geq 1$, it holds that*

$$\begin{aligned}
&\mathsf{KL}\big(\pi_{i,\tau}^\star \,\|\, \pi_i^{(t)}\big) + \eta\tau \mathsf{KL}\big(\overline{\pi}_i^{(\kappa_i^{(t)})} \,\|\, \pi_{i,\tau}^\star\big) \\
&= (1 - \eta\tau) \mathsf{KL}\big(\pi_{i,\tau}^\star \,\|\, \pi_i^{(t-1)}\big) - \eta(\overline{\pi}_i^{(\kappa_i^{(t)})} - \pi_{i,\tau}^\star)^\top A_i (\overline{\pi}^{(\kappa_i^{(t)})} - \pi_\tau^\star) - \psi_i^{(t)} \\
&\quad + \frac{\eta}{\widetilde{\eta}_i^{(t)}} \big\langle \log \overline{\pi}_i^{(\kappa_i^{(t)})} - \log \widetilde{\pi}_i^{(t)}, \overline{\pi}_i^{(\kappa_i^{(t)})} - \widetilde{\pi}_i^{(t)} \big\rangle,
\end{aligned} \tag{79}$$

*where*

$$\begin{aligned}
\psi_i^{(t)} := {}& \Big(1 - \frac{\eta}{\widetilde{\eta}_i^{(t)}}\Big) \mathsf{KL}\big(\pi_i^{(t)} \,\|\, \pi_i^{(t-1)}\big) \\
& + \frac{\eta}{\widetilde{\eta}_i^{(t)}} \Big[ (1 - \widetilde{\eta}_i^{(t)}\tau) \mathsf{KL}\big(\overline{\pi}_i^{(\kappa_i^{(t)})} \,\|\, \pi_i^{(t-1)}\big) + \mathsf{KL}\big(\widetilde{\pi}_i^{(t)} \,\|\, \overline{\pi}_i^{(\kappa_i^{(t)})}\big) + \mathsf{KL}\big(\pi_i^{(t)} \,\|\, \widetilde{\pi}_i^{(t)}\big) \Big].
\end{aligned}$$

We proceed to control the term $\big\langle \log \overline{\pi}_i^{(\kappa_i^{(t)})} - \log \widetilde{\pi}_i^{(t)}, \overline{\pi}_i^{(\kappa_i^{(t)})} - \widetilde{\pi}_i^{(t)} \big\rangle$. By definition, we have

$$\begin{aligned}
\log \widetilde{\pi}_i^{(t)} &\overset{\mathbf{1}}{=} (1 - \widetilde{\eta}_i^{(t)}\tau) \log \pi_i^{(t-1)} + \widetilde{\eta}_i^{(t)} A_i \overline{\pi}^{(\kappa_i^{(t)})} \\
&\overset{\mathbf{1}}{=} (1 - \widetilde{\eta}_i^{(t)}\tau)(1 - \eta\tau)^{t-\kappa_i^{(t)}} \log \pi^{(\kappa_i^{(t)}-1)}
\end{aligned}$$

$$+ \widetilde{\eta}_i^{(t)} \left( A_i \overline{\pi}^{(\kappa_i^{(t)})} + \sum_{l=\kappa_i^{(t)}}^{t-1} (1 - \widetilde{\eta}_i^{(t)} \tau)(1 - \eta\tau)^{t-1-l} A_i \overline{\pi}^{(\kappa_i^{(l)})} \right)$$

and

$$\log \overline{\pi}_i^{(\kappa_i^{(t)})} \overset{\mathbf{1}}{=} (1 - \overline{\eta}\tau) \log \pi^{(\kappa_i^{(t)}-1)} + \overline{\eta} A_i \overline{\pi}^{(\kappa_i^{(t)}-1)}$$

when $\kappa_i^{(t)} \geq 1$. Subtracting the two equations yields

$$\log \overline{\pi}_i^{(\kappa_i^{(t)})} - \log \widetilde{\pi}_i^{(t)} \overset{\mathbf{1}}{=} \widetilde{\eta}_i^{(t)} \Bigg( A_i (\overline{\pi}^{(\kappa_i^{(\kappa_i^{(t)}-1)})} - \overline{\pi}^{(\kappa_i^{(t)})})$$
$$+ \sum_{l=\kappa_i^{(t)}}^{t-1} (1 - \widetilde{\eta}_i^{(t)} \tau)(1 - \eta\tau)^{t-1-l} A_i (\overline{\pi}^{(\kappa_i^{(\kappa_i^{(t)}-1)})} - \overline{\pi}^{(\kappa_i^{(l)})}) \Bigg),$$

(80)

where the $\log \pi^{(\kappa_i^{(t)}-1)}$ terms cancel out due to $(1 - \widetilde{\eta}_i^{(t)} \tau)(1 - \eta\tau)^{t-\kappa_i^{(t)}} = 1 - \overline{\eta}\tau$. It follows that

$$\left\langle \log \overline{\pi}_i^{(\kappa_i^{(t)})} - \log \widetilde{\pi}_i^{(t)}, \pi_i^{(\kappa_i^{(t)})} - \widetilde{\pi}_i^{(t)} \right\rangle$$
$$= \widetilde{\eta}_i^{(t)} \Bigg( \left\langle \overline{\pi}_i^{(\kappa_i^{(t)})} - \widetilde{\pi}_i^{(t)}, A_i(\overline{\pi}^{(\kappa_i^{(\kappa_i^{(t)}-1)})} - \overline{\pi}^{(\kappa_i^{(t)})}) \right\rangle$$
$$+ \sum_{l=\kappa_i^{(t)}}^{t-1} (1 - \widetilde{\eta}_i^{(t)} \tau)(1 - \eta\tau)^{t-1-l} \left\langle \overline{\pi}_i^{(\kappa_i^{(t)})} - \widetilde{\pi}_i^{(t)}, A_i(\overline{\pi}^{(\kappa_i^{(\kappa_i^{(t-1)})})} - \overline{\pi}^{(\kappa_i^{(l)})}) \right\rangle \Bigg)$$
$$\leq \widetilde{\eta}_i^{(t)} \|A\|_\infty \left\| \overline{\pi}_i^{(\kappa_i^{(t)})} - \widetilde{\pi}_i^{(t)} \right\|_1 \sum_{j \in \mathcal{N}_i} \sum_{l=\kappa_i^{(t)}}^{t} \left\| \overline{\pi}_j^{(\kappa_i^{(l)})} - \overline{\pi}_j^{(\kappa_i^{(\kappa_i^{(t-1)})})} \right\|_1. \tag{81}$$

The next lemma establishes an upper bound on the term $\sum_{l=\kappa_i^{(t)}}^{t} \left\| \overline{\pi}_j^{(\kappa_i^{(l)})} - \overline{\pi}_j^{(\kappa_i^{(\kappa_i^{(t-1)})})} \right\|_1$, with the proof postponed to Appendix E.8.

**Lemma 8.** *Let $\nu_j(t)$ denote the time index when agent $j$ receives the payoff from the $t$-th iteration, i.e., $\kappa_j^{(\nu_j(t))} = t$. For $t = 0$, we set $\nu_j(0)$ to an arbitrary index that satisfies $\kappa_j^{(\nu_j(0))} = 0$. When $t \geq 2\gamma + 1$, it holds that*

$$\sum_{l=\kappa_i^{(t)}}^{t} \left\| \overline{\pi}_j^{(\kappa_i^{(l)})} - \overline{\pi}_j^{(\kappa_i^{(\kappa_i^{(t-1)})})} \right\|_1 \leq 4\sqrt{2}(\gamma + 1) \sum_{l=t-2\gamma}^{t+\gamma} \sqrt{\psi_j^{(l)}} + 2\sqrt{2}(\gamma + 1)^2 \sqrt{\psi_j^{(\nu_j(\kappa_i^{(\kappa_i^{(t-1)})}))}},$$

Plugging Lemma 8 into (81) gives

$$\left\langle \log \overline{\pi}_i^{(\kappa_i^{(t)})} - \log \widetilde{\pi}_i^{(t)}, \pi_i^{(\kappa_i^{(t)})} - \widetilde{\pi}_i^{(t)} \right\rangle$$
$$\leq \widetilde{\eta}_i^{(t)} \|A\|_\infty \left\| \overline{\pi}_i^{(\kappa_i^{(k)})} - \widetilde{\pi}_i^{(t)} \right\|_1 \sum_{j \in \mathcal{N}_i} \left[ 4\sqrt{2}(\gamma + 1) \sum_{l=t-2\gamma}^{t+\gamma} \sqrt{\psi_j^{(l)}} + 2\sqrt{2}(\gamma + 1)^2 \sqrt{\psi_j^{(\nu_j(\kappa_i^{(\kappa_i^{(t-1)})}))}} \right]$$
$$\overset{(i)}{\leq} \frac{1}{2} \widetilde{\eta}_i^{(t)} \|A\|_\infty \Bigg\{ 14 d_{\mathsf{max}} (\gamma + 1)^{3/2} \left\| \overline{\pi}_i^{(\kappa_i^{(t)})} - \widetilde{\pi}_i^{(t)} \right\|_1^2$$
$$+ \sum_{j \in \mathcal{N}_i} \left[ 8(\gamma + 1)^{3/2} \sum_{l=t-2\gamma}^{t+\gamma} \psi_j^{(l)} + 4(\gamma + 1)^{5/2} \psi_j^{(\nu_j(\kappa_i^{(\kappa_i^{(t-1)})}))} \right] \Bigg\}$$
$$\overset{(ii)}{\leq} \widetilde{\eta}_i^{(t)} \|A\|_\infty \Bigg\{ 14 d_{\mathsf{max}} (\gamma + 1)^{5/2} \psi_i^{(t)} + \sum_{j \in \mathcal{N}_i} \left[ 4(\gamma + 1)^{3/2} \sum_{l=t-2\gamma}^{t+\gamma} \psi_j^{(l)} + 2(\gamma + 1)^{5/2} \psi_j^{(\nu_j(\kappa_i^{(\kappa_i^{(t-1)})}))} \right] \Bigg\},$$

(82)

where (i) results from Young's inequality

$$\left\|\overline{\pi}_i^{(\kappa_i^{(t)})} - \widetilde{\pi}_i^{(t)}\right\|_1 \sqrt{\psi_j^{(l)}} \leq \frac{1}{2}\left(\frac{1}{\sqrt{2}(\gamma+1)^{1/2}}\left\|\overline{\pi}_i^{(\kappa_i^{(t)})} - \widetilde{\pi}_i^{(t)}\right\|_1^2 + \sqrt{2}(\gamma+1)^{1/2}\psi_j^{(l)}\right)$$

and (ii) follows from $\left\|\overline{\pi}_i^{(\kappa_i^{(t)})} - \widetilde{\pi}_i^{(t)}\right\|_1^2 \leq 2\mathsf{KL}\big(\widetilde{\pi}_i^{(t)} \,\|\, \overline{\pi}_i^{(\kappa_i^{(t)})}\big) \leq 2(\gamma+1)\psi_i^{(t)}$. Plugging (82) into (79) and summing over $i \in V$ yields

$$\mathsf{KL}\big(\pi_\tau^\star \,\|\, \pi^{(t)}\big) + \eta\tau \sum_{i \in V} \mathsf{KL}\big(\overline{\pi}_i^{(\kappa_i^{(t)})} \,\|\, \pi_\tau^\star\big)$$

$$\leq (1 - \eta\tau)\mathsf{KL}\big(\pi_\tau^\star \,\|\, \pi^{(t-1)}\big) - \eta \sum_{i \in V}(\overline{\pi}_i^{(\kappa_i^{(t)})} - \pi_{i,\tau}^\star)^\top A_i(\overline{\pi}^{(\kappa_i^{(t)})} - \pi_\tau^\star)$$

$$- (1 - 14\eta d_{\mathsf{max}}\|A\|_\infty (\gamma+1)^{5/2})\sum_{i \in V}\psi_i^{(t)} + 2\eta\|A\|_\infty (\gamma+1)^{5/2}\sum_{(i,j) \in E}\psi_j^{(\nu_j(\kappa_i^{(\kappa_i^{(t-1)})}))}$$

$$+ 4\eta d_{\mathsf{max}}\|A\|_\infty (\gamma+1)^{3/2}\sum_{l=t-2\gamma}^{t+\gamma}\psi^{(l)}, \tag{83}$$

where we denote $\sum_{i \in V}\psi_i^{(l)}$ by $\psi^{(l)}$ for notation simplicity. We then seek to sum the above equation over $t = 2\gamma + 1, \cdots, T$. Before proceeding, we note that

$$\sum_{t=2\gamma+1}^{T}\sum_{l=t-2\gamma}^{t+\gamma}\psi^{(l)} \leq \sum_{l=1}^{T+\gamma}\sum_{t=l-\gamma}^{l+2\gamma}\psi^{(l)} \leq 3(\gamma+1)\sum_{l=1}^{T+\gamma}\psi^{(l)},$$

and that

$$\sum_{t=2\gamma+1}^{T}\sum_{(i,j) \in E}\psi_j^{(\nu_j(\kappa_i^{(\kappa_i^{(t-1)})}))} \leq \sum_{(i,j) \in E}\sum_{t=0}^{T+\gamma-1}\psi_j^{(t)} \leq d_{\mathsf{max}}\sum_{t=1}^{T+\gamma-1}\psi^{(t)},$$

where the first step is due to the mapping $t \mapsto \nu_j(\kappa_i^{(\kappa_i^{(t-1)})})$ being injective when $t \geq 2\gamma + 1$ (cf. Assumptions 2, 3). Note that $\psi_j^{(t)} = 0$ when $t \leq 0$ and hence can be safely discarded. Taken together, we arrive at

$$\eta\tau\sum_{t=2\gamma+1}^{T}\mathsf{KL}\big(\pi_\tau^\star \,\|\, \pi^{(t)}\big) + \eta\tau\sum_{t=2\gamma+1}^{T}\sum_{i \in V}\mathsf{KL}\big(\overline{\pi}_i^{(\kappa_i^{(t)})} \,\|\, \pi_{i,\tau}^\star\big)$$

$$\leq (1 - \eta\tau)\mathsf{KL}\big(\pi_\tau^\star \,\|\, \pi^{(2\gamma)}\big) - \eta\sum_{t=2\gamma+1}^{T}\sum_{i \in V}(\overline{\pi}_i^{(\kappa_i^{(t)})} - \pi_{i,\tau}^\star)^\top A_i(\overline{\pi}^{(\kappa_i^{(t)})} - \pi_\tau^\star)$$

$$- \left(1 - 14\eta d_{\mathsf{max}}\|A\|_\infty (\gamma+1)^{5/2}\right)\sum_{t=2\gamma+1}^{T}\psi^{(t)} + 12\eta d_{\mathsf{max}}\|A\|_\infty (\gamma+1)^{5/2}\sum_{l=1}^{T+\gamma}\psi^{(l)}$$

$$+ 2\eta d_{\mathsf{max}}\|A\|_\infty (\gamma+1)^{5/2}\sum_{t=1}^{T+\gamma-1}\psi^{(t)}$$

$$\leq (1 - \eta\tau)\mathsf{KL}\big(\pi_\tau^\star \,\|\, \pi^{(2\gamma)}\big) - \eta\sum_{t=2\gamma+1}^{T}\sum_{i \in V}(\overline{\pi}_i^{(\kappa_i^{(t)})} - \pi_{i,\tau}^\star)^\top A_i(\overline{\pi}^{(\kappa_i^{(t)})} - \pi_\tau^\star)$$

$$- \left(1 - 28\eta d_{\mathsf{max}}\|A\|_\infty (\gamma+1)^{5/2}\right)\sum_{t=2\gamma+1}^{T}\psi^{(t)} + 14\eta d_{\mathsf{max}}\|A\|_\infty (\gamma+1)^{5/2}\sum_{l \in \Gamma}\psi^{(l)}$$

$$\leq (1 - \eta\tau)\mathsf{KL}\big(\pi_\tau^\star \,\|\, \pi^{(2\gamma)}\big) - \eta \sum_{t=2\gamma+1}^{T} \sum_{i \in V} (\overline{\pi}_i^{(\kappa_i^{(t)})} - \pi_{i,\tau}^\star)^\top A_i(\overline{\pi}^{(\kappa_i^{(t)})} - \pi_\tau^\star) + \frac{1}{3} \sum_{l \in \Gamma} \psi^{(l)}, \quad (84)$$

where $\Gamma = \{1, \cdots, 2\gamma\} \cup \{T+1, \cdots, T+\gamma\}$. The last step results from the choice of learning rate $\eta \leq \frac{1}{28 d_{\mathsf{max}} \|A\|_\infty (\gamma+1)^{5/2}}$. It now remains to bound the terms $\sum_{t=2\gamma+1}^{T} \sum_{i \in V} (\overline{\pi}_i^{(\kappa_i^{(t)})} - \pi_{i,\tau}^\star)^\top A_i(\overline{\pi}^{(\kappa_i^{(t)})} - \pi_\tau^\star)$, $\mathsf{KL}\big(\pi_\tau^\star \,\|\, \pi^{(2\gamma)}\big)$ and $\sum_{l \in \Gamma} \psi^{(l)}$. In view of Lemma 1, we have

$$- \sum_{t=2\gamma+1}^{T} \sum_{i \in V} (\overline{\pi}_i^{(\kappa_i^{(t)})} - \pi_{i,\tau}^\star)^\top A_i(\overline{\pi}^{(\kappa_i^{(t)})} - \pi_\tau^\star)$$

$$= \sum_{t=\gamma+1}^{T} \sum_{i \in V} (\overline{\pi}_i^{(t)} - \pi_{i,\tau}^\star)^\top A_i(\overline{\pi}^{(t)} - \pi_\tau^\star) - \sum_{t=2\gamma+1}^{T} \sum_{i \in V} (\overline{\pi}_i^{(\kappa_i^{(t)})} - \pi_{i,\tau}^\star)^\top A_i(\overline{\pi}^{(\kappa_i^{(t)})} - \pi_\tau^\star).$$

We remark that each $(\overline{\pi}_i^{(\kappa_i^{(t)})} - \pi_{i,\tau}^\star)^\top A_i(\overline{\pi}^{(\kappa_i^{(t)})} - \pi_\tau^\star)$ term will cancel out due to the mapping $t \mapsto \kappa_i^{(t)}$ being injective when $t \geq \gamma$. In addition, we have a crude bound

$$(\overline{\pi}_i^{(t)} - \pi_{i,\tau}^\star)^\top A_i(\overline{\pi}^{(t)} - \pi_\tau^\star) = \sum_{j \in \mathcal{N}_i} (\overline{\pi}_i^{(t)} - \pi_{i,\tau}^\star)^\top A_{ij}(\overline{\pi}_j^{(t)} - \pi_{j,\tau}^\star) \leq 4 d_{\mathsf{max}} \|A\|_\infty$$

for every $i \in V, t \geq 0$. Applying the bound to the remaining $n\gamma$ terms gives

$$- \sum_{t=2\gamma+1}^{T} \sum_{i \in V} (\overline{\pi}_i^{(\kappa_i^{(t)})} - \pi_{i,\tau}^\star)^\top A_i(\overline{\pi}^{(\kappa_i^{(t)})} - \pi_\tau^\star) \leq 4 n\gamma d_{\mathsf{max}} \|A\|_\infty. \quad (85)$$

The remaining terms $\mathsf{KL}\big(\pi_\tau^\star \,\|\, \pi^{(2\gamma)}\big)$ and $\psi^{(l)}$ can be bounded with the following lemma, with the proof postponed to Appendix E.9.

**Lemma 9.** *It holds for all $i \in V$ and $t \geq 0$ that*

$$\psi_i^{(t)} \leq \eta(d_{\mathsf{max}} \|A\|_\infty (2\gamma + 11) + 3\tau \log |S_i|). \quad (86)$$

*In addition, we have*

$$\mathsf{KL}\big(\pi_{i,\tau}^\star \,\|\, \pi_i^{(2\gamma)}\big) \leq \mathsf{KL}\big(\pi_{i,\tau}^\star \,\|\, \pi_i^{(0)}\big) + 4\eta d_{\mathsf{max}} \|A\|_\infty \gamma. \quad (87)$$

Putting all pieces together, we continue from (84) and show that

$$\eta\tau \sum_{t=2\gamma+1}^{T} \mathsf{KL}\big(\pi_\tau^\star \,\|\, \pi^{(t)}\big)$$

$$\leq (1 - \eta\tau)\mathsf{KL}\big(\pi_\tau^\star \,\|\, \pi^{(2\gamma)}\big) - \eta \sum_{t=2\gamma+1}^{T} \sum_{i \in V} (\overline{\pi}_i^{(\kappa_i^{(t)})} - \pi_{i,\tau}^\star)^\top A_i(\overline{\pi}^{(\kappa_i^{(t)})} - \pi_\tau^\star) + \frac{1}{3} \sum_{l \in \Gamma} \psi^{(l)}$$

$$\leq \mathsf{KL}\big(\pi_{i,\tau}^\star \,\|\, \pi_i^{(0)}\big) + 8\eta n\gamma d_{\mathsf{max}} \|A\|_\infty + \eta\gamma\Big(n d_{\mathsf{max}} \|A\|_\infty (2\gamma + 11) + 3\tau \sum_{i \in V} \log |S_i|\Big)$$

$$\leq \mathsf{KL}\big(\pi_{i,\tau}^\star \,\|\, \pi_i^{(0)}\big) + 8\eta n\Big[\gamma d_{\mathsf{max}} \|A\|_\infty + \gamma\Big(d_{\mathsf{max}} \|A\|_\infty (2\gamma + 11) + 3\tau \log S_{\mathsf{max}}\Big)\Big]$$

$$\leq \mathsf{KL}\big(\pi_{i,\tau}^\star \,\|\, \pi_i^{(0)}\big) + n + 24\eta\tau n\gamma \log S_{\mathsf{max}}.$$

**Bounding the term $\mathsf{KL}\big(\pi_\tau^\star \,\|\, \overline{\pi}^{(t-\gamma+1)}\big)$.** By definition of KL divergence, we have

$$\mathsf{KL}\big(\pi_{i,\tau}^\star \,\|\, \overline{\pi}_i^{(t-\gamma+1)}\big)$$

$$= \mathsf{KL}\big(\pi_{i,\tau}^\star \,\|\, \pi_i^{(t+1)}\big) + \big\langle \pi_{i,\tau}^\star, \log \pi_i^{(t+1)} - \log \overline{\pi}_i^{(t-\gamma+1)} \big\rangle$$

$$= \mathsf{KL}\big(\pi_{i,\tau}^\star \,\|\, \pi_i^{(t+1)}\big) - \mathsf{KL}\big(\overline{\pi}_i^{(t-\gamma+1)} \,\|\, \pi_i^{(t+1)}\big) + \big\langle \pi_{i,\tau}^\star - \overline{\pi}_i^{(t-\gamma+1)}, \log \pi_i^{(t+1)} - \log \overline{\pi}_i^{(t-\gamma+1)} \big\rangle.$$

$$(88)$$

It follows directly from the update rules that

$$\begin{cases} \log \overline{\pi}_i^{(t-\gamma+1)} \overset{\mathbf{1}}{=} (1-\overline{\eta}\tau) \log \pi_i^{(t-\gamma)} + \overline{\eta} A_i \overline{\pi}^{(\kappa_i^{(t-\gamma)})} \\[2mm] \log \pi_i^{(t+1)} \overset{\mathbf{1}}{=} (1-\eta\tau)^{\gamma+1} \log \pi_i^{(t-\gamma)} + \eta \sum_{l=t-\gamma+1}^{t+1} (1-\eta\tau)^{t-l+1} A_i \overline{\pi}^{(\kappa_i^{(l)})}, \end{cases}$$

which enables us to control the term $\left\langle \pi_{i,\tau}^\star - \overline{\pi}_i^{(t-\gamma+1)}, \log \pi_i^{(t+1)} - \log \overline{\pi}_i^{(t-\gamma+1)} \right\rangle$ as

$$\left\langle \pi_{i,\tau}^\star - \overline{\pi}_i^{(t-\gamma+1)}, \log \pi_i^{(t+1)} - \log \overline{\pi}_i^{(t-\gamma+1)} \right\rangle$$

$$= \eta \sum_{l=t-\gamma+1}^{t+1} (1-\eta\tau)^{t-l+1} \left\langle \pi_{i,\tau}^\star - \overline{\pi}_i^{(t-\gamma+1)}, \ A_i(\overline{\pi}^{(\kappa_i^{(t-\gamma)})} - \overline{\pi}^{(\kappa_i^{(l)})}) \right\rangle$$

$$\leq \eta \|A\|_\infty \left\| \pi_{i,\tau}^\star - \overline{\pi}_i^{(t-\gamma+1)} \right\|_1 \sum_{j\in\mathcal{N}_i} \sum_{l=t-\gamma+1}^{t+1} \left\| \overline{\pi}_j^{(\kappa_i^{(t-\gamma)})} - \overline{\pi}_j^{(\kappa_i^{(l)})} \right\|_1. \tag{89}$$

In the same vein as Lemma 8, we can bound the term $\sum_{l=t-\gamma+1}^{t+1} \left\| \overline{\pi}_j^{(\kappa_i^{(t-\gamma)})} - \overline{\pi}_j^{(\kappa_i^{(l)})} \right\|_1$ with $\{\psi_i^{(l)}\}$, as detailed in the following lemma. The proof is omitted due to its similarity with that of Lemma 8.

**Lemma 10.** *When $t \geq 2\gamma$, it holds that*

$$\sum_{l=t-\gamma+1}^{t+1} \left\| \overline{\pi}_j^{(\kappa_i^{(t-\gamma)})} - \overline{\pi}_j^{(\kappa_i^{(l)})} \right\|_1 \leq 4\sqrt{2}(\gamma+1) \sum_{l=t-2\gamma+1}^{t+\gamma+1} \sqrt{\psi_i^{(l)}} + 2\sqrt{2}(\gamma+1)^2 \sqrt{\psi_j^{(\nu_j(\kappa_i^{(t-\gamma)}))}}.$$

Plugging the above lemma into (89), we have

$$\left\langle \pi_{i,\tau}^\star - \overline{\pi}_i^{(t-\gamma+1)}, \log \pi_i^{(t+1)} - \log \overline{\pi}_i^{(t-\gamma+1)} \right\rangle$$

$$\leq \eta \|A\|_\infty \left\| \pi_{i,\tau}^\star - \overline{\pi}_i^{(t-\gamma+1)} \right\|_1 \sum_{j\in\mathcal{N}_i} \left( 4\sqrt{2}(\gamma+1) \sum_{l=t-2\gamma+1}^{t+\gamma+1} \sqrt{\psi_i^{(l)}} + 2\sqrt{2}(\gamma+1)^2 \sqrt{\psi_j^{(\nu_j(\kappa_i^{(t-\gamma)}))}} \right)$$

$$\overset{\text{(i)}}{\leq} \frac{1}{2}\eta \|A\|_\infty \left\{ 14 d_{\mathsf{max}}(\gamma+1)^{3/2} \left\| \pi_{i,\tau}^\star - \overline{\pi}_i^{(t-\gamma+1)} \right\|_1^2 \right.$$

$$\left. + \sum_{j\in\mathcal{N}_i} \left[ 8(\gamma+1)^{3/2} \sum_{l=t-2\gamma+1}^{t+\gamma+1} \psi_j^{(l)} + 4(\gamma+1)^{5/2} \psi_j^{(\nu_j(\kappa_i^{(t-\gamma)}))} \right] \right\}$$

$$\overset{\text{(ii)}}{\leq} \eta \|A\|_\infty \left\{ 14 d_{\mathsf{max}}(\gamma+1)^{3/2} \mathsf{KL}\big( \pi_{i,\tau}^\star \,\|\, \overline{\pi}_i^{(t-\gamma+1)} \big) \right.$$

$$\left. + \sum_{j\in\mathcal{N}_i} \left[ 4(\gamma+1)^{3/2} \sum_{l=t-2\gamma+1}^{t+\gamma+1} \psi_j^{(l)} + 2(\gamma+1)^{5/2} \psi_j^{(\nu_j(\kappa_i^{(t-\gamma)}))} \right] \right\},$$

where (i) results from similar arguments in (82) and (ii) invokes Pinsker's inequality. Substitution of the above inequality into (88) and summing over $i \in V$ leads to

$$(1 - 14\eta d_{\mathsf{max}} \|A\|_\infty (\gamma+1)^{3/2}) \mathsf{KL}\big( \pi_\tau^\star \,\|\, \overline{\pi}^{(t-\gamma+1)} \big)$$

$$\leq \mathsf{KL}\big( \pi_\tau^\star \,\|\, \pi^{(t+1)} \big) + \eta \|A\|_\infty \sum_{(i,j)\in E} \left[ 4(\gamma+1)^{3/2} \sum_{l=t-2\gamma+1}^{t+\gamma+1} \psi_j^{(l)} + 2(\gamma+1)^{5/2} \psi_j^{(\nu_j(\kappa_i^{(t-\gamma)}))} \right]$$

$$\leq \mathsf{KL}\big( \pi_\tau^\star \,\|\, \pi^{(t+1)} \big) + 4\eta d_{\mathsf{max}} \|A\|_\infty (\gamma+1)^{3/2} \sum_{l=t-2\gamma+1}^{t+\gamma+1} \psi^{(l)} + 2\eta d_{\mathsf{max}} \|A\|_\infty (\gamma+1)^{5/2} \psi^{(\nu_j(\kappa_i^{(t-\gamma)}))}.$$

Summing the above inequality over $t = 2\gamma - 1, \cdots, T - 1$ and adding $\sum_{t=2\gamma}^{T-1} \sum_{i\in V} \mathsf{KL}\big( \overline{\pi}_i^{(\kappa_i^{(t+1)})} \,\|\, \pi_{i,\tau}^\star \big)$ to the both sides,

$$\sum_{t=2\gamma}^{T-1} \left( \frac{2}{3} \mathsf{KL}\big( \pi_\tau^\star \,\|\, \overline{\pi}^{(t-\gamma+1)} \big) + \sum_{i\in V} \mathsf{KL}\big( \overline{\pi}_i^{(\kappa_i^{(t+1)})} \,\|\, \pi_{i,\tau}^\star \big) \right)$$

$$\leq \sum_{t=2\gamma}^{T-1} \mathsf{KL}\big(\pi_\tau^\star \,\|\, \pi^{(t+1)}\big) + \sum_{t=2\gamma}^{T-1} \sum_{i\in V} \mathsf{KL}\big(\overline{\pi}_i^{(\kappa_i^{(t+1)})} \,\|\, \pi_{i,\tau}^\star\big)$$

$$+ 4\eta d_{\max} \|A\|_\infty (\gamma+1)^{3/2} \sum_{t=2\gamma}^{T-1} \sum_{l=t-2\gamma+1}^{t+\gamma+1} \psi^{(l)} + 2\eta d_{\max} \|A\|_\infty (\gamma+1)^{5/2} \sum_{t=2\gamma}^{T-1} \psi^{(\nu_j(\kappa_i^{(t-\gamma)}))}$$

$$\stackrel{(i)}{\leq} \frac{1}{\eta\tau}\bigg\{ (1-\eta\tau)\mathsf{KL}\big(\pi_\tau^\star \,\|\, \pi^{(2\gamma)}\big) - \eta \sum_{t=2\gamma+1}^{T} \sum_{i\in V} (\overline{\pi}_i^{(\kappa_i^{(t)})} - \pi_{i,\tau}^\star)^\top A_i(\overline{\pi}^{(\kappa_i^{(t)})} - \pi_\tau^\star)$$

$$- \Big(1 - 28\eta d_{\max} \|A\|_\infty (\gamma+1)^{5/2}\Big) \sum_{t=2\gamma+1}^{T} \psi^{(t)} + 14\eta d_{\max} \|A\|_\infty (\gamma+1)^{5/2} \sum_{l\in\Gamma} \psi^{(l)} \bigg\}$$

$$+ 12\eta d_{\max} \|A\|_\infty (\gamma+1)^{5/2} \sum_{l=1}^{T+\gamma} \psi^{(l)} + 2\eta d_{\max} \|A\|_\infty (\gamma+1)^{5/2} \sum_{t=0}^{T+\gamma-1} \psi^{(l)}$$

$$= \frac{1}{\eta\tau}\bigg\{ (1-\eta\tau)\mathsf{KL}\big(\pi_\tau^\star \,\|\, \pi^{(2\gamma)}\big) - \eta \sum_{t=2\gamma+1}^{T} \sum_{i\in V} (\overline{\pi}_i^{(\kappa_i^{(t)})} - \pi_{i,\tau}^\star)^\top A_i(\overline{\pi}^{(\kappa_i^{(t)})} - \pi_\tau^\star)$$

$$- \Big(1 - 28(1+\tfrac{\eta\tau}{2})\eta d_{\max} \|A\|_\infty (\gamma+1)^{5/2}\Big) \sum_{t=2\gamma+1}^{T} \psi^{(t)} + 14(1+\eta\tau)\eta d_{\max} \|A\|_\infty (\gamma+1)^{5/2} \sum_{l\in\Gamma} \psi^{(l)} \bigg\}.$$

Here, (i) invokes the bound established in (84). We remark that our choice of learning rate

$$\eta \leq \min\bigg\{ \frac{1}{2\tau(\gamma+1)}, \frac{1}{42 d_{\max} \|A\|_\infty (\gamma+1)^{5/2}} \bigg\}$$

guarantees $1 - 28(1+\frac{\eta\tau}{2})\eta d_{\max} \|A\|_\infty (\gamma+1)^{5/2} \geq 0$. This taken together with (85) and Lemma 9 gives

$$\sum_{t=2\gamma}^{T-1} \bigg( \frac{2}{3}\mathsf{KL}\big(\pi_\tau^\star \,\|\, \overline{\pi}^{(t-\gamma+1)}\big) + \sum_{i\in V} \mathsf{KL}\big(\overline{\pi}_i^{(\kappa_i^{(t+1)})} \,\|\, \pi_{i,\tau}^\star\big) \bigg)$$

$$\leq \frac{1}{\eta\tau}\bigg\{ (1-\eta\tau)\mathsf{KL}\big(\pi_\tau^\star \,\|\, \pi^{(2\gamma)}\big) - \eta \sum_{t=2\gamma+1}^{T} \sum_{i\in V} (\overline{\pi}_i^{(\kappa_i^{(t)})} - \pi_{i,\tau}^\star)^\top A_i(\overline{\pi}^{(\kappa_i^{(t)})} - \pi_\tau^\star) + \frac{1}{2}\sum_{l\in\Gamma} \psi^{(l)} \bigg\}$$

$$\leq \frac{1}{\eta\tau}\bigg\{ \mathsf{KL}\big(\pi_{i,\tau}^\star \,\|\, \pi_i^{(0)}\big) + 8\eta n\Big[\gamma d_{\max} \|A\|_\infty + \frac{3\gamma}{2}\Big(d_{\max} \|A\|_\infty (2\gamma+11) + 3\tau \log S_{\max}\Big)\Big] \bigg\}$$

$$\leq \frac{1}{\eta\tau}\bigg\{ \mathsf{KL}\big(\pi_{i,\tau}^\star \,\|\, \pi_i^{(0)}\big) + n + 36\eta\tau n\gamma \log S_{\max} \bigg\}. \tag{90}$$

**Bounding the QRE gap.** With Lemma 3, we have

$$\sum_{t=2\gamma}^{T-\gamma-1} \mathtt{QRE\text{-}Gap}_\tau(\overline{\pi}^{(t+1)}) \leq \sum_{t=2\gamma}^{T-\gamma-1} \Big( \frac{d_{\max}^2 \|A\|_\infty^2}{\tau} \mathsf{KL}\big(\pi_\tau^\star \,\|\, \overline{\pi}^{(t+1)}\big) + \tau\mathsf{KL}\big(\overline{\pi}^{(t+1)} \,\|\, \pi_\tau^\star\big) \Big)$$

$$\leq \max\bigg\{ \frac{3 d_{\max}^2 \|A\|_\infty^2}{2\tau}, \tau \bigg\} \sum_{t=2\gamma}^{T-\gamma-1} \Big( \frac{2}{3}\mathsf{KL}\big(\pi_\tau^\star \,\|\, \overline{\pi}^{(t+1)}\big) + \mathsf{KL}\big(\overline{\pi}^{(t+1)} \,\|\, \pi_\tau^\star\big) \Big).$$

Since the mapping $t \mapsto \nu_i(t)$ is injective, we have

$$\sum_{t=2\gamma}^{T-\gamma-1} \sum_{i\in V} \mathsf{KL}\big(\overline{\pi}_i^{(t+1)} \,\|\, \pi_{i,\tau}^\star\big) = \sum_{t=2\gamma}^{T-\gamma-1} \sum_{i\in V} \mathsf{KL}\big(\overline{\pi}_i^{(\kappa_i^{(\nu_i(t+1))})} \,\|\, \pi_{i,\tau}^\star\big) \leq \sum_{t=2\gamma}^{T-1} \sum_{i\in V} \mathsf{KL}\big(\overline{\pi}_i^{(\kappa_i^{(t+1)})} \,\|\, \pi_{i,\tau}^\star\big).$$

Combining the above two equalities gives

$$\sum_{t=2\gamma}^{T-\gamma-1} \mathtt{QRE\text{-}Gap}_\tau(\overline{\pi}^{(t+1)})$$

$$\leq \max\left\{\frac{3d_{\max}^2\|A\|_\infty^2}{2\tau}, \tau\right\}\left(\sum_{t=2\gamma}^{T-\gamma-1}\frac{2}{3}\mathsf{KL}\big(\pi_\tau^\star \,\|\, \overline{\pi}^{(t+1)}\big) + \sum_{t=2\gamma}^{T-1}\mathsf{KL}\big(\overline{\pi}^{(t+1)} \,\|\, \pi_\tau^\star\big)\right)$$

$$\leq \max\left\{\frac{3d_{\max}^2\|A\|_\infty^2}{2\tau}, \tau\right\}\sum_{t=2\gamma}^{T-1}\left(\frac{2}{3}\mathsf{KL}\big(\pi_\tau^\star \,\|\, \overline{\pi}^{(t-\gamma+1)}\big) + \sum_{i\in V}\mathsf{KL}\big(\overline{\pi}_i^{(\kappa_i^{(t+1)})} \,\|\, \pi_{i,\tau}^\star\big)\right)$$

$$\leq \max\left\{\frac{3d_{\max}^2\|A\|_\infty^2}{2\tau}, \tau\right\}\frac{1}{\eta\tau}\left(\mathsf{KL}\big(\pi_{i,\tau}^\star \,\|\, \pi_i^{(0)}\big) + n + 36\eta\tau n\gamma\log S_{\max}\right),$$

where the last step results from (90).

## E PROOF OF AUXILIARY LEMMAS

### E.1 PROOF OF LEMMA 1

To prove this lemma, we recall a key observation in Cai et al. (2016) that allows one to transform a zero-sum polymatrix game $\mathcal{G} = \{(V,E), \{S_i\}_{i\in V}, \{A_{ij}\}_{(i,j)\in E}\}$ into a pairwise constant-sum polymatrix game $\widetilde{\mathcal{G}} = \{(V,E), \{S_i\}_{i\in V}, \{\widetilde{A}_{ij}\}_{(i,j)\in E}\}$ such that

(1) For every player $i \in V$, it has the same payoff in $\mathcal{G}$ and $\widetilde{\mathcal{G}}$:
$$u_i(s) = \widetilde{u}_i(s), \qquad \forall s \in S.$$

(2) For each pair $(i,j) \in E$, $i \neq j$, the two-player game $\widetilde{\mathcal{G}}$ is constant-sum, i.e., there exist constants $\alpha_{ij} = \alpha_{ji}$, such that
$$\widetilde{A}_{ij}(s_i, s_j) + \widetilde{A}_{ji}(s_j, s_i) = \alpha_{ij} \tag{91}$$
holds for all $s_i \in S_i, s_j \in S_j$.

We are now in a place to prove Lemma 1. Let $\widetilde{\mathcal{G}}$ be the pairwise constant-sum polymatrix game associated with $\mathcal{G}$ after the above payoff preserving transformation. We have

$$\sum_{i\in V}\big[u_i(\pi_i, \pi_{-i}') + u_i(\pi_i', \pi_{-i})\big]$$

$$= \sum_{i\in V}\big[\widetilde{u}_i(\pi_i, \pi_{-i}') + \widetilde{u}_i(\pi_i', \pi_{-i})\big]$$

$$= \sum_{(i,j)\in E}\left[\mathop{\mathbb{E}}_{s_i\sim\pi_i, s_j\sim\pi_j'}\big[\widetilde{A}_{ij}(s_i, s_j)\big] + \mathop{\mathbb{E}}_{s_i\sim\pi_i', s_j\sim\pi_j}\big[\widetilde{A}_{ij}(s_i, s_j)\big]\right]$$

$$= \sum_{(i,j)\in E}\left[\mathop{\mathbb{E}}_{s_i\sim\pi_i, s_j\sim\pi_j'}\big[\widetilde{A}_{ij}(s_i, s_j)\big] + \mathop{\mathbb{E}}_{s_i\sim\pi_i', s_j\sim\pi_j}\big[\alpha_{ij} - \widetilde{A}_{ji}(s_j, s_i)\big]\right]$$

$$= \sum_{(i,j)\in E}\alpha_{ij} = 0,$$

where the penultimate line uses (91), and the last line uses the fact that $\widetilde{\mathcal{G}}$ is also a zero-sum polymatrix game, which satisfies

$$\sum_{(i,j)\in E}\alpha_{ij} = \sum_{(i,j)\in E}\left[\widetilde{A}_{ij}(s_i, s_j) + \widetilde{A}_{ji}(s_j, s_i)\right] = \sum_{i\in V}\widetilde{u}_i(s) + \sum_{j\in V}\widetilde{u}_j(s) = 0$$

for any arbitrary $s \in S$.

### E.2 PROOF OF LEMMA 2

In view of the update rule (7), we have
$$\log\pi_i^{(t+1)} = (1 - \eta\tau)\log\pi_i^{(t)} + \eta A_i\overline{\pi}^{(t+1)} + c_i\mathbf{1}$$

for some constant $c_i$. On the other hand, it follows from the expression of QRE in (4) that

$$\eta\tau \log \pi_{i,\tau}^\star = \eta A_i \pi_\tau^\star + c_i^\star \mathbf{1} \tag{92}$$

for some constant $c_i^\star$. By combining the above two equalities and taking the inner product with $\overline{\pi}_i^{(t+1)} - \pi_{i,\tau}^\star$, we have

$$\left\langle \log \pi_i^{(t+1)} - (1 - \eta\tau) \log \pi_i^{(t)} - \eta\tau \log \pi_{i,\tau}^\star, \; \overline{\pi}_i^{(t+1)} - \pi_{i,\tau}^\star \right\rangle = \eta (\overline{\pi}_i^{(t+1)} - \pi_{i,\tau}^\star)^\top A_i (\overline{\pi}^{(t+1)} - \pi_\tau^\star). \tag{93}$$

Summing the above equality over $i \in V$ gives

$$\begin{aligned}
&\left\langle \log \pi^{(t+1)} - (1 - \eta\tau) \log \pi^{(t)} - \eta\tau \log \pi_\tau^\star, \; \overline{\pi}^{(t+1)} - \pi_\tau^\star \right\rangle \\
&= \eta \sum_{i \in V} (\overline{\pi}_i^{(t+1)} - \pi_{i,\tau}^\star)^\top A_i (\overline{\pi}^{(t+1)} - \pi_\tau^\star) \\
&= \eta \sum_{i \in V} \left[ (\overline{\pi}_i^{(t+1)})^\top A_i \overline{\pi}^{(t+1)} + (\pi_{i,\tau}^\star)^\top A_i \pi_\tau^\star \right] - \eta \sum_{i \in V} \left[ (\overline{\pi}_i^{(t+1)})^\top A_i \pi_\tau^\star + (\pi_{i,\tau}^\star)^\top A_i \overline{\pi}^{(t+1)} \right] \\
&= \eta \sum_{i \in V} \left[ u_i(\overline{\pi}^{(t+1)}) + u_i(\pi_\tau^\star) \right] = 0,
\end{aligned}$$

where the last line follows from $\sum_{i \in V} \left[ (\overline{\pi}_i^{(t+1)})^\top A_i \pi_\tau^\star + (\pi_{i,\tau}^\star)^\top A_i \overline{\pi}^{(t+1)} \right] = 0$ due to Lemma 1, as well as that the game is zero-sum.

### E.3 Proof of Lemma 3

Recalling the definition

$$\begin{aligned}
\texttt{QRE-Gap}_\tau(\pi) &= \max_{i \in V} \left[ \max_{\pi_i' \in \Delta(S_i)} u_{i,\tau}(\pi_i', \pi_{-i}) - u_{i,\tau}(\pi) \right] \\
&\leq \sum_{i \in V} \left[ \max_{\pi_i' \in \Delta(S_i)} u_{i,\tau}(\pi_i', \pi_{-i}) - u_{i,\tau}(\pi) \right] \\
&= \max_{i \in V: \pi_i' \in \Delta(S_i)} \sum_{i \in V} \left[ u_{i,\tau}(\pi_i', \pi_{-i}) - u_{i,\tau}(\pi_i, \pi_{-i}) \right],
\end{aligned}$$

where the inequality holds since $\max_{\pi_i' \in \Delta(S_i)} u_{i,\tau}(\pi_i', \pi_{-i}) - u_{i,\tau}(\pi) \geq u_{i,\tau}(\pi_i, \pi_{-i}) - u_{i,\tau}(\pi) = 0$ for all $i \in V$. We now proceed to decompose

$$\begin{aligned}
&\sum_{i \in V} \left[ u_{i,\tau}(\pi_i', \pi_{-i}) - u_{i,\tau}(\pi_i, \pi_{-i}) \right] \\
&= \sum_{i \in V} \left[ u_{i,\tau}(\pi_i', \pi_{-i}) - u_{i,\tau}(\pi_{i,\tau}^\star, \pi_{-i,\tau}^\star) \right] - \tau \sum_{i \in V} \left( \mathcal{H}(\pi_i) - \mathcal{H}(\pi_{i,\tau}^\star) \right) \\
&= \sum_{i \in V} \left[ u_{i,\tau}(\pi_i', \pi_{-i}) - u_{i,\tau}(\pi_i', \pi_{-i,\tau}^\star) - u_{i,\tau}(\pi_{i,\tau}^\star, \pi_{-i}) + u_{i,\tau}(\pi_{i,\tau}^\star, \pi_{-i,\tau}^\star) \right] \\
&\quad + \sum_{i \in V} \left[ u_{i,\tau}(\pi_{i,\tau}^\star, \pi_{-i}) - u_{i,\tau}(\pi_{i,\tau}^\star, \pi_{-i,\tau}^\star) - \tau \left( \mathcal{H}(\pi_i) - \mathcal{H}(\pi_{i,\tau}^\star) \right) \right] \\
&\quad + \sum_{i \in V} \left[ u_{i,\tau}(\pi_i', \pi_{-i,\tau}^\star) - u_{i,\tau}(\pi_{i,\tau}^\star, \pi_{-i,\tau}^\star) \right] \tag{94}
\end{aligned}$$

where the first line follows from $\sum_{i \in V} (u_{i,\tau}(\pi) - \tau \mathcal{H}(\pi_i)) = \sum_{i \in V} \left( u_{i,\tau}(\pi_\tau^\star) - \tau \mathcal{H}(\pi_{i,\tau}^\star) \right) = 0$ by the definition of zero-sum games. It boils down to control the terms on the RHS of (94).

- To control the first term, by the definition of $u_{i,\tau}$ in (5) (see also (3)), it follows that

$$\begin{aligned}
&u_{i,\tau}(\pi_i', \pi_{-i}) - u_{i,\tau}(\pi_i', \pi_{-i,\tau}^\star) - u_{i,\tau}(\pi_{i,\tau}^\star, \pi_{-i}) + u_{i,\tau}(\pi_{i,\tau}^\star, \pi_{-i,\tau}^\star) \\
&= u_i(\pi_i', \pi_{-i}) - u_i(\pi_i', \pi_{-i,\tau}^\star) - u_i(\pi_{i,\tau}^\star, \pi_{-i}) + u_i(\pi_{i,\tau}^\star, \pi_{-i,\tau}^\star)
\end{aligned}$$

$$= (\pi_i' - \pi_{i,\tau}^\star)^\top A_i(\pi - \pi_\tau^\star) = \sum_{j \in \mathcal{N}_i} (\pi_i' - \pi_{i,\tau}^\star)^\top A_{ij}(\pi_j - \pi_{j,\tau}^\star),$$

which each summand can be further bounded by Young's inequality and Pinsker's inequality as

$$(\pi_i' - \pi_{i,\tau}^\star)^\top A_{ij}(\pi_j - \pi_{j,\tau}^\star) \le \|A\|_\infty \left\|\pi_i' - \pi_{i,\tau}^\star\right\|_1 \left\|\pi_j - \pi_{j,\tau}^\star\right\|_1$$

$$\le \frac{1}{2}\|A\|_\infty \left( \frac{\tau}{d_{\mathsf{max}}\|A\|_\infty} \left\|\pi_i' - \pi_{i,\tau}^\star\right\|_1^2 + \frac{d_{\mathsf{max}}\|A\|_\infty}{\tau} \left\|\pi_j - \pi_{j,\tau}^\star\right\|_1^2 \right)$$

$$\le \|A\|_\infty \left( \frac{\tau}{d_{\mathsf{max}}\|A\|_\infty} \mathsf{KL}\big(\pi_i' \,\|\, \pi_{i,\tau}^\star\big) + \frac{d_{\mathsf{max}}\|A\|_\infty}{\tau} \mathsf{KL}\big(\pi_{j,\tau}^\star \,\|\, \pi_j\big) \right).$$

Summing the inequality over $i, j$ gives

$$\sum_{i \in V} \left[ u_{i,\tau}(\pi_i', \pi_{-i}) - u_{i,\tau}(\pi_i', \pi_{-i,\tau}^\star) - u_{i,\tau}(\pi_{i,\tau}^\star, \pi_{-i}) + u_{i,\tau}(\pi_{i,\tau}^\star, \pi_{-i,\tau}^\star) \right]$$

$$\le \tau \mathsf{KL}\big(\pi' \,\|\, \pi_\tau^\star\big) + \frac{d_{\mathsf{max}}^2 \|A\|_\infty^2}{\tau} \mathsf{KL}\big(\pi_\tau^\star \,\|\, \pi\big). \tag{95}$$

- Regarding the second term, we have

$$\sum_{i \in V} \left[ u_{i,\tau}(\pi_{i,\tau}^\star, \pi_{-i}) - u_{i,\tau}(\pi_{i,\tau}^\star, \pi_{-i,\tau}^\star) - \tau\left(\mathcal{H}(\pi_i) - \mathcal{H}(\pi_{i,\tau}^\star)\right) \right]$$

$$= \sum_{i \in V} \left[ (\pi_{i,\tau}^\star)^\top A_i(\pi - \pi_\tau^\star) + \tau(\pi_i^\top \log \pi_i - (\pi_{i,\tau}^\star)^\top \log \pi_{i,\tau}^\star) \right]$$

$$= \sum_{i \in V} \left[ (\pi_{i,\tau}^\star)^\top A_i(\pi - \pi_\tau^\star) + \tau\left(\langle \pi_i,\, \log \pi_i - \log \pi_{i,\tau}^\star \rangle + \langle \pi_i - \pi_{i,\tau}^\star,\, \log \pi_{i,\tau}^\star \rangle\right) \right]$$

$$= \sum_{i \in V} \left[ (\pi_{i,\tau}^\star)^\top A_i(\pi - \pi_\tau^\star) + (\pi_i - \pi_{i,\tau}^\star)^\top A_i \pi_\tau^\star + \tau \mathsf{KL}\big(\pi_i \,\|\, \pi_{i,\tau}^\star\big) \right]$$

$$= \tau \mathsf{KL}\big(\pi \,\|\, \pi_\tau^\star\big), \tag{96}$$

where the penultimate step follows from (92) and the last step invokes Lemma 1.

- Moving to the last term, we have

$$u_{i,\tau}(\pi_{i,\tau}^\star, \pi_{-i,\tau}^\star) - u_{i,\tau}(\pi_i', \pi_{-i,\tau}^\star) = (\pi_{i,\tau}^\star - \pi_i')^\top A_i \pi_\tau^\star - \tau(\pi_{i,\tau}^\star)^\top \log \pi_{i,\tau}^\star + \tau(\pi_i')^\top \log \pi_i'$$

$$= \tau(\pi_{i,\tau}^\star - \pi_i')^\top \log \pi_{i,\tau}^\star - \tau(\pi_{i,\tau}^\star)^\top \log \pi_{i,\tau}^\star + \tau(\pi_i')^\top \log \pi_i'$$

$$= \tau \mathsf{KL}\big(\pi_i' \,\|\, \pi_{i,\tau}^\star\big). \tag{97}$$

where the second line follows again from (92).

Plugging (95), (96) and (97) into (94) gives

$$\sum_{i \in V} \left[ u_{i,\tau}(\pi_i', \pi_{-i}) - u_{i,\tau}(\pi_i, \pi_{-i}) \right] \le \tau \mathsf{KL}\big(\pi \,\|\, \pi_\tau^\star\big) + \frac{d_{\mathsf{max}}^2 \|A\|_\infty^2}{\tau} \mathsf{KL}\big(\pi_\tau^\star \,\|\, \pi\big).$$

Taking maximum over $\pi'$ finishes the proof.

### E.4 Proof of Lemma 4

Let $\widetilde{\pi}^{(T)} = \frac{1}{T+1} \sum_{t=0}^{T} \overline{\pi}^{(t+1)}$, then $\widetilde{\pi}^{(T)} \in \Delta(S)$. The proof is completed if we can show

$$\sum_{i \in V} \mathsf{Regret}_{i,\tau}(T+1) \ge \sum_{i \in V} \mathsf{Regret}_{i,\tau}\big(\widetilde{\pi}_i^{(T)}, T+1\big) \ge 0, \tag{98}$$

where the first inequality holds trivially since $\mathsf{Regret}_{i,\tau}(T+1) \ge \mathsf{Regret}_{i,\tau}\big(\widetilde{\pi}_i^{(T)}, T\big)$. It then boils down to show the second inequality of the above relation. From the definition of zero-sum polymatrix games, it holds that

$$\sum_{i \in V} \sum_{t=0}^{T} \left\langle \widetilde{\pi}_i^{(T)} - \overline{\pi}_i^{(t+1)},\, A_i \overline{\pi}^{(t+1)} \right\rangle = \sum_{i \in V} \sum_{t=0}^{T} \left\langle \widetilde{\pi}_i^{(T)},\, A_i \overline{\pi}^{(t+1)} \right\rangle$$

$$= \sum_{i \in V} \left\langle \widetilde{\pi}_i^{(T)}, A_i \sum_{t=0}^{T} \overline{\pi}^{(t+1)} \right\rangle$$

$$= (T+1) \sum_{i \in V} \left\langle \widetilde{\pi}_i^{(T)}, A_i \widetilde{\pi}^{(T)} \right\rangle = 0.$$

In addition, applying Jensen's inequality gives

$$\sum_{t=0}^{T} \mathcal{H}(\widetilde{\pi}_i^{(T)}) = (T+1)\mathcal{H}(\widetilde{\pi}_i^{(T)}) \geq \sum_{t=0}^{T} \mathcal{H}(\overline{\pi}_i^{(t+1)}).$$

Combining the above two relations yields

$$\sum_{i \in V} \mathsf{Regret}_{i,\tau}\left(\widetilde{\pi}_i^{(T)}, T+1\right) \geq \sum_{i \in V} \sum_{t=0}^{T} \left( \left\langle \widetilde{\pi}_i^{(T)} - \overline{\pi}_i^{(t+1)}, A_i \overline{\pi}^{(t+1)} \right\rangle + \tau \mathcal{H}(\widetilde{\pi}_i^{(T)}) - \tau \mathcal{H}(\overline{\pi}_i^{(t+1)}) \right) \geq 0,$$

which concludes the proof.

### E.5   PROOF OF LEMMA 5

Taking logarithm on the both sides of (7), we have

$$\log \pi_i^{(t+1)} \overset{\mathbf{1}}{=} (1 - \eta\tau) \log \pi_i^{(t)} + \eta A_i \overline{\pi}^{(t-\gamma+1)}. \tag{99}$$

On the other hand, the definition of QRE in (4) gives

$$\eta\tau \log \pi_{i,\tau}^{\star} \overset{\mathbf{1}}{=} \eta A_i \pi_{\tau}^{\star}.$$

Subtracting the two equalities and taking inner product with $\overline{\pi}_i^{(t-\gamma+1)} - \pi_{i,\tau}^{\star}$, we get

$$\left\langle \log \pi_i^{(t+1)} - (1 - \eta\tau) \log \pi_i^{(t)} - \eta\tau \log \pi_{i,\tau}^{\star}, \overline{\pi}_i^{(t-\gamma+1)} - \pi_{i,\tau}^{\star} \right\rangle$$
$$= \eta \left( \overline{\pi}_i^{(t-\gamma+1)} - \pi_{i,\tau}^{\star} \right)^{\top} A_i \left( \overline{\pi}^{(t-\gamma+1)} - \pi_{\tau}^{\star} \right).$$

Summing the above equality over $i \in V$ leads to

$$\left\langle \log \pi^{(t+1)} - (1 - \eta\tau) \log \pi^{(t)} - \eta\tau \log \pi_{\tau}^{\star}, \overline{\pi}_i^{(t-\gamma+1)} - \pi_{\tau}^{\star} \right\rangle$$
$$= \eta \sum_{i \in V} \left( \overline{\pi}_i^{(t-\gamma+1)} - \pi_{i,\tau}^{\star} \right)^{\top} A_i \left( \overline{\pi}^{(t-\gamma+1)} - \pi_{\tau}^{\star} \right) = 0,$$

where the final step results from Lemma 1.

### E.6   PROOF OF LEMMA 6

Recall from (71) that

$$\mathsf{KL}\left(\pi_{\tau}^{\star} \,\|\, \pi^{(t+1)}\right) = (1 - \eta\tau)\mathsf{KL}\left(\pi_{\tau}^{\star} \,\|\, \pi^{(t)}\right) - (1 - \eta\tau)\mathsf{KL}\left(\overline{\pi}^{(t-\gamma+1)} \,\|\, \pi^{(t)}\right) - \mathsf{KL}\left(\pi^{(t+1)} \,\|\, \overline{\pi}^{(t-\gamma+1)}\right)$$
$$+ \left\langle \log \overline{\pi}^{(t-\gamma+1)} - \log \pi^{(t+1)}, \overline{\pi}^{(t-\gamma+1)} - \pi^{(t+1)} \right\rangle - \eta\tau \mathsf{KL}\left(\overline{\pi}^{(t-\gamma+1)} \,\|\, \pi_{\tau}^{\star}\right). \tag{100}$$

When $t < \gamma$, we have $\overline{\pi}_i^{(t-\gamma+1)} = \pi^{(0)}$. It follows that

$$\log \overline{\pi}_i^{(t-\gamma+1)} = \log \pi^{(0)} \overset{\mathbf{1}}{=} 0,$$

and that

$$\log \pi_i^{(t+1)} \overset{\mathbf{1}}{=} (1 - \eta\tau)^{t+1} \log \pi^{(0)} + \eta \sum_{l=0}^{t} (1 - \eta\tau)^l A_i \overline{\pi}^{(t-\gamma-l+1)}$$

$$\overset{\mathbf{1}}{=} \eta \sum_{l=0}^{t} (1 - \eta\tau)^l A_i \pi^{(0)}.$$

Therefore, we can bound the term $\left\langle \log \overline{\pi}^{(t-\gamma+1)} - \log \pi^{(t+1)}, \, \overline{\pi}^{(t-\gamma+1)} - \pi^{(t+1)} \right\rangle$ as

$$
\left\langle \log \overline{\pi}^{(t-\gamma+1)} - \log \pi^{(t+1)}, \, \overline{\pi}^{(t-\gamma+1)} - \pi^{(t+1)} \right\rangle = \left\langle \eta \sum_{l=0}^{t} (1 - \eta\tau)^l A_i \pi^{(0)}, \, \pi^{(0)} - \pi^{(t+1)} \right\rangle
$$
$$
\leq \eta(t+1) d_{\mathsf{max}} \|A\|_\infty \left\| \pi^{(0)} - \pi^{(t+1)} \right\|_1
$$
$$
\leq 2\eta(t+1) d_{\mathsf{max}} \|A\|_\infty. \tag{101}
$$

Plugging the above inequality into (100) leads to

$$
\mathsf{KL}\big(\pi_\tau^\star \,\|\, \pi^{(t+1)}\big) \leq (1 - \eta\tau)\mathsf{KL}\big(\pi_\tau^\star \,\|\, \pi^{(t)}\big) - (1 - \eta\tau)\mathsf{KL}\big(\overline{\pi}^{(t-\gamma+1)} \,\|\, \pi^{(t)}\big) - \mathsf{KL}\big(\pi^{(t+1)} \,\|\, \overline{\pi}^{(t-\gamma+1)}\big)
$$
$$
+ 2\eta(t+1) d_{\mathsf{max}} \|A\|_\infty.
$$

Applying the above inequality recursively to the iterates $0, 1, \ldots, \gamma - 1$, we arrive at

$$
\mathsf{KL}\big(\pi_\tau^\star \,\|\, \pi^{(\gamma)}\big)
$$
$$
\leq (1 - \eta\tau)^\gamma \mathsf{KL}\big(\pi_\tau^\star \,\|\, \pi^{(0)}\big) - \sum_{l_1=0}^{\gamma-1} (1 - \eta\tau)^{\gamma-1-l_1} \Big[ (1 - \eta\tau)\mathsf{KL}\big(\overline{\pi}^{(l_1-\gamma+1)} \,\|\, \pi^{(l_1)}\big) + \mathsf{KL}\big(\pi^{(l_1+1)} \,\|\, \overline{\pi}^{(l_1-\gamma+1)}\big) \Big]
$$
$$
+ 2\eta \sum_{l_1=0}^{\gamma-1} (1 - \eta\tau)^{\gamma-1-l_1} (l_1 + 1) d_{\mathsf{max}} \|A\|_\infty
$$
$$
\leq (1 - \eta\tau)^\gamma \mathsf{KL}\big(\pi_\tau^\star \,\|\, \pi^{(0)}\big) - \sum_{l_1=0}^{\gamma-1} (1 - \eta\tau)^{\gamma-1-l_1} \Big[ (1 - \eta\tau)\mathsf{KL}\big(\overline{\pi}^{(l_1-\gamma+1)} \,\|\, \pi^{(l_1)}\big) + \mathsf{KL}\big(\pi^{(l_1+1)} \,\|\, \overline{\pi}^{(l_1-\gamma+1)}\big) \Big]
$$
$$
+ 2\eta\gamma^2 d_{\mathsf{max}} \|A\|_\infty.
$$

### E.7 PROOF OF LEMMA 7

Taking logarithm on the both sides of (77) and (78), we get

$$
\eta\big(\log \widetilde{\pi}_i^{(t)} - \log \pi_i^{(t-1)}\big) \overset{\mathbf{1}}{=} \widetilde{\eta}_i^{(t)}\big(\log \pi_i^{(t)} - \log \pi_i^{(t-1)}\big),
$$

or equivalently

$$
\log \pi_i^{(t)} \overset{\mathbf{1}}{=} \frac{\eta}{\widetilde{\eta}_i^{(t)}} \log \widetilde{\pi}_i^{(t)} + \Big(1 - \frac{\eta}{\widetilde{\eta}_i^{(t)}}\Big) \log \pi_i^{(t-1)}.
$$

Taking inner product with $\pi_{i,\tau}^\star - \pi_i^{(t)}$,

$$
\Big\langle \log \pi_i^{(t)} - \frac{\eta}{\widetilde{\eta}_i^{(t)}} \log \widetilde{\pi}_i^{(t)} - \Big(1 - \frac{\eta}{\widetilde{\eta}_i^{(t)}}\Big) \log \pi_i^{(t-1)}, \, \pi_{i,\tau}^\star - \pi_i^{(t)} \Big\rangle = 0.
$$

By definition of KL divergence, we have

$$
\Big\langle \log \pi_i^{(t)} - \frac{\eta}{\widetilde{\eta}_i^{(t)}} \log \widetilde{\pi}_i^{(t)} - \Big(1 - \frac{\eta}{\widetilde{\eta}_i^{(t)}}\Big) \log \pi_i^{(t-1)}, \, \pi_{i,\tau}^\star \Big\rangle
$$
$$
= \Big\langle (\log \pi_i^{(t)} - \log \pi_{i,\tau}^\star) - \frac{\eta}{\widetilde{\eta}_i^{(t)}} (\log \widetilde{\pi}_i^{(t)} - \log \pi_{i,\tau}^\star) - \Big(1 - \frac{\eta}{\widetilde{\eta}_i^{(t)}}\Big)(\log \pi_i^{(t-1)} - \log \pi_{i,\tau}^\star), \, \pi_{i,\tau}^\star \Big\rangle
$$
$$
= -\mathsf{KL}\big(\pi_{i,\tau}^\star \,\|\, \pi_i^{(t)}\big) + \Big(1 - \frac{\eta}{\widetilde{\eta}_i^{(t)}}\Big)\mathsf{KL}\big(\pi_{i,\tau}^\star \,\|\, \pi_i^{(t-1)}\big) + \frac{\eta}{\widetilde{\eta}_i^{(t)}}\mathsf{KL}\big(\pi_{i,\tau}^\star \,\|\, \widetilde{\pi}_i^{(t)}\big),
$$

and

$$
\Big\langle \log \pi_i^{(t)} - \frac{\eta}{\widetilde{\eta}_i^{(t)}} \log \widetilde{\pi}_i^{(t)} - \Big(1 - \frac{\eta}{\widetilde{\eta}_i^{(t)}}\Big) \log \pi_i^{(t-1)}, \, \pi_i^{(t)} \Big\rangle
$$

$$= \frac{\eta}{\widetilde{\eta}_i^{(t)}} \mathsf{KL}\big(\pi_i^{(t)} \,\|\, \widetilde{\pi}_i^{(t)}\big) + \Big(1 - \frac{\eta}{\widetilde{\eta}_i^{(t)}}\Big) \mathsf{KL}\big(\pi_i^{(t)} \,\|\, \pi_i^{(t-1)}\big).$$

Taken together, we get

$$\mathsf{KL}\big(\pi_{i,\tau}^{\star} \,\|\, \pi_i^{(t)}\big) + \frac{\eta}{\widetilde{\eta}_i^{(t)}} \mathsf{KL}\big(\pi_i^{(t)} \,\|\, \widetilde{\pi}_i^{(t)}\big) + \Big(1 - \frac{\eta}{\widetilde{\eta}_i^{(t)}}\Big) \mathsf{KL}\big(\pi_i^{(t)} \,\|\, \pi_i^{(t-1)}\big)$$

$$= \Big(1 - \frac{\eta}{\widetilde{\eta}_i^{(t)}}\Big) \mathsf{KL}\big(\pi_{i,\tau}^{\star} \,\|\, \pi_i^{(t-1)}\big) + \frac{\eta}{\widetilde{\eta}_i^{(t)}} \mathsf{KL}\big(\pi_{i,\tau}^{\star} \,\|\, \widetilde{\pi}_i^{(t)}\big). \tag{102}$$

On the other hand, taking logarithm of (78) and making inner product with $\overline{\pi}_i^{(\kappa_i^{(t)})} - \pi_{i,\tau}^{\star}$ gives

$$\big\langle \log \widetilde{\pi}_i^{(t)} - (1 - \widetilde{\eta}_i^{(t)}\tau) \log \pi_i^{(t-1)} - \widetilde{\eta}_i^{(t)}\tau \log \pi_{i,\tau}^{\star}, \, \overline{\pi}_i^{(\kappa_i^{(t)})} - \pi_{i,\tau}^{\star} \big\rangle$$

$$= \widetilde{\eta}_i^{(t)} (\overline{\pi}_i^{(\kappa_i^{(t)})} - \pi_{i,\tau}^{\star})^{\top} A_i (\overline{\pi}^{(\kappa_i^{(t)})} - \pi_{\tau}^{\star}).$$

Following a similar discussion in (19) gives

$$\mathsf{KL}\big(\pi_{i,\tau}^{\star} \,\|\, \widetilde{\pi}_i^{(t)}\big) = (1 - \widetilde{\eta}_i^{(t)}\tau) \mathsf{KL}\big(\pi_{i,\tau}^{\star} \,\|\, \pi_i^{(t-1)}\big) - (1 - \widetilde{\eta}_i^{(t)}\tau) \mathsf{KL}\big(\overline{\pi}_i^{(\kappa_i^{(t)})} \,\|\, \pi_i^{(t-1)}\big)$$

$$- \widetilde{\eta}_i^{(t)}\tau \mathsf{KL}\big(\overline{\pi}_i^{(\kappa_i^{(t)})} \,\|\, \pi_{i,\tau}^{\star}\big) - \mathsf{KL}\big(\widetilde{\pi}_i^{(t)} \,\|\, \overline{\pi}_i^{(\kappa_i^{(t)})}\big)$$

$$+ \big\langle \log \overline{\pi}_i^{(\kappa_i^{(t)})} - \log \widetilde{\pi}_i^{(t)}, \, \overline{\pi}_i^{(\kappa_i^{(t)})} - \widetilde{\pi}_i^{(t)} \big\rangle - \widetilde{\eta}_i^{(t)} (\overline{\pi}_i^{(\kappa_i^{(t)})} - \pi_{i,\tau}^{\star})^{\top} A_i (\overline{\pi}^{(\kappa_i^{(t)})} - \pi_{\tau}^{\star}). \tag{103}$$

Plugging the above equation into (102),

$$\mathsf{KL}\big(\pi_{i,\tau}^{\star} \,\|\, \pi_i^{(t)}\big) + \frac{\eta}{\widetilde{\eta}_i^{(t)}} \mathsf{KL}\big(\pi_i^{(t)} \,\|\, \widetilde{\pi}_i^{(t)}\big) + \Big(1 - \frac{\eta}{\widetilde{\eta}_i^{(t)}}\Big) \mathsf{KL}\big(\pi_i^{(t)} \,\|\, \pi_i^{(t-1)}\big)$$

$$= (1 - \eta\tau) \mathsf{KL}\big(\pi_{i,\tau}^{\star} \,\|\, \pi_i^{(t-1)}\big) - \eta(\overline{\pi}_i^{(\kappa_i^{(t)})} - \pi_{i,\tau}^{\star})^{\top} A_i (\overline{\pi}^{(\kappa_i^{(t)})} - \pi_{\tau}^{\star})$$

$$- \frac{\eta}{\widetilde{\eta}_i^{(t)}} \left[ (1 - \widetilde{\eta}_i^{(t)}\tau) \mathsf{KL}\big(\overline{\pi}_i^{(\kappa_i^{(t)})} \,\|\, \pi_i^{(t-1)}\big) + \widetilde{\eta}_i^{(t)}\tau \mathsf{KL}\big(\overline{\pi}_i^{(\kappa_i^{(t)})} \,\|\, \pi_{i,\tau}^{\star}\big) + \mathsf{KL}\big(\widetilde{\pi}_i^{(t)} \,\|\, \overline{\pi}_i^{(\kappa_i^{(t)})}\big) \right]$$

$$+ \frac{\eta}{\widetilde{\eta}_i^{(t)}} \big\langle \log \overline{\pi}_i^{(\kappa_i^{(t)})} - \log \widetilde{\pi}_i^{(t)}, \, \overline{\pi}_i^{(\kappa_i^{(t)})} - \widetilde{\pi}_i^{(t)} \big\rangle.$$

Rearranging the terms finishes the proof.

### E.8 PROOF OF LEMMA 8

For notational convenience, we set

$$\phi_i^{(t)} = \Big(1 - \frac{\eta}{\widetilde{\eta}_i^{(t)}}\Big) \big\| \pi_i^{(t)} - \pi_i^{(t-1)} \big\|_1$$

$$+ \frac{\eta}{\widetilde{\eta}_i^{(t)}} \Big( \big\| \overline{\pi}_i^{(\kappa_i^{(t)})} - \pi_i^{(t-1)} \big\|_1 + \big\| \widetilde{\pi}_i^{(t)} - \overline{\pi}_i^{(\kappa_i^{(t)})} \big\|_1 + \big\| \pi_i^{(t)} - \widetilde{\pi}_i^{(t)} \big\|_1 \Big)$$

for all $i \in V, t \geq 0$. By triangular inequality, we have $\phi_i^{(t)} \geq \big\| \pi_i^{(t)} - \pi_i^{(t-1)} \big\|_1$. In addition, we denote by $t_1 \wedge t_2 := \min\{t_1, t_2\}$ and $t_1 \vee t_2 := \max\{t_1, t_2\}$. For $0 < t_1 < t_2$, it holds that

$$\big\| \overline{\pi}_j^{(\kappa_i^{(t_1)})} - \overline{\pi}_j^{(\kappa_i^{(t_2)})} \big\|_1$$

$$\leq \big\| \pi_j^{(\nu_j(\kappa_i^{(t_1)}))} - \pi_j^{(\nu_j(\kappa_i^{(t_2)}))} \big\|_1 + \big\| \overline{\pi}_j^{(\kappa_i^{(t_1)})} - \pi_j^{(\nu_j(\kappa_i^{(t_1)}))} \big\|_1 + \big\| \overline{\pi}_j^{(\kappa_i^{(t_2)})} - \pi_j^{(\nu_j(\kappa_i^{(t_2)}))} \big\|_1$$

$$\leq \sum_{l=(\nu_j(\kappa_i^{(t_1)})+1) \wedge (\nu_j(\kappa_i^{(t_2)})+1)}^{\nu_j(\kappa_i^{(t_1)}) \vee \nu_j(\kappa_i^{(t_2)})} \big\| \pi_j^{(l)} - \pi_j^{(l-1)} \big\|_1 + \big\| \overline{\pi}_j^{(\kappa_i^{(t_1)})} - \widetilde{\pi}_j^{(\nu_j(\kappa_i^{(t_1)}))} \big\|_1 + \big\| \widetilde{\pi}_j^{(\nu_j(\kappa_i^{(t_1)}))} - \pi_j^{(\nu_j(\kappa_i^{(t_1)}))} \big\|_1$$

$$+ \left\|\overline{\pi}_j^{(\kappa_i^{(t_2)})} - \widetilde{\pi}_j^{(\nu_j(\kappa_i^{(t_2)}))}\right\|_1 + \left\|\widetilde{\pi}_j^{(\nu_j(\kappa_i^{(t_2)}))} - \pi_j^{(\nu_j(\kappa_i^{(t_2)}))}\right\|_1$$

$$\leq \sum_{l=(\nu_j(\kappa_i^{(t_1)})+1)\wedge(\nu_j(\kappa_i^{(t_2)})+1)}^{\nu_j(\kappa_i^{(t_1)})\vee\nu_j(\kappa_i^{(t_2)})} \phi_j^{(l)} + \frac{\widetilde{\eta}_j^{(\nu_j(\kappa_i^{(t_1)}))}}{\eta}\phi_j^{(\nu_j(\kappa_i^{(t_1)}))} + \frac{\widetilde{\eta}_j^{(\nu_j(\kappa_i^{(t_2)}))}}{\eta}\phi_j^{(\nu_j(\kappa_i^{(t_2)}))}$$

$$\leq \sum_{l=(\nu_j(\kappa_i^{(t_1)})+1)\wedge(\nu_j(\kappa_i^{(t_2)})+1)}^{\nu_j(\kappa_i^{(t_1)})\vee\nu_j(\kappa_i^{(t_2)})} \phi_j^{(l)} + (\gamma+1)\phi_j^{(\nu_j(\kappa_i^{(t_1)}))} + (\gamma+1)\phi_j^{(\nu_j(\kappa_i^{(t_2)}))}. \tag{104}$$

Therefore, we have

$$\sum_{k=\kappa_i^{(t)}}^{t} \left\|\overline{\pi}_j^{(\kappa_i^{(k)})} - \overline{\pi}_j^{(\kappa_i^{(\kappa_i^{(t-1)})})}\right\|_1$$

$$\leq \sum_{k=\kappa_i^{(t)}}^{t} \left\{ \sum_{l=(\nu_j(\kappa_i^{(\kappa_i^{(t-1)})})+1)\wedge(\nu_j(\kappa_i^{(k)})+1)}^{\nu_j(\kappa_i^{(\kappa_i^{(t-1)})})\vee\nu_j(\kappa_i^{(k)})} \phi_j^{(l)} + (\gamma+1)\phi_j^{(\nu_j(\kappa_i^{(\kappa_i^{(t-1)})}))} + (\gamma+1)\phi_j^{(\nu_j(\kappa_i^{(k)}))} \right\}. \tag{105}$$

Since $0 \vee (t - \gamma) \leq \kappa_i^{(t)} \leq t \leq \nu_i(t) \leq t + \gamma$ for all $i \in V$, $t \geq 0$, the first term can be bounded by

$$\sum_{l=(\nu_j(\kappa_i^{(\kappa_i^{(t-1)})})+1)\wedge(\nu_j(\kappa_i^{(k)})+1)}^{\nu_j(\kappa_i^{(\kappa_i^{(t-1)})})\vee\nu_j(\kappa_i^{(k)})} \phi_j^{(l)} \leq \sum_{l=(t-2\gamma)\wedge(k-\gamma+1)}^{(t+\gamma-1)\vee(k+\gamma)} \phi_j^{(l)} \leq \sum_{l=t-2\gamma}^{t+\gamma} \phi_j^{(l)}.$$

In addition, the mapping $k \mapsto \nu_j(\kappa_i^{(k)})$ is injective when $k \geq \gamma$ (cf. Assumption 2 and 3). It follows that

$$\sum_{k=\kappa_i^{(t)}}^{t} \phi_j^{(\nu_j(\kappa_i^{(k)}))} \leq \sum_{l=\kappa_i^{(t)}-\gamma}^{t+\gamma} \phi_j^{(l)} \leq \sum_{l=t-2\gamma}^{t+\gamma} \phi_j^{(l)}$$

Plugging the above inequalities into (105) yields

$$\sum_{k=\kappa_i^{(t)}}^{t} \left\|\overline{\pi}_j^{(\kappa_i^{(k)})} - \overline{\pi}_j^{(\kappa_i^{(\kappa_i^{(t-1)})})}\right\|_1$$

$$\leq (t+1-\kappa_i^{(t)}) \sum_{l=t-2\gamma}^{t+\gamma} \phi_j^{(l)} + (t+1-\kappa_i^{(t)})(\gamma+1)\phi_j^{(\nu_j(\kappa_i^{(\kappa_i^{(t-1)})}))} + (\gamma+1)\sum_{l=t-2\gamma}^{t+\gamma} \phi_j^{(l)}$$

$$\leq 2(\gamma+1) \sum_{l=t-2\gamma}^{t+\gamma} \phi_j^{(l)} + (\gamma+1)^2\phi_j^{(\nu_j(\kappa_i^{(\kappa_i^{(t-1)})}))}.$$

Finally, we control the term $\phi_i^{(t)}$ with $\psi_i^{(t)}$ as:

$$(\phi_i^{(t)})^2 = \left( \left(1 - \frac{\eta}{\widetilde{\eta}_i^{(t)}}\right)^{1/2} \cdot \left(1 - \frac{\eta}{\widetilde{\eta}_i^{(t)}}\right)^{1/2} \left\|\pi_i^{(t)} - \pi_i^{(t-1)}\right\|_1 \right.$$

$$+ \left(\frac{\eta}{\widetilde{\eta}_i^{(t)}}(1 - \widetilde{\eta}_i^{(t)}\tau)^{-1}\right)^{1/2} \cdot \left(\frac{\eta}{\widetilde{\eta}_i^{(t)}}(1 - \widetilde{\eta}_i^{(t)}\tau)\right)^{1/2} \left\|\overline{\pi}_i^{(\kappa_i^{(t)})} - \pi_i^{(t-1)}\right\|_1$$

$$\left. + \left(\frac{\eta}{\widetilde{\eta}_i^{(t)}}\right)^{1/2} \cdot \left(\frac{\eta}{\widetilde{\eta}_i^{(t)}}\right)^{1/2} \left(\left\|\widetilde{\pi}_i^{(t)} - \overline{\pi}_i^{(\kappa_i^{(t)})}\right\|_1 + \left\|\pi_i^{(t)} - \widetilde{\pi}_i^{(t)}\right\|_1\right)\right)^2$$

$$\overset{(i)}{\leq} \left(1 - \frac{\eta}{\widetilde{\eta}_i^{(t)}} + \frac{\eta}{\widetilde{\eta}_i^{(t)}}\left(2 + (1 - \widetilde{\eta}_i^{(t)}\tau)^{-1}\right)\right)\left[\left(1 - \frac{\eta}{\widetilde{\eta}_i^{(t)}}\right)\left\|\pi_i^{(t)} - \pi_i^{(t-1)}\right\|_1^2\right.$$

$$+ \frac{\eta}{\widetilde{\eta}_i^{(t)}} \Big( (1 - \widetilde{\eta}_i^{(t)}\tau) \big\|\overline{\pi}_i^{(\kappa_i^{(t)})} - \pi_i^{(t-1)}\big\|_1^2 + \big\|\widetilde{\pi}_i^{(t)} - \overline{\pi}_i^{(\kappa_i^{(t)})}\big\|_1^2 + \big\|\pi_i^{(t)} - \widetilde{\pi}_i^{(t)}\big\|_1^2 \Big) \Big]$$

$$\overset{(ii)}{\leq} 2 \left( 2 + (1 - \widetilde{\eta}_i^{(t)}\tau)^{-1} \right) \left[ \left( 1 - \frac{\eta}{\widetilde{\eta}_i^{(t)}} \right) \mathsf{KL}\big(\pi_i^{(t)} \,\|\, \pi_i^{(t-1)}\big) \right.$$

$$\left. + \frac{\eta}{\widetilde{\eta}_i^{(t)}} \Big( (1 - \widetilde{\eta}_i^{(t)}\tau)\mathsf{KL}\big(\overline{\pi}_i^{(\kappa_i^{(t)})} \,\|\, \pi_i^{(t-1)}\big) + \mathsf{KL}\big(\widetilde{\pi}_i^{(t)} \,\|\, \overline{\pi}_i^{(\kappa_i^{(t)})}\big) + \mathsf{KL}\big(\pi_i^{(t)} \,\|\, \widetilde{\pi}_i^{(t)}\big) \Big) \right]$$

$$\overset{(iii)}{\leq} 8\psi_i^{(t)}, \tag{106}$$

where (i) applies Cauchy-Schwarz inequality, (ii) invokes Pinsker's inequality and (iii) is due to $\widetilde{\eta}_i^{(t)}\tau \leq (\gamma+1)\eta\tau \leq 1/2$. Combining the above two inequalities finishes the proof.

### E.9 PROOF OF LEMMA 9

We start with verifying the claim (86). Recall that

$$\psi_i^{(t)} := \left( 1 - \frac{\eta}{\widetilde{\eta}_i^{(t)}} \right) \mathsf{KL}\big(\pi_i^{(t)} \,\|\, \pi_i^{(t-1)}\big)$$

$$+ \frac{\eta}{\widetilde{\eta}_i^{(t)}} \left[ (1 - \widetilde{\eta}_i^{(t)}\tau)\mathsf{KL}\big(\overline{\pi}_i^{(\kappa_i^{(t)})} \,\|\, \pi_i^{(t-1)}\big) + \mathsf{KL}\big(\widetilde{\pi}_i^{(t)} \,\|\, \overline{\pi}_i^{(\kappa_i^{(t)})}\big) + \mathsf{KL}\big(\pi_i^{(t)} \,\|\, \widetilde{\pi}_i^{(t)}\big) \right].$$

We introduce the following standard Lemma (see e.g., (Cen et al., 2020, Appendix A.2)), which allows us to bound control $\mathsf{KL}\big(\pi_i \,\|\, \pi_i'\big)$ properly:

**Lemma 11.** *Given $\pi_i, \pi_i' \in \Delta(S_i)$ and $w \in \mathbb{R}^{|S_i|}$ with $\log \pi_i \overset{\mathbf{1}}{=} \log \pi_i' + w$, we have*

$$\mathsf{KL}\big(\pi_i \,\|\, \pi_i'\big) \leq \big\|\log \pi_i - \log \pi_i'\big\|_\infty \leq 2\big\|w\big\|_\infty.$$

Therefore, it suffices to figure out the terms $\log \pi_i^{(t)} - \log \pi_i^{(t-1)}$, $\log \pi_i^{(t)} - \log \widetilde{\pi}_i^{(t)}$, $\log \widetilde{\pi}_i^{(t)} - \log \overline{\pi}_i^{(\kappa_i^{(t)})}$ and $\log \overline{\pi}_i^{(\kappa_i^{(t)})} - \log \pi_i^{(t-1)}$.

- **Bounding** $\mathsf{KL}\big(\pi_i^{(t)} \,\|\, \pi_i^{(t-1)}\big)$ **and** $\mathsf{KL}\big(\pi_i^{(t)} \,\|\, \widetilde{\pi}_i^{(t)}\big)$. The following equations follow directly from (77) and (78):

$$\begin{cases} \log \pi_i^{(t)} - \log \pi_i^{(t-1)} \overset{\mathbf{1}}{=} \eta([A_i\overline{\pi}^{(\kappa_i^{(t)})}]_k - \tau \log \pi_i^{(t-1)}) \\ \log \pi_i^{(t)} - \log \widetilde{\pi}_i^{(t)} \overset{\mathbf{1}}{=} (\eta - \widetilde{\eta}_i^{(t)})([A_i\overline{\pi}^{(\kappa_i^{(t)})}]_k - \tau \log \pi_i^{(t-1)}) \end{cases}. \tag{107}$$

In addition, we have the following bound w.r.t. the order of $\big\|\log \pi_i^{(t-1)}\big\|_\infty$, which we shall establish momentarily.

$$\big\|\tau \log \pi_i^{(t-1)}\big\|_\infty \leq \tau \log |S_i| + 2d_{\mathsf{max}}\|A\|_\infty. \tag{108}$$

This taken together with Lemma 11 yields

$$\begin{cases} \mathsf{KL}\big(\pi_i^{(t)} \,\|\, \pi_i^{(t-1)}\big) \leq \eta(3d_{\mathsf{max}}\|A\|_\infty + \tau \log |S_i|) \\ \mathsf{KL}\big(\pi_i^{(t)} \,\|\, \widetilde{\pi}_i^{(t)}\big) \leq (\widetilde{\eta}_i^{(t)} - \eta)(3d_{\mathsf{max}}\|A\|_\infty + \tau \log |S_i|) \end{cases}. \tag{109}$$

- **Bounding** $\mathsf{KL}\big(\widetilde{\pi}_i^{(t)} \,\|\, \overline{\pi}_i^{(\kappa_i^{(t)})}\big)$. When $\kappa_i^{(t)} \geq 1$, we recall from (80) that:

$$\log \overline{\pi}_i^{(\kappa_i^{(t)})} - \log \widetilde{\pi}_i^{(t)} \overset{\mathbf{1}}{=} \widetilde{\eta}_i^{(t)} \bigg( A_i(\overline{\pi}^{(\kappa_i^{(t)}-1)} - \overline{\pi}^{(\kappa_i^{(t)})})$$

$$+ \sum_{l=\kappa_i^{(t)}}^{t-1} (1 - \widetilde{\eta}_i^{(t)}\tau)(1-\eta\tau)^{t-1-l}A_i(\overline{\pi}^{(\kappa_i^{(t)}-1)} - \overline{\pi}^{(\kappa_i^{(l)})}) \bigg), \tag{110}$$

which leads to a crude bound

$$\mathsf{KL}\big(\overline{\pi}_i^{(\kappa_i^{(t)})} \,\|\, \widetilde{\pi}_i^{(t)}\big) \leq \widetilde{\eta}_i^{(t)} d_{\mathsf{max}} \|A\|_\infty \, (t - \kappa_i^{(t)} + 1) \leq \widetilde{\eta}_i^{(t)} d_{\mathsf{max}} \|A\|_\infty \, (\gamma + 1).$$

When $\kappa_i^{(t)} = 0$, we have

$$
\begin{aligned}
\log \overline{\pi}_i^{(\kappa_i^{(t)})} - \log \widetilde{\pi}_i^{(t)} &\overset{\mathbf{1}}{=} -\log \widetilde{\pi}_i^{(t)} \\
&\overset{\mathbf{1}}{=} -(1 - \widetilde{\eta}_i^{(t)}\tau)(1 - \eta\tau)^{t-1} \log \pi^{(0)} \\
&\quad - \widetilde{\eta}_i^{(t)} \left( A_i \overline{\pi}^{(\kappa_i^{(t)})} + \sum_{l=\kappa_i^{(t)}+1}^{t-1} (1 - \widetilde{\eta}_i^{(t)}\tau)(1 - \eta\tau)^{t-1-l} A_i \overline{\pi}^{(\kappa_i^{(l)})} \right) \\
&\overset{\mathbf{1}}{=} -\widetilde{\eta}_i^{(t)} \left( A_i \overline{\pi}^{(\kappa_i^{(t)})} + \sum_{l=\kappa_i^{(t)}+1}^{t-1} (1 - \widetilde{\eta}_i^{(t)}\tau)(1 - \eta\tau)^{t-1-l} A_i \overline{\pi}^{(\kappa_i^{(l)})} \right),
\end{aligned}
\tag{111}
$$

which yields

$$\mathsf{KL}\big(\overline{\pi}_i^{(\kappa_i^{(t)})} \,\|\, \widetilde{\pi}_i^{(t)}\big) \leq \widetilde{\eta}_i^{(t)} d_{\mathsf{max}} \|A\|_\infty \, t \leq \widetilde{\eta}_i^{(t)} d_{\mathsf{max}} \|A\|_\infty \, (\gamma + 1).$$

- **Bounding** $\mathsf{KL}\big(\overline{\pi}_i^{(\kappa_i^{(t)})} \,\|\, \pi_i^{(t-1)}\big)$. Note that

$$\log \overline{\pi}_i^{(\kappa_i^{(t)})} - \log \pi_i^{(t-1)} = (\log \overline{\pi}_i^{(\kappa_i^{(t)})} - \log \widetilde{\pi}_i^{(t)}) + (\log \widetilde{\pi}_i^{(t)} - \log \pi_i^{(t)}) + (\log \pi_i^{(t)} - \log \pi_i^{(t-1)}).$$

This yields, by equations (107), (110), (111) and associated bounds,

$$\mathsf{KL}\big(\overline{\pi}_i^{(\kappa_i^{(t)})} \,\|\, \pi_i^{(t-1)}\big) \leq \widetilde{\eta}_i^{(t)} d_{\mathsf{max}} \|A\|_\infty \, (\gamma + 1) + \widetilde{\eta}_i^{(t)} (3 d_{\mathsf{max}} \|A\|_\infty + \tau \log |S_i|).$$

Putting all pieces together, we conclude that

$$
\begin{aligned}
\psi_i^{(t)} &= \left( 1 - \frac{\eta}{\widetilde{\eta}_i^{(t)}} \right) \mathsf{KL}\big(\pi_i^{(t)} \,\|\, \pi_i^{(t-1)}\big) \\
&\quad + \frac{\eta}{\widetilde{\eta}_i^{(t)}} \left[ (1 - \widetilde{\eta}_i^{(t)}\tau) \mathsf{KL}\big(\overline{\pi}_i^{(\kappa_i^{(t)})} \,\|\, \pi_i^{(t-1)}\big) + \mathsf{KL}\big(\widetilde{\pi}_i^{(t)} \,\|\, \overline{\pi}_i^{(\kappa_i^{(t)})}\big) + \mathsf{KL}\big(\pi_i^{(t)} \,\|\, \widetilde{\pi}_i^{(t)}\big) \right] \\
&\leq 3\eta(3 d_{\mathsf{max}} \|A\|_\infty + \tau \log |S_i|) + 2\eta d_{\mathsf{max}} \|A\|_\infty \, (\gamma + 1) \\
&= \eta(d_{\mathsf{max}} \|A\|_\infty \, (2\gamma + 11) + 3\tau \log |S_i|).
\end{aligned}
$$

It remains to prove the claim (87):

$$
\begin{aligned}
\mathsf{KL}\big(\pi_{i,\tau}^\star \,\|\, \pi_i^{(2\gamma)}\big) &= \mathsf{KL}\big(\pi_{i,\tau}^\star \,\|\, \pi_i^{(0)}\big) + \big\langle \pi_{i,\tau}^\star, \, \log \pi_i^{(0)} - \log \pi_i^{(2\gamma)} \big\rangle \\
&\leq \mathsf{KL}\big(\pi_{i,\tau}^\star \,\|\, \pi_i^{(0)}\big) + \big\| \log \pi_i^{(0)} - \log \pi_i^{(2\gamma)} \big\|_\infty \\
&\leq \mathsf{KL}\big(\pi_{i,\tau}^\star \,\|\, \pi_i^{(0)}\big) + 2 \left\| \eta \sum_{l=1}^{2\gamma} (1 - \eta\tau)^{2\gamma-l} A_i \overline{\pi}^{(\kappa_i^{(l)})} \right\|_\infty \\
&\leq \mathsf{KL}\big(\pi_{i,\tau}^\star \,\|\, \pi_i^{(0)}\big) + 4\eta d_{\mathsf{max}} \|A\|_\infty \, \gamma,
\end{aligned}
$$

where the third step results from $\log \pi_i^{(2\gamma)} \overset{\mathbf{1}}{=} (1 - \eta\tau)^{2\gamma} \log \pi_i^{(0)} + \eta \sum_{l=1}^{2\gamma} (1 - \eta\tau)^{2\gamma-l} A_i \overline{\pi}^{(\kappa_i^{(l)})}$ and Lemma 11.

**Proof of the claim (108).** First, we prove by induction that for any $k, l \in S_i$,

$$\log \pi_i^{(t)}(k) - \log \pi_i^{(t)}(l) \leq \frac{2 d_{\mathsf{max}} \|A\|_\infty}{\tau}, \qquad \forall t \geq 0. \tag{112}$$

Note that the claim trivially holds for $t = 0$ with the uniform initialization $\pi_i^{(0)} = \frac{1}{|S_i|}\mathbf{1}$, $\forall i \in V$. Assume that (112) holds for all $t' \leq t - 1$. Note that $\log \pi_i^{(t)} \overset{\mathbf{1}}{=} (1 - \eta\tau) \log \pi_i^{(t-1)} + \eta A_i \overline{\pi}^{(\kappa_i^{(t)})}$, we have

$$\log \pi_i^{(t)}(k) - \log \pi_i^{(t)}(l) = (1 - \eta\tau) \left( \log \pi_i^{(t-1)}(k) - \log \pi_i^{(t-1)}(l) \right) + \eta \left( [A_i \overline{\pi}^{(\kappa_i^{(t)})}]_k - [A_i \overline{\pi}^{(\kappa_i^{(t)})}]_l \right)$$

$$\leq (1 - \eta\tau)\frac{2d_{\mathsf{max}} \|A\|_\infty}{\tau} + 2\eta d_{\mathsf{max}} \|A\|_\infty$$
$$= \frac{2d_{\mathsf{max}} \|A\|_\infty}{\tau},$$

where the second line follows from the induction hypothesis (112). This completes the induction at the $t$-th iteration. It follows that for all $i \in V$ and $t \geq 0$,

$$\log \pi_i^{(t)}(l) \geq \log \left( \max_{k \in S_i} \pi_i^{(t)}(k) \right) - \frac{2d_{\mathsf{max}} \|A\|_\infty}{\tau} \geq - \log |S_i| - \frac{2d_{\mathsf{max}} \|A\|_\infty}{\tau}. \tag{113}$$

