# OpenReview forum: "Asynchronous Gradient Play in Zero-Sum Multi-agent Games"
_ICLR.cc/2023/Conference — ICLR 2023 poster_

### Official Review · Reviewer_ukoW · 2022-10-24

**Confidence:** 2
**Correctness:** 4
**Technical Novelty And Significance:** 2
**Empirical Novelty And Significance:** 2
**Recommendation:** 6

**Clarity, Quality, Novelty And Reproducibility:**

Overall, well-written. There is some lack of comparison to prior works in a meaningful way to understand the novelty of the setting with delays.

**Strength And Weaknesses:**

+ interesting setting for 0-sum polymatrix games with delays
+ delays are studied in several contexts and so this paper extends prior knowledge but in the case of random or fixed delays, under some assumptions

weaknesses:
- lack of comparison to previous works without delays in terms of technical details and novelty: I am not sure how technically challenging this "delays" setting is, given all prior works that have significantly studied the problem. Can the authors provide a comparison or an insight for how their analysis significantly differs from previous analyses? What do the authors perceive as their main technical contribution?

**Summary Of The Paper:**

The paper studies zero-sum polymatrix games where there are several kinds of delayed feedbacks. One such type is random delay, and another type is fixed delay. The paper gives new analyses for single-timescale and two-timescale OMWU proving last iterate convergence for the former and finite-time convergence for the latter.

The main conttribution is to systematically study the impact of having delays, when each player plays according to the entropy-regularized OMWU. Formally, for the case of synchronous updates single-timescale OMWU needs 1/epsilon iterations to reach an epsilon-Nash, and they extend it to the asynchronous setting with required number of iterations 1/epsilon^2. Furthermore, this requires several mild assumptions on the delays (roughly the delays are not completely adversarial, but are random). Moreover, if delays are constant and known, the authors obtain a 1/\epsilong guarantee to \epsilon Nash by adopting two-timescale OMWU.

**Summary Of The Review:**

Overall, this paper builds upon prior works and studies NE of standard algorithms, with the twist that there are delays, either random or fixed. The algorithms have been used before, which is both good and bad. Good because the algorithms are natural, bad in terms of the novelty. In some sense, the paper's main message is that delays can be handled and nothing goes too wrong with them, even using prior algorithms. My main concern has to do with the technical aspects of the paper: I am not too convinced that this is not an incremental set of results, given the extensive analyses for the setting without the delays. Perhaps the authors can address this concern.

---

> ### Author Response · Authors · 2022-11-16
> **Response to Reviewer ukoW**
>
> We thank the reviewer for the useful feedback! We want to mention that while the delayed feedback setting has been extensively studied in the literature, most of them focus on achieving better sublinear individual guarantee. In the setting of multi-player zero-sum polymatrix games, this translates into the convergence of the average policy to the NE — the convergence of the actual policy in use, however, is not guaranteed (In fact, we are not aware of any existing work that has addressed the convergence issue with delayed feedback in a multi-player game setup). Therefore, our analysis distinguish itself from the majority of existing literature by studying the last-iterate convergence to the QRE/NE instead.
>
> Technically speaking, the closest existing work to ours is (Cen et al., 2021), which studied two-player zero-sum games without delays using single-timescale OMWU. We have generalized beyond the two-player setting by studying the zero-sum polymatrix game, and demonstrated how delay slows down the convergence of the single-timescale OMWU algorithm (due to the smaller range of learning rates), and devised a new two-timescale OMWU method to combat delay. Our analysis deviates from the work of (Cen et al., 2021) due to the presence of delayed feedback. In particular, one of the key error terms $\eta(\overline\pi_i^{(t+1)}-\pi_{i,\tau}^\star)A_i(\overline\pi^{(t+1)}-\pi_{\tau}^\star)$ (see proof of Lemma 2) cancel out when summing over $i\in V$ in the synchronous setting, as a consequence of zero-sum polymatrix game's special property. In the asynchronous setting, however, the corresponding term $\eta\sum_{i\in V}(\overline\pi_i^{(t+1)}-\pi_{i,\tau}^\star)A_i(\overline\pi^{(\kappa_i^{(t)})}-\pi_{\tau}^\star)$ misaligns in the time index of $\overline{\pi}$ and no longer admits straightforward treatment, leading to an additional error proportional to, intuitively speaking, $\eta^2\tau^{-1}\cdot \texttt{delay}$. This necessitates a fairly small choice of $\eta$ and hence slows down the convergence. The two-timescale method is motivated by this observation, which enables us to view $\overline\pi^{(\kappa_i^{(t)})}$ as an approximation of $\pi^{(t)}$ and replace the misaligned error terms with $\eta(\overline\pi_i^{(\kappa_i^{(t)})}-\pi_{i,\tau}^\star)A_i(\overline\pi^{(\kappa_i^{(t)})}-\pi_{\tau}^\star)$ — which makes cancellation once again possible and yields faster convergence. To the best of our knowledge, these technical treatments and observations are indeed novel, and are verified through numerical experiments.

---

> > ### Comment · Reviewer_ukoW · 2022-12-05
> > **ACK for rebuttal**
> >
> > Thank you for your response to my comments.

---

### Official Review · Reviewer_RvDc · 2022-10-24

**Confidence:** 3
**Correctness:** 2
**Technical Novelty And Significance:** 3
**Empirical Novelty And Significance:** 2
**Recommendation:** 6

**Clarity, Quality, Novelty And Reproducibility:**

Studying MWU algorithms in the asynchronous (delayed feedback) setting for polymatrix games is novel. Previous studies have been limited to either two-player games or synchronous (i.e., real-time) feedback.

The novelty in the algorithm is limited. It is unclear from the paper, whether the two-timescale algorithm is novel. I am not familiar enough with the area to extrapolate.

EDIT: After further discussion with the committee, author responses, my own study etc., I am more convinced of the novelty of the results in this paper. I am increasing my rating of this paper to above the acceptance threshold.

I felt that the motivation of the introduction and motivation of the paper could have been more clearly presented. The introduction talked about the general importance of understanding games, iteration dynamics, and asynchronicty, but I'm not sure why this *particular* problem is of interest.



**Details Of Ethics Concerns:**

No concerns.

**Strength And Weaknesses:**

Strengths:
(1) the problem seems fairly natural, and this is the first paper that is moving beyond the two-player regimes while studying asynchronous (delayed) dynamic.

(2) the introduction of the two-timescale algorithm that achieves better convergence rates under bounded adversarial delays adds some novelty to the paper.

Neither here, nor there
(1) the theorems are supported by some empirical experiments. On the other hand, the experiments seem to be on synthetic data that is created to meet the hypotheses of the theorem, rather than being experiments on real data.

Weaknesses:
(1) I feel the introduction and motivation of the paper could have been written better. While I don't directly work in this field, I am conversant in it, yet I was not able to identify *exactly* why this problem is of interest (beyond it being a, generally speaking, "natural question"). Even starting at the second sentence "While conventional wisdom leans towards the paradigm of centralized learning," I did not understand what this 'conventional wisdom' was or where it was coming from.

(2) The main technical body of the paper is mostly a listing of theorems. I did not find any proofs or intuitions about how the theorems are proved; and I do not know how technically difficult or interesting the theorems are. On the positive side, sometimes these theorems are followed by some examples of scenarios that meet the hypotheses of the theorems. However, at the end of the paper, I am not entirely sure where these results are relevant and whether they are difficult to obtain. (Once again, I don't work specifically in this area, but I have some familiarity with the general area of TML+economics and computation and these flavors of results.)


**Summary Of The Paper:**

This paper designs variants of the Multiplicative Weights Update algorithm to find epsilon-approximate equilibria (Nash and Quantal response) for zero-sum (multiplayer) polymatrix games. The main algorithm is called OMWU (optimistic multiplicative weight updates) and comes in two variants: single-timescale and two-timescale. The analysis shows that single-timescale OMWU converges linearly if feedback is instantaneous or delays come from a distribution that vanishes exponentially. two-timescale OMWU achieves last-iterate eps-Nash convergence and average-iterate eps-QRE convergence when there are bounded adversarial delays. Convergence rates are generally linear.

**Summary Of The Review:**

The paper starts the journey towards understanding asynchronous MWU convergence for multiplayer games. The general direction seems promising. I am not sure how compelling the motivation is though, and it is hard to evaluate how difficult the results are to obtain / how interesting they are technically in the context of previous results.

The "Correctness" question below is hard to answer since there are no proofs in the paper. There are some experiments that seem to support the claims however. The theorems may well be true, I just don't see any proofs in the paper.

---

> ### Author Response · Authors · 2022-11-16
> **Response to Reviewer RvDc**
>
> We thank the reviewer for the useful feedback! We want to start by mentioning that due to page limit, the proofs are provided in the supplemental material of the submission. We understand the reviewer might have missed it, and now have included in the pdf of the paper directly in the revision. We have also revised the paper significantly to better motivate the algorithm designs and sharpened some of our analysis.
>
> >I feel the introduction and motivation of the paper could have been written better. While I don't directly work in this field, I am conversant in it, yet I was not able to identify exactly why this problem is of interest (beyond it being a, generally speaking, "natural question").
>
> Finding Nash equilibrium lies in the center of game theory. However, previous literatures mostly focused on two-player games, or they considered the scenarios where the feedback were obtained immediately after making decisions. Due to communication slowdowns and congestions, delayed feedback is omnipresent in real-life scenarios (see also our response to reviewer VVFX). The untimely information can significantly influence the actions of the players and impair the utilities. Thus it is quite interesting to devise algorithms against the adversarial delays.
>
> > Even starting at the second sentence "While conventional wisdom leans towards the paradigm of centralized learning," I did not understand what this 'conventional wisdom' was or where it was coming from.
>
> Centralized learning refers to the learning paradigm where a central server that collects all players' payoff matrices and makes decisions for each player, which is impossible as the number of players grow to thousands or even to millions and with supervision of user protection legacy. The LP formulation for solving NE in previous literatures was one of such example. In contrast, independent learning algorithms, where players aim to learn in a decentralized and symmetric manner, are relatively much less studied. See for example https://arxiv.org/abs/2111.11743 for some further background discussion.
>
> > The novelty in the algorithm is limited. It is unclear from the paper, whether the two-timescale algorithm is novel. I am not familiar enough with the area to extrapolate.
>
> Thanks for the comment! While the idea of adopting two-timescale update rules has been explored in the literature, these efforts, however, are not applicable to the setting considered in this work and fall short of providing provable convergence (In fact, we are not aware of any prior work that has addressed the convergence issue with delayed feedback in a multi-agent game setup).
>
> In particular, \(Fasoulakis et al., 2022\) focus on the two-player zero-sum games under the synchronous setting, while \(Hsieh et al., 2020\) is restricted to (i) the online-learning setting, (ii) constant delay and (iii) Euclidean-based update rules (i.e., the updates are additive rather than multiplicative). We would like to mention that the specific choice of learning rate ratio $\overline{\eta}/\eta$ in this work, as a key algorithmic design in two-timescale methods, stems from our novel analysis and is verified by numerical simulations. For more discussion on technical novelty, please check our response to reviewer ukoW.

---

> > ### Comment · Reviewer_RvDc · 2022-11-18
> > **Response**
> >
> > Thank you to the authors for addressing many of the points in the review.
> >
> > I am glad to have gotten more detail on the novelty of the algorithms presented here.
> >
> > Wherever this paper ends up getting published, I think it would be helpful for the authors to include all the material in this (and other responses) that motivate these PARTICULAR types of games and why their study is important in the introduction.
> >
> > An unaddressed concern from my review remains:
> > (*) The main technical body of the paper is mostly a listing of theorems. I did not find any proofs or intuitions about how the theorems are proved; and I do not know how technically difficult or interesting the theorems are. On the positive side, sometimes these theorems are followed by some examples of scenarios that meet the hypotheses of the theorems. However, at the end of the paper, I am not entirely sure where these results are relevant and whether they are difficult to obtain. (Once again, I don't work specifically in this area, but I have some familiarity with the general area of TML+economics and computation and these flavors of results.)

---

> > > ### Author Response · Authors · 2022-11-20
> > > **Response to Reviewer RvDc**
> > >
> > > Thanks for reading our response and we are happy you have it helpful to clarify some of your concerns. Regarding the proof, it was originally included in the supplemental materials and now we have also included in the revised paper immediately after the main paper. Due to page limits, we have to defer all the analysis in the appendix, while only keep the theorems and corresponding discussions in the main paper. Please allow us to further elaborate the relevance and technical novelty of our results.
> > > - Relevance of our results:  Delayed feedback are commonly encountered in real-world applications where the players only receive the payoff information sent from a previous round instead of the current round, due to communication slowdowns and congestions. Each player uses his/her feedback to make decision and untimely information can significantly influence the players' actions and impair their utilities. Therefore, it is important to understand the sensitivity of learning algorithms in the presence of delayed feedbacks, and how to counteract the negative impacts resulting from them. There has been little literature on finding the NE and QRE in zero-sum polymatrix games under delayed feedbacks, and our work provides some of the first understandings to this important issue under a wide range of delay assumptions.
> > > - Technical novelty and challenges of our results: Our analysis deviates significantly from prior analyses (e.g. (Cen et al., 2021)), and many interesting and novel observations emerge due to the presence of delayed feedback. For example, one of the key error terms $\eta(\overline\pi_i^{(t+1)}-\pi_{i,\tau}^\star)A_i(\overline\pi^{(t+1)}-\pi_{\tau}^\star)$ (see proof of Lemma 2) cancel out when summing over $i\in V$ in the synchronous setting, as a consequence of zero-sum polymatrix game's special property. In the asynchronous setting, however, the corresponding term $\eta\sum_{i\in V}(\overline\pi_i^{(t+1)}-\pi_{i,\tau}^\star)A_i(\overline\pi^{(\kappa_i^{(t)})}-\pi_{\tau}^\star)$ misaligns in the time index of $\overline{\pi}$ and no longer admits straightforward treatment, leading to an additional error proportional to, intuitively speaking, $\eta^2\tau^{-1}\cdot \texttt{delay}$. This necessitates a fairly small choice of $\eta$ and hence slows down the convergence of the single-timescale method. The two-timescale method is motivated by this observation, which enables us to view $\overline\pi^{(\kappa_i^{(t)})}$ as an approximation of $\pi^{(t)}$ and replace the misaligned error terms with $\eta(\overline\pi_i^{(\kappa_i^{(t)})}-\pi_{i,\tau}^\star)A_i(\overline\pi^{(\kappa_i^{(t)})}-\pi_{\tau}^\star)$ --- which makes cancellation once again possible and yields faster convergence. To the best of our knowledge, these technical treatments and observations are indeed novel and nontrivial, and are verified through numerical experiments.
> > >
> > > We will definitely incorporate these useful discussions in the next version. We will greatly appreciate if the reviewer can explicitly point to specific items that are still confusing or unclear that prevent you from recommending our paper, and we'll be happy to engage in further discussions that help to increase your perception of our work.

---

### Official Review · Reviewer_VVFX · 2022-10-25

**Confidence:** 2
**Correctness:** 3
**Technical Novelty And Significance:** 3
**Empirical Novelty And Significance:** 3
**Recommendation:** 6

**Clarity, Quality, Novelty And Reproducibility:**

Some parts of this paper are hard to read:

How to obtain Eq.(8). \pi in QRE and NE may not be the same due to Eq.(5)?

In Eq.(6), \tau is the regularization parameter or temperature, then what is u_i(\tau)?

How to remove t in the rate, i.e., O, before Remark 1?

What is the intuition of Assumptions 1, 2, and 3? Are they realistic in a real-world scenario?

Why does KL increase in Figures 1(b) and 1(c)?

Minor:
At the beginning of Section 1.2, ‘excessive gap technique  of Nesterov’ needs a reference.

**Strength And Weaknesses:**

This paper seems to propose a new problem and then use some technique to solve it, but the property of the proposed problem and intuition behind the technique is unclear:

Why are the zero-sum polymatrix games under delay feedback important? What does ‘delay feedback’ mean in zero-sum polymatrix games? What are real-world scenarios that can be modeled by zero-sum polymatrix games under delay feedback? Why do we need to care about NE and QRE in these games?

What is the challenge (e.g., complexity) for computing the Nash equilibrium and Quantal response equilibrium in zero-sum polymatrix games under delay feedback?

Why does synchronous optimization not work in zero-sum polymatrix games under delay feedback?

Are there any existing algorithms that can solve the new problem?

How to obtain Eq.(8)? \pi in QRE and NE may not be the same due to Eq.(5). Is it possible to show the result for NE instead of using this formula and the results of QRE?

How to remove t in the rate, i.e., O, before Remark 1?

What is the intuition of Assumptions 1, 2, and 3? Are they realistic in real-world scenarios?

Why do we need to consider a two-timescale problem?


**Summary Of The Paper:**

This paper studies asynchronous gradient plays in zero-sum polymatrix games under delayed feedbacks, while significant efforts have been made to understand zero-sum two-player matrix games. They first establish that the last iterate of the entropy-regularized optimistic multiplicative weight updates (OMWU) method converges linearly to the quantal response equilibrium (QRE), the solution concept under bounded rationality, in the absence of delays or under the randomly delayed feedbacks with some assumptions. They further demonstrate that entropy-regularized OMWU with two-timescale learning rates enjoys faster last-iterate convergence under fixed delays and continues to converge provably even when the delays are arbitrarily bounded.

**Summary Of The Review:**

This paper seems to propose a new problem and then use some technique to solve it, but the property of the proposed problem and the intuition behind the technique is unclear.

---

> ### Author Response · Authors · 2022-11-16
> **Response to Reviewer VVFX (1/2)**
>
> We thank the reviewer for the useful feedback!
> > Why are the zero-sum polymatrix games under delay feedback important? What does ‘delay feedback’ mean in zero-sum polymatrix games? What are real-world scenarios that can be modeled by zero-sum polymatrix games under delay feedback?
>
> For delayed feedback, there commonly exist real-life scenarios where the players only receive the payoff information sent from a previous round instead of the current round, due to communication slowdowns and congestions. Each player uses his/her feedback to make decision and untimely information can significantly influence the players' actions and impair their utilities. For example, consider the online security game (Cai et al., 2016), which is modeled by a bipartite graph connecting the evaders and inspectors, with exit points. Each evader chooses one exit point and each inspector chooses one point to check. For each pair of connected inspector and evader, the inspector will get one point for each successful check and the evaders get each for one escape. Due to the servers' communication slowdowns, the players may only receive their feedback of points after many rounds. This is a real-life scenerio that justifies the importance of considering delayed feedback in zero-sum polymatrix games.
>
> > Why do we need to care about NE and QRE in these games?
>
> Finding equilibria is the central goal in game theory, because these equilibria represent certain steady state of the game. For example, when achieving the NE, each player can no longer improve their own self-interests by unilaterally deviating from the NE. The QRE is a natural extension of the NE under bounded rationality, which extends NE by modeling the fact that individuals may only be partially rational when making decisions, and can be motivated when the payoffs suffer from noise (McKelvey & Palfrey, 1995).
>
> > What is the challenge (e.g., complexity) for computing the Nash equilibrium and Quantal response equilibrium in zero-sum polymatrix games under delay feedback?
>
> There is lack of literatures considering finding the NE and QRE in zero-sum polymatrix games, especially when it comes to algorithms using independent and symmetric updates with last-iterate convergence guarantees. Moreover, previous works were mostly limited to the study of individual regret, rather than the convergence to the equilibrium in a multi-player game, or the study of two-player zero-sum games without delayed feedbacks. In terms of technical challenges, we need to carefully study the impact of the delays on the convergence behavior in a quantitative manner, leading to several interesting results. For example, single-timescale OMWU, which works ideally in the case without delays, no longer performs as desired in the presence of delays. Therefore, new algorithms are devised by using two-timescale learning rates to achieve faster convergence rates in the presence of delays. The analysis of these algorithms require careful manipulations which are not present in the analysis without delays.
>
> > Why does synchronous optimization not work in zero-sum polymatrix games under delay feedback?
>
> Synchronous optimization requires retrieving and sharing information across multiple agents, which raises questions in terms of both privacy and efficiency. For example, it requires a central server that collects all players' payoff matrices and makes decisions for each player, which is impossible as the number of players grow to thousands or even to millions and with supervision of user protection legacy.
>
> > Are there any existing algorithms that can solve the new problem?
>
> To the best of our knowledge, there is no literature taking delayed feedback into consideration of finding NE and QRE in zero-sum polymatrix games, in particular when it comes to algorithms with independent and symmetric updates with provable last-iterate convergence guarantees.
>
> > How to obtain Eq.(8)? $\pi$ in QRE and NE may not be the same due to Eq.(5). Is it possible to show the result for NE instead of using this formula and the results of QRE?
>
> To see why the relation holds, note that
> \begin{aligned}
> \\texttt{NE-Gap}(\\pi) & =
> \\max_{i\\in V} [ \\max_{\\pi_i'\\in \\Delta(S_i)} u_{i}(\\pi_i', \\pi_{-i}) - u_{i}(\\pi) ] \\\\
> & =  \max_{i\in V} [ \max_{\pi_i'\in \Delta(S_i)} u_{i,\tau}(\pi_i', \pi_{-i}) - u_{i,\tau}(\pi)+\tau(\pi_i')^\top \log\pi_i'-\tau\pi_i^\top\log\pi_i ]  \\\\
>  &		\\le \\max_{i\\in V} [ \\max_{\\pi_i'\\in \\Delta(S_i)} u_{i,\\tau}(\\pi_i', \\pi_{-i}) - u_{i,\\tau}(\\pi) ]+ \\tau \\log S_{\\max}  \\\\
>  &= \\texttt{QRE-Gap}\_{\\tau}(\\pi)+\\tau\\log S_{\\max} .
> \\end{aligned}
> Therefore, an $\epsilon/2$-QRE is also an $\epsilon$-NE by setting $\tau=\frac{\epsilon}{2\max_{i\in V}\log|S_i|}$. However, directly solving NE is generally difficult even in the two-player game. They either require the uniqueness of the NE or has quite pessimistic rate of convergence (see for example, the literature discussion in (Cen et al, NeurIPS 2021)).

---

> ### Author Response · Authors · 2022-11-16
> **Response to Reviewer VVFX (2/2)**
>
> > How to remove t in the rate, i.e., O, before Remark 1?
>
> The iteration complexity to finding $\epsilon$-QRE follows from setting $(1-\eta \tau)^T\lesssim \epsilon$ (ignoring non-important terms), leading to $T\gtrsim \widetilde{\mathcal{O}}(\frac{1}{\eta\tau}\log\frac{1}{\epsilon})$. Translating to NE by setting $\tau \asymp \epsilon$ and $1/\\eta \\asymp d\_{\\max}\\|A\\|\_{\\infty}$, we obtain an iteration complexity of $\widetilde{\mathcal{O}}(\frac{d_{\max}\|A\|_{\infty}}{\epsilon})$ for finding an $\epsilon$-NE. Other iteration complexities can be simplified similarly.
>
> > What is the intuition of Assumptions 1, 2, and 3? Are they realistic in real-world scenarios?
>
> They are common settings in the analysis of asynchronous learning algorithms as we pointed out in the article. Assumption 1 can apply to scenarios where the delay comes in a Poisson process, which is the most common setting for service systems and independent arrivals. The boundedness of delay is also common in real life, since the game servers will give guarantees of the players' service quality. Assumption 3 ensures that the players will not receive repeated feedback despite delay. This simulates the data packages in communications.
>
> > Why do we need to consider a two-timescale problem?
>
> As we have shown in Section 3, the performance of single-timescale OMWU is impaired by the delayed feedback. By applying a two-timescale OMWU, we adopt a larger learning rate for the extrapolation step to counteract the delay, which allows us to achieve a faster convergence in Theorem 3. In addition, we also obtain performance guarantees of two-timescale OMWU under more general delay assumptions, which can cover more application scenarios of interest.
>
> In the more technical perspective, the motivation is to better control the terms such as $\sum_{i\in V}(\overline\pi_i^{(t+1)}-\pi_{i,\tau}^\star)A_i(\overline\pi^{(\kappa_i^{(t)})}-\pi_{\tau}^\star)$. As $\kappa_i^{(t)}$'s are different for different $i\in V$, we need to adjust the predictive step sizes in order to cancel out the negative effects of such terms.
>
> > In Eq.(6), $\tau$ is the regularization parameter or temperature, then what is $u_i(\tau)$?
>
> Thanks for the comment! This is a typo, we should have written $u_i(\pi)$ rather than $u_i(\tau)$.
>
> > Why does KL increase in Figures 1\(b\) and 1\(c\)?
>
> Fig. 1 \(b\) and \(c\) compare the effect of different choices of learning rates $\eta,\overline\eta$ on the performance of the proposed methods, where the feedback is permuted with bounded delay $\gamma=25$, rather than the iterative curves with varying iterations. It demonstrates that two-timescale OMWU outperforms single-timescale OMWU given appropriate choices of learning rate $\eta$. On the other hand, \(c\) demonstrates that the choice of $\overline\eta=\tau^{-1}(1-(1-\eta\tau)^{\gamma+1})$ suggested by the theory indeed leads to near-optimal performance of two-timescale OMWU.
>
> > At the beginning of Section 1.2, ‘excessive gap technique of Nesterov’ needs a reference.
>
> Thank you for the comment. We have added references to this part.

---

### Author Response · Authors · 2022-11-16
**Response to all reviewers**

We thank all the reviewers for their time and thoughtful reviews and suggestions. We have updated a version of our paper where we have polished the writing and sharpened some proof. We have strenghtened the motivation on why it is necessary to study learning rate separation in the presence of delayed feedback, and highlighted our contributions. We take this opportunity to stress our key results as follows.

* We have highlighted that single-timescale OMWU, while allows last-iterate linear convergence to the QRE under random delays, **suffers from a slowdown of the rate** due to a much smaller allowable range of learning rates, and therefore calls for new algorithm designs. In particular, when converting to finding an $\epsilon$-NE, the rate degenerates to $\widetilde{\mathcal{O}}(\epsilon^{-2})$ ignoring other salient parameters, which is an order worse than the bound $\widetilde{\mathcal{O}}(\epsilon^{-1})$ without delays.
* We further demonstrate that by using two-timescale OMWU, where the learning rates separation leverages a known and fixed delay in a delay-aware manner, the convergence rate can be **significantly accelerated**. In particular, when converting to finding an $\epsilon$-NE, the rate recovers to $\widetilde{\mathcal{O}}(\epsilon^{-1})$ ignoring other salient parameters, which matches the rate of $\widetilde{\mathcal{O}}(\epsilon^{-1})$ without delays.
* Furthermore, we establish the finite-time convergence of the average-iterate (as well as the asympototic convergence of the last-iterate) of two-timescale OMWU under arbitrarily permutated bounded delay, where the delay sequence can be even adversarial. This gives evidence to the robustness of asynchonous gradient play even under the argubly worst delay scenarios, when using two-timescale learning rates. In this revision, we have further **improved the rate** from $\widetilde{\mathcal{O}}(\epsilon^{-4})$ to $\widetilde{\mathcal{O}}(\epsilon^{-3})$ for finding an $\epsilon$-NE.

Note that previous work (Cen et al, 2021) on entropy-regularized OMWU only studied the single-timescale version for two-player zero-sum games, while we significantly go beyond this by looking at multi-agent games in the presence of various kinds of delay assumptions, which elucidates the limitation of single-timescale OMWU in the presence of delays, and necessitates the introduction of two-timescale OMWU.

---

### Decision · Program_Chairs · 2023-01-20

**Decision:**

Accept: poster

**Justification For Why Not Higher Score:**

The paper had several issues that would not warrant higher exposure.

**Justification For Why Not Lower Score:**

Despite its problems, the paper provided new results which would be of interest to several ML subcommunities.

**Metareview: Summary, Strengths And Weaknesses:**

This paper studies the convergence of gradient-based algorithms to quantal response equilibria in two-player zero-sum games (and their polymatrix generalizations). The algorithm studied by the authors is based on an entropic regularization template that was first studied in a game-theoretic setting by [1,2], and provides a range of convergence results for both fixed and random delays.

The delay model studied by the authors is not the one commonly followed in the literature. For example, in the standard works [3,4,5,6], it is assumed that an action at time $t$ triggers a delay $d_t$, so the information from action $t$ is observed at time $t+d_t$. By contrast, the authors assume that, at time $t$, a player observes information from round $t-d_t$. The two models may appear superficially similar but, in fact, they are fundamentally different: in the former, there may be rounds where the learner receives no information, others where information from many rounds is received at once, but the same piece of information is never received twice; in the latter, the learner always receives one piece of information from some previous round, but may receive the exact same piece of information several times. While the former model of delays has been extensively studied in the literature, the authors' model is much less common, and the authors did not discuss it at sufficient length - a fact which was criticized during the committee discussion phase.

Another issue concerns the fact that the authors seem to ignore several recent results regarding the convergence of asynchronous gradient play in games, see e.g., [7,8] below.

Despite these criticisms, the committee felt that the paper cleared the acceptance bar (albeit marginally so), so there were no objections to an accept recommendation - but with the explicit understanding that the above issues should be included in the camera-ready version of the paper.


*References*

1. David S. Leslie and E. J. Collins, Individual Q-learning in normal form games, SIAM Journal on Control and Optimization 44 (2005), no. 2, 495–514.
1. Pierre Coucheney, Bruno Gaujal, and Panayotis Mertikopoulos, Penalty-regulated dynamics and robust learning procedures in games, Mathematics of Operations Research 40 (2015), no. 3, 611– 633.
1. Kent Quanrud and Daniel Khashabi, Online learning with adversarial delays, NIPS ’15: Proceedings of the 29th International Conference on Neural Information Processing Systems, 2015.
1. Pooria Joulani, András György, and Csaba Szepesvári, Online learning under delayed feedback, ICML ’13: Proceedings of the 30th International Conference on Machine Learning, 2013.
1. Pooria Joulani, András György, and Csaba Szepesvári, Delay-tolerant online convex optimization: Unified analysis and adaptive-gradient algorithms, AAAI ’16: Proceedings of the 30th Conference on Artificial Intelligence, 2016.
1. Claire Vernade, Olivier Cappé, and Vianney Perchet, Stochastic bandit models for delayed conversions, UAI’ 17: Proceedings of the 33rd Annual Conference on Uncertainty in Artificial Intelligence, 2017.
1. Amélie Héliou, Panayotis Mertikopoulos, and Zhengyuan Zhou, Gradient-free online learning in continuous games with delayed rewards, ICML ’20: Proceedings of the 37th International Conference on Machine Learning, 2020.
1. Zhengyuan Zhou, Panayotis Mertikopoulos, Nicholas Bambos, Peter W. Glynn, and Claire Tomlin, Countering feedback delays in multi-agent learning, NIPS ’17: Proceedings of the 31st International Conference on Neural Information Processing Systems, 2017.

**Note From Pc:**

if the above contains the word "oral" or "spotlight" please see: "oral" presentation means -> notable-top-5% and "spotlight" means -> notable-top-25%. As stated in our emails, we are disassociating presentation type from AC recommendations

**Summary Of Ac-Reviewer Meeting:**

This was not a borderline paper.